# Mini-Batch Optimization of Contrastive Loss

**Jaewoong Cho**[*]                                         *jwcho@krafton.com*
*KRAFTON*

**Kartik Sreenivasan**[*]                    *kartik.sreenivasan@databricks.com*
*Databricks*

**Keon Lee**                                               *keonlee@krafton.com*
*KRAFTON*

**Kyunghoo Mun**                                        *kmun@andrew.cmu.edu*
*Carnegie Mellon University*

**Soheun Yi**                                          *soheuny@andrew.cmu.edu*
*Carnegie Mellon University*

**Jeong-Gwan Lee**                               *jeonggwan.lee@krafton.com*
*KRAFTON*

**Anna Lee**                                              *annalee@krafton.com*
*KRAFTON*

**Jy-yong Sohn**                                     *jysohn1108@gmail.com*
*Yonsei University*

**Dimitris Papailiopoulos**                              *dimitris@papail.io*
*University of Wisconsin-Madison*

**Kangwook Lee**                                      *kangwook.lee@wisc.edu*
*University of Wisconsin-Madison*
*KRAFTON*

**Reviewed on OpenReview:** *https://openreview.net/forum?id=Nux7OVXpJ9*

## Abstract

Contrastive learning has gained significant attention as a pre-training method for self-supervised learning due to its ability to leverage large amounts of unlabeled data. A contrastive loss function ensures that embeddings of positive sample pairs (e.g., from the same class or different views of the same data) are similar, while embeddings of negative pairs are dissimilar. However, practical constraints such as large memory requirements make it infeasible to consider all possible positive and negative pairs, leading to the use of mini-batches. In this paper, we investigate the theoretical aspects of mini-batch optimization in contrastive learning with the InfoNCE loss. We show that mini-batch optimization is equivalent to full-batch optimization if and only if all $\binom{N}{B}$ mini-batches are selected, while sub-optimality may arise when examining only a subset. We then demonstrate that utilizing high-loss mini-batches can speed up SGD convergence and propose a spectral clustering-based approach for identifying these high-loss mini-batches. Our experimental results validate our theoretical findings and demonstrate that our proposed algorithm outperforms vanilla SGD, providing a better understanding of mini-batch optimization in contrastive learning.

---

[*]Equal Contributions.

# 1 Introduction

Contrastive learning is commonly used as a pre-training method for self-supervised learning, due to its ability to leverage the vast amount of freely available unlabeled data (Jaiswal et al., 2020). The contrastive loss function is designed to ensure that the embeddings of two samples are similar if they are considered a "positive" pair, in cases such as coming from the same class (Khosla et al., 2020), being an augmented version of one another (Chen et al., 2020a), or being two different modalities of the same data (Radford et al., 2021). Conversely, if two samples do not form a positive pair, they are considered a "negative" pair, and the contrastive loss encourages their embeddings to be dissimilar.

In practice, it is not feasible to consider all possible positive and negative pairs when implementing a contrastive learning algorithm due to the quadratic memory requirement $\mathcal{O}(N^2)$ when working with $N$ samples. To mitigate this issue of *full-batch training*, practitioners typically choose a set of *mini-batches* of size $B = \mathcal{O}(1)$, and consider the loss computed for positive and negative pairs within each of the $N/B$ batches (Chen et al., 2022; 2020a; Hu et al., 2021; Zeng et al., 2021; Chen et al., 2021; Zolfaghari et al., 2021; Gadre et al., 2023). For instance, Gadre et al. (2023) train a model on a dataset where $N = 1.28 \times 10^7$ with $B = 4096$. This approach results in a memory requirement of $\mathcal{O}(B^2) = \mathcal{O}(1)$ for each mini-batch, and a total computational complexity linear in the number of chosen mini-batches. Despite the widespread practical use of mini-batch optimization in contrastive learning, there remains a lack of theoretical understanding as to whether this approach is truly reflective of the original goal of minimizing *full-batch* contrastive loss. This paper examines the theoretical aspects of optimizing mini-batches loaded for contrastive learning with the InfoNCE loss (Oord et al., 2018). Our study primarily focuses on this loss function due to its fundamental importance in contrastive learning and its analytical tractability.

**Main Contributions.** The main contributions of this paper are twofold. First, we show that under certain parameter settings, mini-batch optimization is equivalent to full-batch optimization if and only if all $\binom{N}{B}$ mini-batches are selected. The results are based on an interesting connection between contrastive learning and the neural collapse (Lu & Steinerberger, 2022). From a computational complexity perspective, the identified equivalence condition may be seen as somewhat prohibitive, as it implies that all $\binom{N}{B} = \mathcal{O}(N^B)$ mini-batches must be considered.

Our second contribution, however, shows that Ordered SGD (OSGD) algorithm (Kawaguchi & Lu, 2020) can be effective in finding mini-batches that contain the most informative pairs and thereby speeding up convergence. We show that the convergence result from Kawaguchi & Lu (2020) can be applied directly to contrastive learning. We also show that OSGD can improve the convergence rate of SGD by a constant factor in certain scenarios. Furthermore, in a novel approach to address the challenge of applying OSGD to the $\binom{N}{B}$ mini-batch optimization (which involves examining $\mathcal{O}(N^B)$ batches to select high-loss ones), we reinterpret the batch selection as a min-cut problem in graph theory. This transformative shift empowers us to select high-loss batches efficiently via a spectral clustering algorithm established in this field. The following informal theorems summarize the main findings.

**Theorem 1** (informal). *Under certain parameter settings, the mini-batch optimization is equivalent to full-batch optimization if and only if all $\binom{N}{B}$ mini-batches are selected. Although $\binom{N}{B}$ mini-batch contrastive loss and full-batch loss are neither identical nor differ by a constant factor, the optimal solutions for both mini-batch and full-batch are identical (see Section 4).*

**Theorem 2** (informal). *In a demonstrative toy example, OSGD operating on the principle of selecting high-loss batches, can potentially converge to the optimal solution of mini-batch contrastive loss optimization faster by a constant factor compared to SGD (see Section 5.1).*

We validate our theoretical findings and the efficacy of the proposed spectral clustering-based batch selection method by conducting experiments on both synthetic and real data. On synthetic data, we show that our proposed batch-selection algorithms do indeed converge to the optimal solution of *full-batch optimization* significantly faster than the baselines. We also conduct experiments by pre-training on CIFAR-100 (Krizhevsky et al., 2009) and Tiny ImageNet (Le & Yang, 2015) using the proposed method. We evaluate the performance of the subsequent models on retrieval tasks. Our experiments on real datasets demonstrate that our batch selection method outperforms vanilla SGD in practically relevant settings.

## 2 Related Work

**Contrastive losses.** Contrastive learning has been used for several decades to learn a similarity metric to be used later for verification or recognition applications (Misra & Maaten, 2020; Aberdam et al., 2021). Chopra et al. (2005) proposed one of the early versions of contrastive loss which has been updated and improved over the years (Sohn, 2016; Song & Ermon, 2020; Schroff et al., 2015; Khosla et al., 2020; Oord et al., 2018). More recently, contrastive learning has been shown to rival and even surpass traditional supervised learning methods, particularly on image classification tasks (Chen et al., 2020b; Bachman et al., 2019). Further, its multi-modal adaptation leverages vast unstructured data, extending its effectiveness beyond image and text modalities (Radford et al., 2021; Jia et al., 2021; Pham et al., 2021; Ma et al., 2021; Sachidananda et al., 2022; Elizalde et al., 2023; Goel et al., 2022; Lee et al., 2022; Ramesh et al., 2021; 2022). Unfortunately, these methods require extremely large batch sizes in order to perform effectively. Follow-up works showed that using momentum or carefully modifying the augmentation schemes can alleviate this issue to some extent (He et al., 2020; Chen et al., 2020b; Grill et al., 2020; Wang & Qi, 2022).

**Effect of batch size.** While most successful applications of contrastive learning use large batch sizes (*e.g.*, 32,768 for CLIP and 8,192 for SimCLR), recent efforts have focused on reducing batch sizes and improving convergence rates (Yeh et al., 2022; Chen et al., 2022). Yuan et al. (2022) carefully studied the effect of the requirements on the convergence rate when a model is trained for minimizing SimCLR loss, and proved that the gradient of the solution is bounded by $\mathcal{O}(\frac{1}{\sqrt{B}})$. They also propose SogCLR, an algorithm with a modified gradient update where the correction term allows for an improved convergence rate with better dependence on $B$. It is shown that the performance for small batches can be improved with the technique called hard negative mining (Robinson et al., 2021; Kalantidis et al., 2020; Zhang & Stratos, 2021).

**Neural collapse.** Neural collapse is a phenomenon observed in Papyan et al. (2020) where the final classification layer of deep neural nets collapses to the simplex Equiangular Tight Frame (ETF) when trained well past the point of zero training error (Ji et al., 2022; Zhou et al., 2022). Lu & Steinerberger (2022) proved that this occurs when minimizing cross-entropy (CE) loss over the unit ball. Jiang et al. (2023) generalized neural collapse to practically relevant settings where the number of classes is larger than the dimension of embeddings. Extending beyond CE loss minimization, Graf et al. (2021) delved into the optimal embeddings by minimizing supervised contrastive (SC) loss in balanced datasets and compared them with CE loss minimization. Kini et al. (2023) extended this study by characterizing the optimal embeddings of SC loss minimization combined with ReLU in the last layer, which can learn symmetric embedding structures despite data imbalances, and the sufficient and necessary conditions for mini-batch to achieve the same optimal embeddings of full-batch optimization. Our work explores this problem for the unsupervised contrastive learning setting. We characterize (1) the optimal embeddings that minimize the InfoNCE loss for any given temperature parameter under certain assumptions, and (2) conditions for mini-batch to achieve the same optimal embeddings of full-batch optimization.

**Optimal permutations for SGD.** The performance of SGD without replacement under different permutations of samples has been well studied in the literature (Bottou, 2009; Recht & Re, 2012; Recht & Ré, 2013; Nagaraj et al., 2019; Ying et al., 2020; Ahn et al., 2020; Rajput et al., 2020; Mishchenko et al., 2020; Safran & Shamir, 2021b;a; Gürbüzbalaban et al., 2021; Nguyen et al., 2021; Lu et al., 2021; Rajput et al., 2022; Tran et al., 2021; Lu et al., 2022; Cha et al., 2023; Cho & Yun, 2023). One can view batch selection in contrastive learning as a method to choose a specific permutation among the possible $\binom{N}{B}$ mini-batches of size $B$. However, it is important to note that these bounds do not indicate an improved convergence rate for general non-convex functions and thus would not apply to the contrastive loss, particularly in the setting where the embeddings come from a shared embedding network. We show that in the case of OSGD (Kawaguchi & Lu, 2020), we can indeed prove that contrastive loss satisfies the necessary conditions in order to guarantee convergence.

## 3  Problem Setting

Suppose we are given a dataset $\{(\boldsymbol{x}_i, \boldsymbol{y}_i)\}_{i=1}^N$ of $N$ positive pairs (data sample pairs that are conceptually similar or related), where $\boldsymbol{x}_i$ and $\boldsymbol{y}_i$ are two different *views* of the same data. Note that this setup includes both the multi-modal setting (*e.g.*, CLIP (Radford et al., 2021)) and the uni-modal setting (*e.g.*, SimCLR (Chen et al., 2020a)) as follows. For the multi-modal case, one can view $(\boldsymbol{x}_i, \boldsymbol{y}_i)$ as two different modalities of the same data, e.g., $\boldsymbol{x}_i$ is the image of a scene while $\boldsymbol{y}_i$ is the text description of the scene. For the uni-modal case, one can consider $\boldsymbol{x}_i$ and $\boldsymbol{y}_i$ as different augmented images from the same image.

We consider the contrastive learning problem where the goal is to find embedding vectors for $\{\boldsymbol{x}_i\}_{i=1}^N$ and $\{\boldsymbol{y}_i\}_{i=1}^N$, such that the embedding vectors of positive pairs $(\boldsymbol{x}_i, \boldsymbol{y}_i)$ are similar, while ensuring that the embeddings of other (negative) pairs are well separated. Let $\boldsymbol{u}_i \in \mathbb{R}^d$ be the embedding vector of $\boldsymbol{x}_i$, and $\boldsymbol{v}_i \in \mathbb{R}^d$ be the embedding vector of $\boldsymbol{y}_i$. In practical settings, one typically considers parameterized encoders so that $\boldsymbol{u}_i = f_{\boldsymbol{\theta}}(\boldsymbol{x}_i)$ and $\boldsymbol{v}_i = g_{\boldsymbol{\phi}}(\boldsymbol{y}_i)$. We define embedding matrices $\boldsymbol{U} := [\boldsymbol{u}_1, \dots \boldsymbol{u}_N]$ and $\boldsymbol{V} := [\boldsymbol{v}_1, \dots, \boldsymbol{v}_N]$ which are the collections of embedding vectors. Now, we focus on the setting of directly optimizing the embeddings instead of model parameters $\boldsymbol{\theta}$ and $\boldsymbol{\phi}$ in order to gain theoretical insights into learning embeddings. Note that this setting is commonly utilized to achieve a deeper understanding of the underlying principles and mechanisms (Graf et al., 2021; Kini et al., 2023). Consider the problem of directly optimizing the embeddings for $N$ pairs which is given by

$$\begin{aligned}
\min_{U,V} \quad & \mathcal{L}^{\mathrm{con}}(U, V) \\
\text{s.t.} \quad & \|\boldsymbol{u}_i\| = 1, \|\boldsymbol{v}_i\| = 1 \quad \forall i \in [N],
\end{aligned} \tag{1}$$

where $\|\cdot\|$ denotes the $\ell_2$ norm, the set $[N]$ denotes the collection of all integers from 1 to $N$. In this work, we focus on the standard InfoNCE loss (Oord et al., 2018) defined as

$$\mathcal{L}^{\mathrm{con}}(\boldsymbol{U}, \boldsymbol{V}) := -\frac{1}{N} \sum_{i=1}^N \log\left(\frac{e^{\boldsymbol{u}_i^\intercal \boldsymbol{v}_i / \tau}}{\sum_{j=1}^N e^{\boldsymbol{u}_i^\intercal \boldsymbol{v}_j / \tau}}\right) - \frac{1}{N} \sum_{i=1}^N \log\left(\frac{e^{\boldsymbol{v}_i^\intercal \boldsymbol{u}_i / \tau}}{\sum_{j=1}^N e^{\boldsymbol{v}_i^\intercal \boldsymbol{u}_j / \tau}}\right), \tag{2}$$

with $\tau$ being a positive scalar known as a temperature parameter. Note that $\mathcal{L}^{\mathrm{con}}(\boldsymbol{U}, \boldsymbol{V})$ is the full-batch version of the loss which contrasts all embeddings with each other. However, due to the large computational complexity and memory requirements during optimization, practitioners often consider the following mini-batch version instead. Note that there exist $\binom{N}{B}$ different mini-batches, each of which has $B$ samples. For $k \in \left[\binom{N}{B}\right]$, let $\mathcal{B}_k$ be the $k$-th mini-batch satisfying $\mathcal{B}_k \subset [N]$ and $|\mathcal{B}_k| = B$. Let $\boldsymbol{U}_{\mathcal{B}_k} := \{\boldsymbol{u}_i\}_{i \in \mathcal{B}_k}$ and $\boldsymbol{V}_{\mathcal{B}_k} := \{\boldsymbol{v}_i\}_{i \in \mathcal{B}_k}$. Then, the contrastive loss for the $k$-th mini-batch is $\mathcal{L}^{\mathrm{con}}(\boldsymbol{U}_{\mathcal{B}_k}, \boldsymbol{V}_{\mathcal{B}_k})$.

## 4  Optimization for Full-Batch and Mini-Batch

In this section, we investigate the relationship between the problem of optimizing the full-batch loss $\mathcal{L}^{\mathrm{con}}(\boldsymbol{U}, \boldsymbol{V})$ and the problem of optimizing the mini-batch loss $\mathcal{L}^{\mathrm{con}}(\boldsymbol{U}_{\mathcal{B}_k}, \boldsymbol{V}_{\mathcal{B}_k})$. Towards this goal, we prove three main results, the proof of which are in Appendix C.1.

- We derive the optimal solution that minimizes the full-batch loss (Lemma 1, Theorem 3).

- We show that the solution that minimizes the average of $\binom{N}{B}$ mini-batch losses is identical to the one that minimizes the full-batch loss (Proposition 1, Theorem 4).

- We show that minimizing the mini-batch loss summed over only a strict subset of $\binom{N}{B}$ mini-batches can lead to a sub-optimal solution that does not minimize the full-batch loss (Theorem 5).

### 4.1  Full-batch Contrastive Loss Optimzation

In this section, we characterize the optimal solution for the full-batch loss minimization in Equation (1). We start by providing the definition of the simplex equiangular tight frame (ETF) which turns out to be the

optimal solution in certain cases. The original definition of ETF (Sustik et al., 2007) is for $N$ vectors in a $d$-dimensional space where $N \geq d+1$ [1]. Papyan et al. (2020) define the ETF for the case where $N \leq d+1$ to characterize the phenomenon of neural collapse. In our work, we will introduce and use the latter definition of simplex ETFs which is stated below.

**Definition 1** (Simplex ETF). We call a set of $N$ vectors $\{\boldsymbol{u}_i\}_{i=1}^N$ form a simplex Equiangular Tight Frame (ETF) if $\|\boldsymbol{u}_i\| = 1, \forall i \in [N]$ and $\boldsymbol{u}_i^\mathsf{T} \boldsymbol{u}_j = -1/(N-1), \forall i \neq j$.

In the following Lemma, we first prove that the optimal solution of full-batch contrastive learning is the simplex ETF for $N \leq d+1$ which follows almost directly from Lu & Steinerberger (2022).

**Lemma 1** (Optimal solution when $N \leq d+1$). *Suppose $N \leq d+1$. Then, the optimal solution $(\boldsymbol{U}^\star, \boldsymbol{V}^\star)$ of the full-batch contrastive learning problem in Equation (1) satisfies two properties: (i) $\boldsymbol{U}^\star = \boldsymbol{V}^\star$, and (ii) the columns of $\boldsymbol{U}^\star$ form a simplex ETF.*

Recall that in this work, we explore the setting of directly optimizing embeddings as an initial effort to gain theoretical insight. However, Lemma 1 presents the result that can be extended to a more practically relevant setting where embeddings are generated through linear encoders parameterized by $\theta$ and $\phi$: $\boldsymbol{u}_i = \boldsymbol{W}_\theta \boldsymbol{x}_i$ and $\boldsymbol{v}_i = \boldsymbol{W}_\phi \boldsymbol{y}_i$. The details of this setting and the characteristics of the optimal encoders are stated below.

**Corollary 1.** *Let us consider a setting where embedding vectors are generated via overparameterized linear encoders $\boldsymbol{W}_\theta, \boldsymbol{W}_\phi \in \mathbb{R}^{d \times d_0}$: $\boldsymbol{u}_i = \boldsymbol{W}_\theta \boldsymbol{x}_i$, $\boldsymbol{v}_i = \boldsymbol{W}_\phi \boldsymbol{y}_i$, where $\boldsymbol{x}_i, \boldsymbol{y}_i \in \mathbb{R}^{d_0}$. We define $\boldsymbol{X} := [\boldsymbol{x}_1, \ldots, \boldsymbol{x}_N]$ and $\boldsymbol{Y} := [\boldsymbol{y}_1, \ldots, \boldsymbol{y}_N]$. Suppose $d, d_0 \geq N-1$ and $\boldsymbol{X}^\mathsf{T}\boldsymbol{X}, \boldsymbol{Y}^\mathsf{T}\boldsymbol{Y}$ are invertible. Then, the optimal solution $(\boldsymbol{W}_\theta^*, \boldsymbol{W}_\phi^*)$ is given as: (i) $\boldsymbol{W}_\theta^* = \boldsymbol{U}^\star(\boldsymbol{X}^\mathsf{T}\boldsymbol{X})^{-1}\boldsymbol{X}^\mathsf{T}$; (ii) $\boldsymbol{W}_\phi^* = \boldsymbol{V}^\star(\boldsymbol{Y}^\mathsf{T}\boldsymbol{Y})^{-1}\boldsymbol{Y}^\mathsf{T}$, where $(\boldsymbol{U}^\star, \boldsymbol{V}^\star)$ satisfies the two properties in Lemma 1.*

Several practical scenarios satisfy $N > d+1$. However, the approach used in Lu & Steinerberger (2022) cannot be directly applied for $N > d+1$, leaving it as an open problem. While solving the open problem for the general case seems difficult, we characterize the optimal solution for the specific case of $N = 2d$, subject to the conditions stated below. However, the characterization for this case is still challenging, so we employ a certain conjecture; we conjecture that the optimal embeddings $(\boldsymbol{U}^\star, \boldsymbol{V}^\star)$ for $N = 2d$ are *symmetric* and *antipodal*. These properties of embeddings are defined as follows.

**Definition 2** (Symmetric and Antipodal). Embedding matrices $\boldsymbol{U}$ and $\boldsymbol{V}$ are called *symmetric and antipodal* if $(\boldsymbol{U}, \boldsymbol{V})$ satisfies two properties: (i) Symmetric *i.e.*, $\boldsymbol{U} = \boldsymbol{V}$; (ii) Antipodal *i.e.*, for each $i \in [N]$, there exists $j(i)$ such that $\boldsymbol{u}_{j(i)} = -\boldsymbol{u}_i$.

Note that the symmetric property also holds for the optimal embeddings when $N \leq d+1$, and the antipodal property is a common assumption in geometric problems such as the sphere covering problem in Borodachov (2022). This provides a rationale for our conjecture.

Theorem 3 shows that when $N = 2d$, the optimal solution for the full-batch loss minimization, under a *symmetric* and *antipodal* configuration, form a cross-polytope which is defined as the following.

**Definition 3** (Simplex cross-polytope). We call a set of $N$ vectors $\{\boldsymbol{u}\}_{i=1}^N$ form a simplex cross-polytope if, for all $i$, the following three conditions hold: $\|\boldsymbol{u}_i\| = 1$; there exists a unique $j$ such that $\boldsymbol{u}_i^\mathsf{T}\boldsymbol{u}_j = -1$; and $\boldsymbol{u}_i^\mathsf{T}\boldsymbol{u}_k = 0$ for all $k \notin \{i, j\}$.

**Theorem 3** (Optimal solution when $N = 2d$). *Let*

$$(\boldsymbol{U}^\star, \boldsymbol{V}^\star) := \arg \min_{(\boldsymbol{U}, \boldsymbol{V}) \in \mathcal{A}} \mathcal{L}^{\mathrm{con}}(\boldsymbol{U}, \boldsymbol{V}) \tag{3}$$

$$s.t. \quad \|\boldsymbol{u}_i\| = 1, \|\boldsymbol{v}_i\| = 1 \quad \forall i \in [N],$$

*where $\mathcal{A} := \{(\boldsymbol{U}, \boldsymbol{V}) : \boldsymbol{U}, \boldsymbol{V}$ are symmetric and antipodal$\}$. Then, the columns of $\boldsymbol{U}^\star$ form a simplex cross-polytope for $N = 2d$.*

*Proof Outline.* By the antipodality assumption, we can apply Jensen's inequality to $N-2$ indices without itself $\boldsymbol{u}_i$ and antipodal point $-\boldsymbol{u}_i$ for a given $i \in [N]$. Then we show that the simplex cross-polytope

---

[1]See Def. 4 in Appendix B for the full definition

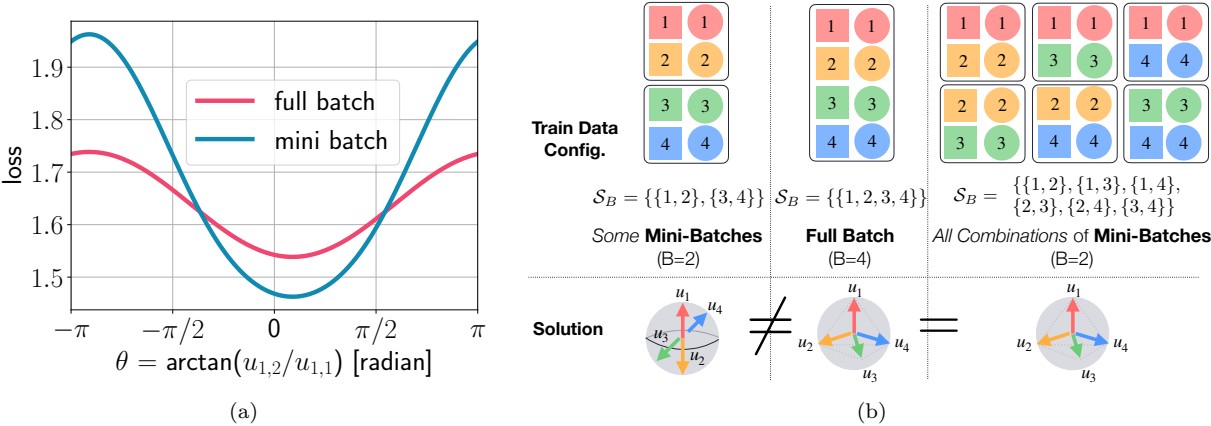

Figure 1: (a) Comparing mini-batch loss and full-batch loss when $N = 10$, $B = 2$, and $d = 2$. We illustrate this by manipulating a single embedding vector $\boldsymbol{u}_1$ while maintaining all other embeddings ($\boldsymbol{v}_1$ and $\{\boldsymbol{u}_i, \boldsymbol{v}_i\}_{i=2}^{10}$) at their optimal solutions. Specifically, $\boldsymbol{u}_1 = [u_{1,1}, u_{1,2}]$ is varied as $[\cos(\theta), \sin(\theta)]$ for $\theta \in [-\pi, \pi]$. While the two loss functions are not identical, corroborating Proposition 1, their minimizers align, providing empirical support for Theorem 4; (b) The relationship between full-batch and mini-batch optimization in contrastive learning. Consider optimizing $N = 4$ pairs of $d = 3$ dimensional embedding vectors $\{(\boldsymbol{u}_i, \boldsymbol{v}_i)\}_{i=1}^{N}$ where $\boldsymbol{u}_i$ and $\boldsymbol{v}_i$ are shown as colored square and circle, respectively. The index $i$ is written in the square/circle. The black rounded box represents a batch. We compare three batch selection options: (i) full batch, *i.e.*, $B = 4$, (ii) all $\binom{N}{B} = 6$ mini-batches with size $B = 2$, and (iii) some mini-batches. Here, $\mathcal{S}_B$ is the set of mini-batches where each mini-batch is represented by the set of constituent samples' indices. Our theoretical/empirical findings are: the optimal embedding that minimizes full-batch loss and the one that minimizes the sum of $\binom{N}{B}$ mini-batch losses are identical, while the one that minimizes the mini-batch losses summed over only a strict *subset* of $\binom{N}{B}$ batches does not guarantee the negative correlation between $\boldsymbol{u}_i$ and $\boldsymbol{u}_j$ for $i \neq j$. This illustration is supported by our mathematical results in Thms. 4 and 5.

minimizes this lower bound while satisfying the conditions that make the applications of Jensen's inequality tight. See Appendix C for the full proof.

For the general case of $N > d + 1$, excluding $N = 2d$, we still leave it as an open problem.

## 4.2 Mini-batch Loss Optimization

Here we consider the mini-batch contrastive loss optimization problem, where we first choose multiple mini-batches of size $B$ and then find $\boldsymbol{U}, \boldsymbol{V}$ that minimize the sum of contrastive losses computed for the chosen mini-batches. Note that this is the loss that is typically considered in the contrastive learning since computing the full-batch loss is intractable in practice. Let us consider a subset of all possible $\binom{N}{B}$ mini-batches and denote their indices by $\mathcal{S}_B \subseteq \left[\binom{N}{B}\right]$. For a fixed $\mathcal{S}_B$, the mini-batch optimization problem is formulated as:

$$\min_{\boldsymbol{U}, \boldsymbol{V}} \quad \mathcal{L}_{\text{mini}}^{\text{con}}(\boldsymbol{U}, \boldsymbol{V}; \mathcal{S}_B) \tag{4}$$

$$\text{s.t.} \quad \|\boldsymbol{u}_i\| = 1, \|\boldsymbol{v}_i\| = 1 \quad \forall i \in [N],$$

where the loss of given mini-batches is $\mathcal{L}_{\text{mini}}^{\text{con}}(\boldsymbol{U}, \boldsymbol{V}; \mathcal{S}_B) := \frac{1}{|\mathcal{S}_B|} \sum_{i \in \mathcal{S}_B} \mathcal{L}^{\text{con}}(\boldsymbol{U}_{\mathcal{B}_i}, \boldsymbol{V}_{\mathcal{B}_i})$. To analyze the relationship between the full-batch loss minimization in Equation (1) and the mini-batch loss minimization in Equation (4), we first compare the objective functions of two problems as below.

**Proposition 1.** *The mini-batch loss and full-batch loss are not identical, nor is one a simple scaling of the other by a constant factor. In other words, when $\mathcal{S}_B = \left[\binom{N}{B}\right]$, for all $B \geq 2$, there exists no constant $c$ such that $\mathcal{L}_{\text{mini}}^{\text{con}}(\boldsymbol{U}, \boldsymbol{V}; \mathcal{S}_B) = c \cdot \mathcal{L}^{\text{con}}(\boldsymbol{U}, \boldsymbol{V})$ for all $\boldsymbol{U}, \boldsymbol{V}$.*

We visualize this proposition in Figure 1(a). Interestingly, the following result shows that the optimal solutions of both problems are identical.

**Theorem 4** (Optimization with all possible $\binom{N}{B}$ mini-batches). *Suppose $B \geq 2$. The set of minimizers of the $\binom{N}{B}$ mini-batch problem in Equation (4) is the same as that of the full-batch problem in Equation (1)*

*for two cases: (i) $N \leq d + 1$, and (ii) $N = 2d$ and the pairs $(\boldsymbol{U}, \boldsymbol{V})$ are restricted to those satisfying the conditions stated in Def. 2. In such cases, the solutions $(\boldsymbol{U}, \boldsymbol{V})$ for the $\binom{N}{B}$ mini-batch optimization problem satisfies the following: Case (i) $\{\boldsymbol{u}_i\}_{i=1}^N$ forms a simplex ETF and $\boldsymbol{u}_i = \boldsymbol{v}_i$ for all $i \in [N]$; Case (ii): $\{\boldsymbol{u}_i\}_{i=1}^N$ forms a simplex cross-polytope.*

*Proof Outline.* Similar to the proof of Lemma 1, we bound the objective function from below using Jensen's inequality. We then show that this lower bound is equivalent to a scaling of the bound from the proof of Lemma 1 by employing non-trivial counting arguments. Furthermore, we demonstrate that both the simplex ETF and the simplex cross-polytope minimize this lower bound for each case while satisfying the conditions that make the applications of Jensen's inequality tight. See Appendix C.1 for the detailed proof.

Now, we present mathematical results specifying the cases when the solutions of mini-batch optimization and full-batch optimization *differ*. First, we show that when $B = 2$, minimizing the mini-batch loss over any strict subset of $\binom{N}{B}$ batches, is not equivalent to minimizing the full-batch loss.

**Theorem 5** (Optimization with fewer than $\binom{N}{B}$ mini-batches)**.** *Suppose $B = 2$ and $N \leq d + 1$. Then, the minimizer of Equation (4) for $\mathcal{S}_B \subsetneq \left[\binom{N}{B}\right]$ is not the minimizer of the full-batch optimization in Equation (1).*

*Proof Outline.* We show that there exist embedding vectors that are not the simplex ETF, and have a strictly lower objective value. This implies that the optimal solution of any set of mini-batches that does not contain all $\binom{N}{2}$ mini-batches is not the same as that of the full-batch problem. See Appendix C.1 for the detailed proof.

The result of Theorem 5 is extended to the general case of $B \geq 2$, under some mild assumption; please check Proposition 2 and 3 in Appendix C.1.

Figure 1 summarizes the main findings in this section. These findings suggest that mini-batch optimization can achieve the optimal embeddings of full-batch optimization, as long as all $\binom{N}{B}$ possible mini-batches are considered. However, this poses a computational challenge, as it involves loading all these mini-batches. In the following section, we introduce efficient algorithms for speeding up the $\binom{N}{B}$ mini-batch optimization.

## 5 Ordered SGD for Mini-Batch Contrastive Learning

Recall that the optimal embeddings for the full-batch optimization problem in Equation (1) can be obtained by minimizing the sum of $\binom{N}{B}$ mini-batch losses. An easy way of achieving the optimal embeddings is using gradient descent (GD) on the sum of losses for $\binom{N}{B}$ mini-batches, or using a stochastic approach that applies GD on the loss for a randomly chosen mini-batch. Recent works found that applying GD on selective batches outperforms SGD in some cases (Kawaguchi & Lu, 2020; Lu et al., 2021; Loshchilov & Hutter, 2015). A natural question is, whether this holds for mini-batch *contrastive* learning, *i.e.*, (i) Is SGD enough to guarantee good convergence on mini-batch contrastive learning?, and (ii) Can we come up with a batch selection method that performs better than vanilla SGD? To answer this question:

- We show that Ordered SGD (OSGD) (Kawaguchi & Lu, 2020) can potentially accelerate convergence compared to vanilla SGD in a demonstrative toy example (Section 5.1), and extend the convergence results from Kawaguchi & Lu (2020) to mini-batch contrastive loss optimization (Section 5.2).

- We reformulate the batch selection problem into a min-cut problem in graph theory (Cormen et al., 2022), by considering a graph with $N$ nodes where each node is each positive pair and each edge represents a proxy to the contrastive loss between two nodes. This allows us to devise an efficient batch selection algorithm by leveraging a spectral clustering algorithm (Ng et al., 2001) (Section 5.3).

### 5.1 Convergence Comparison in a Toy Example: OSGD vs. SGD

This section investigates the convergence of two gradient-descent-based methods, OSGD and SGD. The below lemma shows that the contrastive loss is geodesically non-quasi-convex, which implies the hardness of proving the convergence to the optimal embeddings of gradient-based methods for contrastive learning in Equation (1). The proof is provided in Appendix C.2.

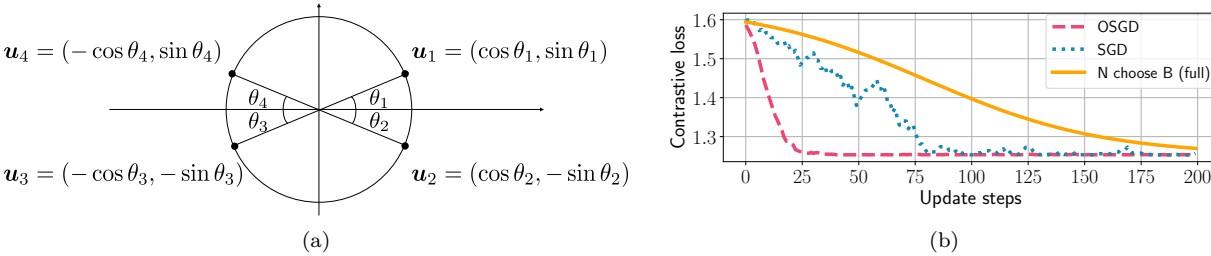

Figure 2: (a) Toy example considered in Section 5.1; (b) The training loss curves of three algorithms (OSGD, SGD, and $\binom{N}{B}$ full-batch gradient descent) applied on the toy example when $N = 4$ and $B = 2$. The x-axis represents the number of update steps, while the y-axis displays the loss in Equation (2). OSGD converges the fastest among the three methods.

**Lemma 2.** *Contrastive loss $\mathcal{L}^{\mathrm{con}}(\boldsymbol{U}, \boldsymbol{V})$ is a geodesically non-quasi-convex function of $\boldsymbol{U}, \boldsymbol{V}$ on $\mathcal{T} = \{(\boldsymbol{U}, \boldsymbol{V}) : \|\boldsymbol{u}_i\| = \|\boldsymbol{v}_i\| = 1, \forall i \in [N]\}$.*

In order to compare the convergence of OSGD and SGD to the optimal embeddings, we focus on a toy example where convergence to the optimal solution is achievable with appropriate initialization. Consider a scenario where we have $N = 4$ embedding vectors $\{\boldsymbol{u}_i\}_{i=1}^N$ with $\boldsymbol{u}_i \in \mathbb{R}^2$. Each embedding vector is defined as $\boldsymbol{u}_1 = (\cos\theta_1, \sin\theta_1); \boldsymbol{u}_2 = (\cos\theta_2, -\sin\theta_2); \boldsymbol{u}_3 = (-\cos\theta_3, -\sin\theta_3); \boldsymbol{u}_4 = (-\cos\theta_4, \sin\theta_4)$ for parameters $\{\theta_i\}_{i=1}^N$. For all $i$, the initial parameters are set as $\theta_i^{(0)} = \epsilon > 0$, and the other embedding vectors are initialized as $\boldsymbol{v}_i^{(0)} = \boldsymbol{u}_i^{(0)}$. This setting is illustrated in Figure 2(a). Over time step $t$, we consider updating the parameters $\boldsymbol{\theta}^{(t)} := [\theta_1^{(t)}, \theta_2^{(t)}, \theta_3^{(t)}, \theta_4^{(t)}]$ using the gradient descent based method on $\mathcal{L}^{\mathrm{con}}(\boldsymbol{U}_\mathcal{B}, \boldsymbol{V}_\mathcal{B})$ with a learning rate $\eta$: $\boldsymbol{\theta}^{(t+1)} = \boldsymbol{\theta}^{(t)} - \eta\nabla_{\boldsymbol{\theta}}\mathcal{L}^{\mathrm{con}}(\boldsymbol{U}_{\mathcal{B}^{(t)}}, \boldsymbol{V}_{\mathcal{B}^{(t)}})$. At each time step $t$, each learning algorithm begins by selecting a mini-batch $\mathcal{B}^{(t)} \subset \{1, 2, 3, 4\}$ with batch size $|\mathcal{B}^{(t)}| = 2$. SGD *randomly* selects a mini-batch, while OSGD selects a mini-batch as follows: $\mathcal{B}^{(t)} = \arg\max_{\mathcal{B} \in \mathcal{S}} \mathcal{L}^{\mathrm{con}}(\boldsymbol{U}_\mathcal{B}(\boldsymbol{\theta}^{(t)}), \boldsymbol{V}_\mathcal{B}(\boldsymbol{\theta}^{(t)}))$. For a sufficiently small margin $\rho > 0$, let $T_{\mathrm{OSGD}}, T_{\mathrm{SGD}}$ be the minimal time required for the algorithms to reach the condition that each component of the expected value $\mathbb{E}[\boldsymbol{\theta}^{(T)}]$ falls within $(\pi/4 - \rho, \pi/4)$:

$$T_{OSGD} := \min\{T \geq 0 : \mathbb{E}[\boldsymbol{\theta}_{OSGD}^{(T)}] \in (\pi/4 - \rho, \pi/4)^N\};$$
$$T_{SGD} := \min\{T \geq 0 : \mathbb{E}[\boldsymbol{\theta}_{SGD}^{(T)}] \in (\pi/4 - \rho, \pi/4)^N\},$$

where $\boldsymbol{\theta}_{OSGD}^{(t)}$ and $\boldsymbol{\theta}_{SGD}^{(t)}$ denote the model parameters at step $t$ when OSGD and SGD are used, respectively. Under this setting, the following theorem compares OSGD and SGD, in terms of the lower bound on the time required for the convergence to the optimal solution.

**Theorem 6.** *Consider the described setting where the parameters $\boldsymbol{\theta}^{(t)}$ of embedding vectors are updated, as shown in Figure 2(b). Suppose there exist $\tilde{\epsilon}, \overline{T}$ such that for all $t$ satisfying $\mathcal{B}^{(t)} = \{1, 3\}$ or $\{2, 4\}$, $\|\nabla_{\boldsymbol{\theta}^{(t)}}\mathcal{L}^{\mathrm{con}}(\boldsymbol{U}_{\mathcal{B}^{(t)}}, \boldsymbol{V}_{\mathcal{B}^{(t)}})\| \leq \tilde{\epsilon}$, and $T_{\mathrm{OSGD}}, T_{\mathrm{SGD}} < \overline{T}$. Then, we have the following inequalities:*

$$T_{\mathrm{OSGD}} \geq \tau \cdot \frac{\pi/4 - \rho - \epsilon + O(\eta^2\epsilon + \eta\epsilon^3)}{\eta\epsilon}, \quad T_{\mathrm{SGD}} \geq \frac{3(e^{2/\tau} + 1)}{e^{2/\tau} - 1}\tau \cdot \frac{\pi/4 - \rho - \epsilon + O(\eta^2\epsilon + \eta^2\tilde{\epsilon})}{\eta\epsilon + O(\eta\epsilon^3 + \eta\tilde{\epsilon})}.$$

**Corollary 2.** *Suppose lower bounds of $T_{\mathrm{OSGD}}, T_{\mathrm{SGD}}$ in Theorem 6 are tight, and the learning rate $\eta$ is small enough. Then, $T_{\mathrm{OSGD}}/T_{\mathrm{SGD}} = (e^{2/\tau} - 1)/3(e^{2/\tau} + 1) < 1$.*

In Figure 2(b), we present training loss curves of the full-batch contrastive loss in Equation (2) for various algorithms implemented on the toy example. One can observe that the loss of all algorithms eventually converges to 1.253, the optimal loss achievable when the solution satisfies $\boldsymbol{u}_i = \boldsymbol{v}_i$ and $\{\boldsymbol{u}_i\}_{i=1}^N$ form simplex cross-polytope. Note that the optimality of this solution is given in Theorem 4. This empirical evidence corroborates our theoretical findings, indicating that OSGD converges faster than SGD.

Figure 3: Histograms of batch counts for $N/B$ batches, for the contrastive loss measured from ResNet-18 models trained on CIFAR-100 using SGD, where $N$=50,000 and $B$=20. Each plot is derived from a distinct training epoch. Here we compare two batch selection methods: (i) random shuffling $N$ samples and partition them into $N/B$ batches of size $B$, (ii) our SC method given in Algorithm 1. The histograms show that batches generated through the proposed spectral clustering method tend to contain a higher proportion of large loss values when compared to random batch selection. Similar results are observed in different settings, details of which are given in Appendix E.1.

## 5.2 Convergence of OSGD in Mini-batch Contrastive Learning Setting

Recall that it is challenging to prove the convergence of gradient-descent-based methods to the optimal embeddings for contrastive learning problems in Equation (1) due to the non-quasi-convexity of the contrastive loss $\mathcal{L}^{\text{con}}$. Instead, we show that OSGD minimizes a modification of the empirical contrastive loss $\tilde{\mathcal{L}}^{\text{con}}$ that considers the order of the batch losses, and is guaranteed to converge at a sublinear rate to a critical point the loss by extending the results in Kawaguchi & Lu (2020). We consider a proxy, the *weighted* contrastive loss defined as $\widetilde{\mathcal{L}}^{\text{con}}(\boldsymbol{U}, \boldsymbol{V}) \coloneqq \frac{1}{q} \sum_{j=1}^{\binom{N}{B}} \gamma_j \mathcal{L}^{\text{con}}(\boldsymbol{U}_{\mathcal{B}_{(j)}}, \boldsymbol{V}_{\mathcal{B}_{(j)}})$ with $\gamma_j = \sum_{l=0}^{q-1} \binom{j-1}{l} \binom{\binom{N}{B}-j}{k-l-1} / \binom{\binom{N}{B}}{k}$ for two arbitrary natural numbers $k, q \leq \binom{N}{B}$ where $\mathcal{B}_{(j)}$ is a mini-batch with $j$-th largest loss among batches of size $B$. Indeed, this is a natural objective obtained by applying OSGD to our problem. OSGD updates the embeddings using the gradient averaged over $q$ batches that have *the largest losses* among randomly chosen $k$ batches (see Algorithm 2 in Appendix C.2). Let $\boldsymbol{U}^{(t)}, \boldsymbol{V}^{(t)}$ be the updated embedding matrices when applying OSGD for $t$ steps starting from $\boldsymbol{U}^{(0)}, \boldsymbol{V}^{(0)}$, using the learning rate $\eta_t$. Then the following theorem, proven in Appendix C.2, holds.

**Theorem 7** (Convergence results). *Consider sampling $t^\star$ from $[T]$ with probability proportional to $\{\eta_t\}_{t=0}^{T}$, that is, $\mathbb{P}(t^\star = t) = \eta_t / (\sum_{i=0}^{T} \eta_i)$. Then $\forall \rho > \rho_0 = \sqrt{8/B\tau^2} + 4e^{2/\tau}/B\tau^2$,*

$$\mathbb{E}\left[\left\|\nabla\widetilde{\mathcal{L}}^{\text{con}}(\boldsymbol{U}^{(t^\star)}, \boldsymbol{V}^{(t^\star)})\right\|^2\right] \leq \frac{(\rho + \rho_0)^2}{\rho(\rho - \rho_0)} \frac{\left(\widetilde{\mathcal{L}}^{\text{con}}(\boldsymbol{U}^{(0)}, \boldsymbol{V}^{(0)}) - \widetilde{\mathcal{L}}^{\text{con}\star}\right) + 8\rho \sum_{t=0}^{T} \eta_t^2}{\sum_{t=0}^{T} \eta_t},$$

*where $\widetilde{\mathcal{L}}^{\text{con}\star}$ denotes the minimized value of $\widetilde{\mathcal{L}}^{\text{con}}$.*

Given sufficiently small learning rate $\eta_t \sim O(t^{-1/2})$, $\mathbb{E}\|\nabla\widetilde{\mathcal{L}}^{\text{con}}\|^2$ decays at the rate of $\widetilde{O}(T^{-1/2})$.

## 5.3 Spectral Clustering-based Approach

Applying OSGD to mini-batch contrastive learning has a potential benefit as shown in Section 5.1, but it also has some challenges. Choosing the best $q$ batches with high loss in OSGD is only doable after we evaluate losses of all $\binom{N}{B}$ combinations. This process requires the computational complexity of $O(N^B \log N^B)$, which is computationally infeasible for large $N$. A naive solution to tackle this challenge is to first randomly choose $k$ batches and then select $q$ high-loss batches among $k$ batches. However, this naive random batch selection method does not guarantee that the chosen $q$ batches have the highest loss among all $\binom{N}{B}$ candidates.

Motivated by these issues of OSGD, we relax this problem by reducing our search space such that the $q = N/B$ chosen batches $\mathcal{B}_1, \cdots, \mathcal{B}_q$ form a partition of $N$ samples, *i.e.*, $\mathcal{B}_i \cap \mathcal{B}_j = \varnothing$ and $\cup_{i \in [q]} \mathcal{B}_i = [N]$. We suggest an alternative batch selection method inspired by graph theory. The mini-batch contrastive loss

---

**Algorithm 1:** Spectral Clustering Method

---

**Input:** the number of positive pairs $N$, mini-batch size $B$, embedding matrices: $\boldsymbol{U}$, $\boldsymbol{V}$
**Output:** selected mini-batches $\{\mathcal{B}_j\}_{j=1}^{N/B}$

**1** Construct the affinity matrix $A$: $\qquad A_{ij} = \begin{cases} w(i,j) \text{ in Equation (5)} & \text{if } i \neq j \\ 0 & \text{else} \end{cases}$

**2** Construct the degree matrix $D$ from $A$: $D_{ij} = \begin{cases} 0 & \text{if } i \neq j \\ \sum_{j=1}^{N} A_{ij} & \text{else} \end{cases}$

**3** $L \leftarrow D - A;\ k \leftarrow N/B$

**4** Find the $k$ largest eigenvectors of $L$, denoted as $V_k \in \mathbb{R}^{N \times k}$

**5** Normalize the rows of $V_k$ to have unit $\ell_2$-norm

**6** Apply the $K$-means clustering algorithm on the rows of the normalized $V_k$ to get cluster centers $Z \in \mathbb{R}^{k \times k}$

**7** Construct a bipartite graph $\mathcal{G}_{\mathsf{assign}}$: (i) the first partite set is $V_k$ and (ii) the second set is the collection of $B$ copies of each center in $Z$

**8** Compute distances between row vectors of $V_k$ and $B$ copies of each center in $Z$, and assign these as edge weights in $\mathcal{G}_{\mathsf{assign}}$

**9** Solve the minimum weight matching problem in $\mathcal{G}_{\mathsf{assign}}$ using a method such as the Hungarian algorithm

**10** **return** $\{\mathcal{B}_j\}_{j=1}^{N/B}$

---

$\mathcal{L}^{\mathsf{con}}(U_{\mathcal{B}}, V_{\mathcal{B}})$ for a given batch $\mathcal{B}$ is lower bounded using Jensen's inequality as follows:

$$
\begin{aligned}
\mathcal{L}^{\mathsf{con}}(\boldsymbol{U}_{\mathcal{B}}, \boldsymbol{V}_{\mathcal{B}}) &= -\frac{1}{B}\sum_{i\in\mathcal{B}}\log\left(\frac{e^{\boldsymbol{u}_i^{\mathsf{T}}\boldsymbol{v}_i/\tau}}{\sum_{j=1}^{N}e^{\boldsymbol{u}_i^{\mathsf{T}}\boldsymbol{v}_j/\tau}}\right) - \frac{1}{B}\sum_{i=\in\mathcal{B}}\log\left(\frac{e^{\boldsymbol{v}_i^{\mathsf{T}}\boldsymbol{u}_i/\tau}}{\sum_{j=1}^{N}e^{\boldsymbol{v}_i^{\mathsf{T}}\boldsymbol{u}_j/\tau}}\right) \\
&= \frac{1}{B}\left\{\sum_{i\in\mathcal{B}}\log\left(1+\sum_{j\in\mathcal{B}\setminus\{i\}}e^{\boldsymbol{u}_i^{\mathsf{T}}(\boldsymbol{v}_j-\boldsymbol{v}_i)/\tau}\right) + \sum_{i\in\mathcal{B}}\log\left(1+\sum_{j\in\mathcal{B}\setminus\{i\}}e^{\boldsymbol{v}_i^{\mathsf{T}}(\boldsymbol{u}_j-\boldsymbol{u}_i)/\tau}\right)\right\} \\
&\geq \frac{1}{B(B-1)}\left\{\sum_{i\in\mathcal{B}}\sum_{j\in\mathcal{B}\setminus\{i\}}\log\left(1+(B-1)e^{\boldsymbol{u}_i^{\mathsf{T}}(\boldsymbol{v}_j-\boldsymbol{v}_i)/\tau}\right) + \log\left(1+(B-1)e^{\boldsymbol{v}_i^{\mathsf{T}}(\boldsymbol{u}_j-\boldsymbol{u}_i)/\tau}\right)\right\}.
\end{aligned}
$$

We employ this lower bound due to a nice property that can be expressed as a summation of losses over a pair $(i, j)$ of samples within batch $\mathcal{B}$. We consider a graph $\mathcal{G}$ with $N$ nodes and the weight $w(k, l)$ between node $k$ and $l$ is defined as

$$
w(k,l) := \sum_{(i,j)\in\{(k,l),(l,k)\}}\log(1+(B-1)e^{\boldsymbol{u}_i^{\mathsf{T}}(\boldsymbol{v}_j-\boldsymbol{v}_i)/\tau}) + \log(1+(B-1)e^{\boldsymbol{v}_i^{\mathsf{T}}(\boldsymbol{u}_j-\boldsymbol{u}_i)/\tau}). \tag{5}
$$

Recall that our goal is to choose $q$ batches having the highest contrastive losses among $\binom{N}{B}$ batches. In such a scenario, our target problem is equivalent to the problem of clustering $N$ nodes in graph $\mathcal{G}$ into $q$ clusters with equal size, where the objective is to minimize the sum of weights of inter-cluster edges. This problem is the min-cut problem (Cormen et al., 2022). To approximate the solution to this problem based on the graph $\mathcal{G}$, we propose a variant of the well-known spectral clustering algorithm (Ng et al., 2001). Since the spectral clustering algorithm cannot guarantee to assign an equal number of nodes to each cluster, we introduce an additional step to ensure that each cluster (batch) has the equal number $B$ of positive pairs. This step addresses a minimum weight matching problem in a bipartite graph (Crouse, 2016) involving $N$ data points and replicated cluster centers from spectral clustering. One partite set is the collection of data points and the other set consists of $B$ copies of each cluster center. The aim is to allocate exactly $B$ data points to each center, minimizing overall cost, defined as the total distance from data points to their respective centers. This ensures each cluster receives an equal number of data points while minimizing the assignment cost. The pseudo-code of our batch selection method[2] is provided in Algorithm 1. The computational complexity of our *spectral clustering* (SC) method is $O(k^2 B^2 N)$, while the complexity of OSGD is $O(N^B \log N^B)$ due to the necessity of sorting $\binom{N}{B}$ mini-batches. We provide a detailed complexity comparison of the algorithms in Appendix F. Figure 3 shows that the proposed SC method indeed favors batches with larger loss values.

---

[2]Our algorithm finds $N/B$ clusters *at once*, instead of only finding a single best cluster. Compared with such an alternative approach, our method is (i) more efficient when we update models for multiple iterations, and (ii) guaranteed to load all samples with $N/B$ batches, thus expected to have better convergence (Bottou, 2009; Haochen & Sra, 2019; Gürbüzbalaban et al., 2021).

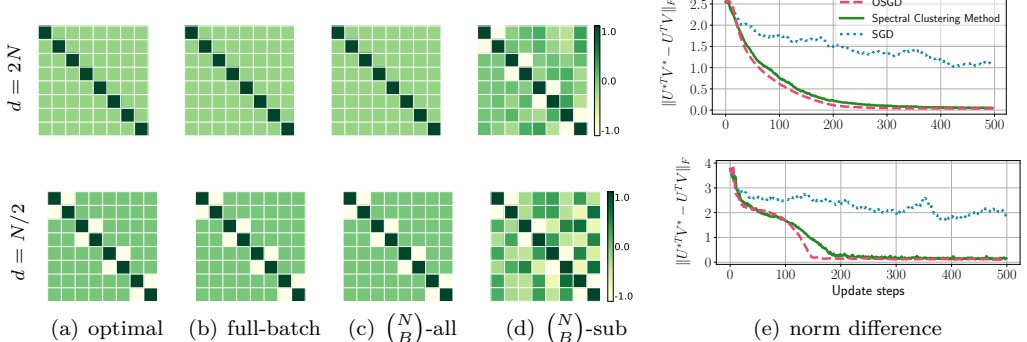

(a) optimal    (b) full-batch    (c) $\binom{N}{B}$-all    (d) $\binom{N}{B}$-sub      (e) norm difference

Figure 4: The behavior of embedding matrices $\boldsymbol{U}, \boldsymbol{V}$ optimized by different batch selection methods for $N = 8$ and $B = 2$. (a)-(d): Heatmap of $N \times N$ matrix visualizing the pairwise inner products $\boldsymbol{u}_i^\intercal \boldsymbol{v}_j$, where (a): ground-truth solution (ETF for $d = 2N$, cross-polytope for $d = N/2$), (b): optimized the full-batch loss with GD, (c): optimized the sum of $\binom{N}{B}$ mini-batch losses with GD, (d): optimized a partial sum of $\binom{N}{B}$ mini-batch losses with GD. Note that both (b) and (c) reaches the ground-truth solution in (a), while (d) does not, supporting our theoretical results in Section 4.2. Further, (e) compares the convergence of three mini-batch selection algorithms: 1) SGD, 2) OSGD, and 3) our spectral clustering method, when updating embeddings for 500 steps. OSGD and our method nearly converge to the optimal solution, while SGD does not. Here, $y$-axis represents the Frobenius norm of the difference between the heatmaps of the optimal solution and the updated embeddings, denoted by $\|\boldsymbol{U}^{\star\intercal}\boldsymbol{V}^{\star} - \boldsymbol{U}^\intercal\boldsymbol{V}\|_F$.

# 6 Experiments

We validate our theoretical findings and the effectiveness of the proposed method by providing experimental results on synthetic and real datasets. We first show that experimental results on synthetic dataset coincides with two main theoretical results: (i) the relationship between the full-batch contrastive loss and the mini-batch contrastive loss given in Section 4, (ii) the analysis on the convergence of OSGD and the proposed SC method given in Section 5. To demonstrate the practicality of our method, we provide experimental results on CIFAR-100 (Krizhevsky et al., 2009) and Tiny ImageNet (Le & Yang, 2015). Details of the experimental setting can be found in Appendix E, and our code is available at https://github.com/krafton-ai/mini-batch-cl.

## 6.1 Synthetic Dataset

Consider optimizing the embedding matrices $\boldsymbol{U}, \boldsymbol{V}$ using GD, where each column of $\boldsymbol{U}, \boldsymbol{V}$ is initialized as a multivariate normal vector and then normalized as $\|\boldsymbol{u}_i\| = \|\boldsymbol{v}_i\| = 1, \forall i$. We use learning rate $\eta = 0.5$, and apply the normalization step at every iteration.

First, we compare the minimizers of three optimization problems: (i) full-batch optimization in Equation (1); (ii) mini-batch optimization in Equation (4) with $\mathcal{S}_B = \left[\binom{N}{B}\right]$; (iii) mini-batch optimization with $\mathcal{S}_B \subsetneq \left[\binom{N}{B}\right]$. We apply GD to each problem for $N = 8$ and $B = 2$, obtain the updated embedding matrices, and then show the heatmap plot of $N \times N$ gram matrix containing all the pairwise inner products $\boldsymbol{u}_i^\intercal \boldsymbol{v}_j$ in Figure 4(b)-(d). One can observe that when either full-batch or all $\binom{N}{B}$ mini-batches are used for training, the trained embedding vectors reach a simplex ETF and simplex cross-polytope solutions for $d = 2N$ and $d = N/2$, respectively, as proved in Theorem 4. In contrast, when a strict subset of $\binom{N}{B}$ mini-batches is used for training, these solutions are not achieved.

Second, we compare the convergence speed of three algorithms: (i) the proposed SC method; (ii) OSGD which updates the model using the top-1 batch that exhibits the highest loss from among the $\binom{N}{B}$ possible batches at each step (see Algorithm 5); and (iii) SGD (see Algorithm 4). Figure 4(e) shows that both OSGD and the proposed method nearly converge to the ground-truth solutions proved in Theorem 4 within 500 steps, while SGD does not. We obtain similar results for other values of $N$ and $d$, given in Appendix E.2.

Table 1: Top-1 retrieval accuracy on CIFAR-100 (or Tiny ImageNet), when each algorithm employs one of the following datasets to pretrain ResNet-18 with SimCLR and SogCLR objective: (1) CIFAR-100, (2) Tiny ImageNet, (3) CIFAR-100-C, or (4) Tiny ImageNet-C. SC algorithm proposed in Section 5.3 outperforms existing baselines.

| | Image Retrieval | | | | | | | |
| | CIFAR-100 | | Tiny ImageNet | | CIFAR-100-C | | Tiny ImageNet-C | |
| | SimCLR | SogCLR | SimCLR | SogCLR | SimCLR | SogCLR | SimCLR | SogCLR |
|---|---|---|---|---|---|---|---|---|
| SGD | 70.29 | 60.66 | 78.39 | 67.48 | 38.48 | 38.34 | 42.42 | 45.04 |
| OSGD (approximated) | 70.00 | 61.48 | 78.57 | 69.31 | 38.55 | 38.71 | 43.11 | 45.63 |
| SC | **74.71** | **73.32** | **82.95** | **79.73** | **38.99** | **41.51** | **43.45** | **46.81** |

## 6.2 Real Datasets

In this section, we show that the proposed SC method is effective in more practical settings where the embedding is learned by a parameterized encoder, and can be easily applied to existing uni-modal frameworks, such as SimCLR (Chen et al., 2020a) and SogCLR (Yuan et al., 2022). We integrate different batch selection methods from (i) the proposed SC method; (ii) OSGD *without replacement* (see Algorithm 6), an approximated OSGD, selected due to the intractability of evaluating all $\binom{N}{B}$ batches on real datasets; and (iii) SGD (see Algorithm 4) into these frameworks. We compare the pre-trained models' performances in the retrieval downstream tasks.

We conduct mini-batch contrastive learning with the mini-batch size $B = 32$ using ResNet18-based encoders on CIFAR-100 and Tiny ImageNet datasets. All learning is executed on a single NVIDIA A100 GPU. The training code and hyperparameters are based on the official codebase of SogCLR[3] (Yuan et al., 2022). We use LARS optimizer (You et al., 2017) with the momentum of 0.9 and the weight decay of $10^{-6}$. We utilize the learning rate scheduler which starts with a warm-up phase in the initial 10 epochs, during which the learning rate increases linearly to the maximum value $\eta_{\max} = 0.075\sqrt{B}$. After this warm-up stage, we employ a cosine annealing (half-cycle) schedule for the remaining epochs. For the approximated OSGD, we employ $k = 1500$, $q = 150$. To expedite batch selection in the proposed SC, we begin by randomly partitioning $N$ positive pairs into $kB$-sized clusters, using $k = 40$. We then apply the SC method to each $kB$ cluster to generate $k$ mini-batches, resulting in a total of $k \times (N/kB) = N/B$ mini-batches. We train models for a total of 100 epochs. We measure the performances of the models under the retrieval task which is defined as finding the positive pair image of a given image among all pairs (the number of images of the validation dataset): (1) We extract normalized embedding features from augmented images by employing the pre-trained models; (2) We identify the pair image of the given augmented image among all of the validation images with the cosine similarity of embedding vectors.

We also measure performances of the model pre-trained on CIFAR-10 (or Tiny ImageNet) under a harder setting, where the various corruptions are applied per image so that we can consider a set of corrupted images as hard negative samples. We employ CIFAR-100-C (or Tiny ImageNet-C) which is the corrupted dataset (Hendrycks & Dietterich, 2019) designed for robustness evaluation. CIFAR-100-C (Tiny ImageNet-C) has the same images as CIFAR-100 (Tiny ImageNet), but these images have been altered by 19 (15) different types of corruption (e.g., image noise, blur, etc.). Each type of corruption has five severity levels. We utilize images corrupted at severity level 1. These images tend to be more similar to each other than those corrupted at higher severity levels, which consequently makes it more challenging to retrieve positive pairs among other images. To perform the retrieval task, we follow the following procedures: (i) We apply two distinct augmentations to each image to generate *positive* pairs; (ii) We extract normalized embedding features from augmented images by employing the pre-trained models; (iii) we identify the pair image of the given augmented image among augmentations of 19 (15) corrupted images with the cosine similarity of embedding vectors. This process is iterated across 10K CIFAR-100 images (10K Tiny-ImageNet images). The top-1 accuracy measures a percentage of retrieved images that match its positive pair image, where each pair contains two different modalities stemming from a single image. Table 1 presents the top-1 retrieval accuracy on CIFAR-100, Tiny ImageNet, and their corrupted counterparts. The proposed SC method

---

[3]https://github.com/Optimization-AI/SogCLR

surpasses SGD and the approximated OSGD. We show the efficacy of the SC method, yet it has a limitation in that the proposed method requires more time for batch selection compared to SGD. To mitigate this issue in practice, we suggest parallel batch selection using the SC method while updating models through gradient descent. This approach involves using the same batches for multiple epochs as the next batch selection proceeds, based on the assumption that batches with higher loss values do not rapidly change. Exploring this approach is beyond the scope of this work so we leave it as a future work.

## 7  Conclusion

We theoretically analyzed mini-batch contrastive learning with the InfoNCE loss. First, we showed that the solution of mini-batch optimization and that of full-batch optimization are identical if and only if all $\binom{N}{B}$ mini-batches are considered. Second, we analyzed the convergence of OSGD and devised spectral clustering (SC) method, a new batch selection method which handles the complexity issue of OSGD in mini-batch contrastive learning. Experimental results support our theoretical findings and the superiority of SC. Our future work includes (1) characterizing the optimal embeddings for $N > d + 1$ case, (2) generalizing our findings to broader contrastive losses, and (3) exploring a complexity-efficient batch selection method.

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

## Organization of the Appendix

1. Appendix A outlines the limitations of this paper.

2. In Appendix B we introduce some additional definitions for posterity.

3. In Appendix C we provide detailed proofs of the results from the main paper as well as any intermediate results/lemmas that we found useful.

   (a) Specifically, in Appendix C.1 we provide proofs of the results from Section 4 which focuses on the relationship between the optimal solutions for minimizing the mini-batch and full-batch constrastive loss.

   (b) Appendix 5 contains the proofs of results from Section C.2 which concern the application of Ordered SGD to mini-batch contrastive learning.

   (c) Appendix C.3 is intended to supplement Appendix 5 in that it contains auxiliary notation and proofs required in the proof of Theorem 7.

4. Appendix D specifies the pseudo-code and details for the two algorithms: (i) Stochastic Gradient Descent and (ii) Ordered SGD.

5. Appendix E provide some additional results.

6. In Appendix F we provide computational complexities for the three algorithms.

## A Limitations

We note that our theoretical results have two major limitations:

1. While we would like to extend our results to the general case of $N > d + 1$, we were only able to characterize the optimal solution for the specific case of $N = 2d$. Furthermore, our result for the case of $N = 2d$ in Theorem 4 requires the use of the conjecture that the optimal solution is symmetric and antipodal. However, as mentioned by Lu & Steinerberger (2022), the general case of $N > d + 1$ seems quite challenging in the non-asymptotic regime.

2. In practice, the embeddings are usually the output of a shared neural network encoder. However, our results are for the case when the embeddings only have a norm constraint. Thus, our results do not readily indicate any generalization to unseen data. We expect however, that it is possible to extend our results to the shared encoder setting by assuming sufficient overparameterization.

## B Additional Definitions

**Definition 4** (Sustik et al. (2007)). A set of $N$ vectors $\{\boldsymbol{u}_i\}_{i=1}^N$ in the $\mathbb{R}^d$ form an equiangular tight frame (ETF) if (i) they are all unit norm: $\|\boldsymbol{u}_i\| = 1$ for every $i \in [N]$, (ii) they are equiangular: $\|\boldsymbol{u}_i^\mathsf{T} \boldsymbol{u}_j\| = \alpha \geq 0$ for all $i \neq j$ and some $\alpha \geq 0$, and (iii) they form a tight frame: $\boldsymbol{U}\boldsymbol{U}^\mathsf{T} = (N/d)\mathbb{I}_d$ where $\boldsymbol{U}$ is a $d \times N$ matrix whose columns are $\boldsymbol{u}_1, \boldsymbol{u}_2, \ldots, \boldsymbol{u}_N$, and $\mathbb{I}_d$ is the $d \times d$ identity matrix.

## C Proofs

### C.1 Proofs of results from Section 4

**Lemma 1** (Optimal solution when $N \leq d + 1$). *Suppose $N \leq d + 1$. Then, the optimal solution $(\boldsymbol{U}^\star, \boldsymbol{V}^\star)$ of the full-batch contrastive learning problem in Equation (1) satisfies two properties: (i) $\boldsymbol{U}^\star = \boldsymbol{V}^\star$, and (ii) the columns of $\boldsymbol{U}^\star$ form a simplex ETF.*

*Proof.* First, we define the contrastive loss as the sum of two symmetric one-sided contrastive loss terms to simplify the notation. We denote the following term as the one-sided contrastive loss

$$\mathcal{L}(\boldsymbol{U}, \boldsymbol{V}) = \frac{1}{N} \sum_{i=1}^{N} -\log \left( \frac{e^{\boldsymbol{u}_i^\intercal \boldsymbol{v}_i / \tau}}{\sum_{j=1}^{N} e^{\boldsymbol{u}_i^\intercal \boldsymbol{v}_j / \tau}} \right). \tag{6}$$

Then, the overall contrastive loss is given by the sum of the two one-sided contrastive losses:

$$\mathcal{L}^{\mathrm{con}}(\boldsymbol{U}, \boldsymbol{V}) = \mathcal{L}(\boldsymbol{U}, \boldsymbol{V}) + \mathcal{L}(\boldsymbol{V}, \boldsymbol{U}). \tag{7}$$

Since $\mathcal{L}^{\mathrm{con}}$ is symmetric in its arguments, results pertaining to the optimum of $\mathcal{L}(\boldsymbol{U}, \boldsymbol{V})$ readily extend to $\mathcal{L}^{\mathrm{con}}$. Now, let us consider the simpler problem of minimizing the one-sided contrastive loss from Equation (6) which reduces the problem to exactly the same setting as Lu & Steinerberger (2022):

$$\mathcal{L}(\boldsymbol{U}, \boldsymbol{V}) = \frac{1}{N} \sum_{i=1}^{N} -\log \left( \frac{e^{\boldsymbol{u}_i^\intercal \boldsymbol{v}_i / \tau}}{\sum_{j=1}^{N} e^{\boldsymbol{u}_i^\intercal \boldsymbol{v}_j / \tau}} \right)$$

$$= \frac{1}{N} \sum_{i=1}^{N} \log \left( 1 + \sum_{j=1, j \neq i}^{N} e^{(\boldsymbol{v}_j - \boldsymbol{v}_i)^\intercal \boldsymbol{u}_i / \tau} \right).$$

Note that, we have for any fixed $1 \leq i \leq N$,

$$\sum_{j=1, j \neq i}^{N} e^{(\boldsymbol{v}_j - \boldsymbol{v}_i)^\intercal \boldsymbol{u}_i / \tau} = e^{-(\boldsymbol{v}_i^\intercal \boldsymbol{u}_i / \tau)} \sum_{j=1, j \neq i}^{N} e^{\boldsymbol{v}_j^\intercal \boldsymbol{u}_i / \tau}$$

$$= (N-1) e^{-(\boldsymbol{v}_i^\intercal \boldsymbol{u}_i / \tau)} \left( \frac{1}{N-1} \right) \sum_{j=1, j \neq i}^{N} e^{\boldsymbol{v}_j^\intercal \boldsymbol{u}_i / \tau}$$

$$\overset{(a)}{\geq} (N-1) e^{-(\boldsymbol{v}_i^\intercal \boldsymbol{u}_i / \tau)} \exp \left( \frac{1}{N-1} \sum_{j=1, j \neq i}^{N} \boldsymbol{v}_j^\intercal \boldsymbol{u}_i / \tau \right)$$

$$\overset{(b)}{=} (N-1) e^{-(\boldsymbol{v}_i^\intercal \boldsymbol{u}_i / \tau)} \exp \left( \frac{\boldsymbol{v}^\intercal \boldsymbol{u}_i - \boldsymbol{v}_i^\intercal \boldsymbol{u}_i}{\tau (N-1)} \right)$$

$$= (N-1) \exp \left( \frac{\boldsymbol{v}^\intercal \boldsymbol{u}_i - N(\boldsymbol{v}_i^\intercal \boldsymbol{u}_i)}{\tau (N-1)} \right), \tag{8}$$

where $(a)$ follows by applying Jensen inequality for $e^t$ and $(b)$ follows from $\boldsymbol{v} := \sum_{i=1}^{N} \boldsymbol{v}_i$. Since $\log(\cdot)$ is monotonic, we have that $x > y \Rightarrow \log(x) > \log(y)$ and therefore,

$$\mathcal{L}(\boldsymbol{U}, \boldsymbol{V}) \geq \frac{1}{N} \sum_{i=1}^{N} \log \left[ 1 + (N-1) \exp \left( \frac{\boldsymbol{v}^\intercal \boldsymbol{u}_i}{\tau (N-1)} - \frac{N(\boldsymbol{v}_i^\intercal \boldsymbol{u}_i)}{\tau (N-1)} \right) \right]$$

$$\overset{(c)}{\geq} \log \left[ 1 + (N-1) \exp \left( \frac{1}{N} \sum_{i=1}^{N} \left( \frac{\boldsymbol{v}^\intercal \boldsymbol{u}_i}{\tau (N-1)} - \frac{N(\boldsymbol{v}_i^\intercal \boldsymbol{u}_i)}{\tau (N-1)} \right) \right) \right]$$

$$\overset{(d)}{=} \log \left[ 1 + (N-1) \exp \left( \frac{1}{N} \left( \frac{\boldsymbol{v}^\intercal \boldsymbol{u}}{\tau (N-1)} - \frac{N}{\tau (N-1)} \sum_{i=1}^{N} (\boldsymbol{v}_i^\intercal \boldsymbol{u}_i) \right) \right) \right] \tag{9}$$

where $(c)$ follows by applying Jensen inequality to the convex function $\phi(t) = \log(1 + ae^{bt})$ for $a, b > 0$, and $(d)$ follow from $\boldsymbol{u} := \sum_{i=1}^{N} \boldsymbol{u}_i$.

Note that for equalities to hold in Equation (8) and (9), we need constants $c_i, c$ such that

$$\boldsymbol{v}_j^\intercal \boldsymbol{u}_i = c_i \quad \forall j \neq i, \tag{10}$$

$$\frac{\boldsymbol{v}^\intercal \boldsymbol{u}_i}{N-1} - \frac{N(\boldsymbol{v}_i^\intercal \boldsymbol{u}_i)}{N-1} = c \quad \forall i \in [N]. \tag{11}$$

Since $\log(\cdot)$ and $\exp(\cdot)$ are both monotonic, minimizing the lower bound in Equation (8) is equivalent to

$$\min \quad \frac{\boldsymbol{v}^\mathsf{T}\boldsymbol{u}}{N-1} - \frac{N}{N-1}\sum_{i=1}^{N}\boldsymbol{v}_i^\mathsf{T}\boldsymbol{u}_i$$

$$\Leftrightarrow \max \quad N\sum_{i=1}^{N}\boldsymbol{v}_i^\mathsf{T}\boldsymbol{u}_i - \Big(\sum_{i=1}^{N}\boldsymbol{v}_i\Big)^\mathsf{T}\Big(\sum_{i=1}^{N}\boldsymbol{u}_i\Big). \tag{12}$$

All that remains is to show that the solution that maximizes Eq 12 also satisfies the conditions in Equation (10) and (11). To see this, first note that the maximization problem can be written as

$$\max \quad \boldsymbol{v}_{\text{stack}}^\mathsf{T}((N\mathbb{I}_N - \mathbf{1}_N\mathbf{1}_N^\mathsf{T}) \otimes \mathbb{I}_d)\boldsymbol{u}_{\text{stack}}$$

where $\boldsymbol{v}_{\text{stack}} = (\boldsymbol{v}_1, \boldsymbol{v}_2, \dots, \boldsymbol{v}_n)$ is a vector in $\mathbb{R}^{Nd}$ formed by stacking the vectors $\boldsymbol{v}_i$ together. $\boldsymbol{u}_{\text{stack}}$ is similarly defined. $\mathbb{I}_N$ denotes the $N \times N$ identity matrix, $\mathbf{1}_N$ denotes the all-one vector in $\mathbb{R}^n$, and $\otimes$ denotes the Kronecker product. It is easy to see that $\|\boldsymbol{u}_{\text{stack}}\| = \|\boldsymbol{v}_{\text{stack}}\| = \sqrt{N}$ since each $\|\boldsymbol{u}_i\| = \|\boldsymbol{v}_i\| = 1$. Since the eigenvalues of $A \otimes B$ are the product of the eigenvalues of $A$ and $B$, in order to analyze the spectrum of the middle term in the above maximization problem, it suffices to just consider the eigenvalues of $(N\mathbb{I}_N - \mathbf{1}_N\mathbf{1}_N^\mathsf{T})$. As shown by the elegant analysis in Lu & Steinerberger (2022), $(N\mathbb{I}_N - \mathbf{1}_N\mathbf{1}_N^\mathsf{T})\boldsymbol{p} = N\boldsymbol{p}$ for any $\boldsymbol{p} \in \mathbb{R}^N$ such that $\sum_{i=1}^{N}\boldsymbol{p}_i = 0$ and $(N\mathbb{I}_N - \mathbf{1}_N\mathbf{1}_N^\mathsf{T})\boldsymbol{q} = 0$ for any $\boldsymbol{q} \in \mathbb{R}^N$ such that $\boldsymbol{q} = k\mathbf{1}_N$ for some $k \in \mathbb{R}$. Therefore it follows that its eigenvalues are $N$ with multiplicity $(N-1)$ and $0$. Since its largest eigenvalue is $N$ and since $\|\boldsymbol{u}_{\text{stack}}\| = \|\boldsymbol{v}_{\text{stack}}\| = \sqrt{N}$, applying cauchy schwarz inequality, we have that

$$\max \quad \boldsymbol{v}_{\text{stack}}^\mathsf{T}(N\mathbb{I}_N - \mathbf{1}_N\mathbf{1}_N^\mathsf{T}) \otimes \mathbb{I}_d\boldsymbol{u}_{\text{stack}}^\mathsf{T}$$
$$= \|\boldsymbol{v}_{\text{stack}}\| \cdot \|(N\mathbb{I}_n - \mathbf{1}_n\mathbf{1}_n^\mathsf{T}) \otimes \mathbb{I}_d)\| \cdot \|\boldsymbol{u}_{\text{stack}}\|$$
$$= \sqrt{N}(N)\sqrt{N}$$
$$= N^2.$$

Moreover, we see that setting $\boldsymbol{u}_i = \boldsymbol{v}_i$ and setting $\{\boldsymbol{u}_i\}_{i=1}^{N}$ to be the simplex ETF attains the maximum above while also satisfying the conditions in Equation (10) and (11) with $c_i = -1/(N-1)$ and $c = -N/(N-1)$. Therefore, the inequalities in Equation (8) and (9) are actually equalities for $\boldsymbol{u}_i = \boldsymbol{v}_i$ when they are chosen to be the simplex ETF in $\mathbb{R}^d$ which is attainable since $d \geq N-1$. Therefore, we have shown that if $\boldsymbol{U}^\star = \{\boldsymbol{u}_i^\star\}_i = 1^N$ is the simplex ETF and $\boldsymbol{u}_i^\star = \boldsymbol{v}_i^\star \ \forall i \in [N]$, then $\boldsymbol{U}^\star, \boldsymbol{V}^\star = \arg\min_{\boldsymbol{U},\boldsymbol{V}} \mathcal{L}(\boldsymbol{U}, \boldsymbol{V})$ over the unit sphere in $\mathbb{R}^n$. All that remains is to show that this is also the minimizer for $\mathcal{L}^{\text{con}}$.

First note that $\boldsymbol{U}^\star, \boldsymbol{V}^\star$ is also the minimizer for $\mathcal{L}(\boldsymbol{V}, \boldsymbol{U})$ through symmetry. One can repeat the proof exactly by simply exchanging $\boldsymbol{u}_i$ and $\boldsymbol{v}_i$ to see that this is indeed true. Now recalling Equation (7), we have

$$\min \mathcal{L}^{\text{con}} = \min \left(\mathcal{L}(\boldsymbol{U}, \boldsymbol{V}) + \mathcal{L}(\boldsymbol{U}, \boldsymbol{V})\right)$$
$$\geq \min \left(\mathcal{L}(\boldsymbol{U}, \boldsymbol{V})\right) + \min \left(\mathcal{L}(\boldsymbol{U}, \boldsymbol{V})\right) \tag{13}$$
$$= \mathcal{L}(\boldsymbol{U}^\star, \boldsymbol{V}^\star) + \mathcal{L}(\boldsymbol{V}^\star, \boldsymbol{U}^\star).$$

However, since the minimizer of both terms in Equation (13) is the same, the inequality becomes an equality. Therefore, we have shown that $(\boldsymbol{U}^\star, \boldsymbol{V}^\star)$ is the minimizer of $\mathcal{L}^{\text{con}}$ completing the proof. $\qquad \square$

**Remark 1.** In the proof of the above Lemma, we only show that the simplex ETF attains the minimum loss in Equation (1), but not that it is the only minimizer. The proof of Lu & Steinerberger (2022) can be extended to show that this is indeed true as well. We omit it here for ease of exposition.

**Theorem 3** (Optimal solution when $N = 2d$). *Let*

$$(\boldsymbol{U}^\star, \boldsymbol{V}^\star) := \arg\min_{(\boldsymbol{U},\boldsymbol{V}) \in \mathcal{A}} \mathcal{L}^{\text{con}}(\boldsymbol{U}, \boldsymbol{V}) \tag{3}$$

$$s.t. \quad \|\boldsymbol{u}_i\| = 1, \|\boldsymbol{v}_i\| = 1 \quad \forall i \in [N],$$

*where* $\mathcal{A} := \{(\boldsymbol{U}, \boldsymbol{V}) : \boldsymbol{U}, \boldsymbol{V} \text{ are symmetric and antipodal}\}$. *Then, the columns of* $\boldsymbol{U}^\star$ *form a simplex cross-polytope for* $N = 2d$.

*Proof.* By applying the logarithmic property that allows division to be represented as subtraction,

$$\mathcal{L}(\boldsymbol{U}, \boldsymbol{V}) = -\frac{1}{N} \sum_{i=1}^{N} \log \left( \frac{e^{\boldsymbol{u}_i^{\mathsf{T}} \boldsymbol{v}_i / \tau}}{\sum_{j=1}^{N} e^{\boldsymbol{u}_i^{\mathsf{T}} \boldsymbol{v}_j / \tau}} \right)$$

$$= -\frac{1}{N} \sum_{i=1}^{N} \left[ \boldsymbol{u}_i^{\mathsf{T}} \boldsymbol{v}_i / \tau - \log \left( \sum_{j=1}^{N} e^{\boldsymbol{u}_i^{\mathsf{T}} \boldsymbol{v}_j / \tau} \right) \right].$$

Since $\boldsymbol{U} = \boldsymbol{V}$ (*symmetric* property), the contrastive loss satisfies

$$\mathcal{L}^{\text{con}}(\boldsymbol{U}, \boldsymbol{V}) = 2\mathcal{L}(\boldsymbol{U}, \boldsymbol{U})$$

$$= -\frac{2}{N} \sum_{i=1}^{N} \left[ \boldsymbol{u}_i^{\mathsf{T}} \boldsymbol{u}_i / \tau - \log \left( \sum_{j=1}^{N} e^{\boldsymbol{u}_i^{\mathsf{T}} \boldsymbol{u}_j / \tau} \right) \right]$$

$$= -\frac{2}{\tau} + \frac{2}{N} \sum_{i=1}^{N} \log \left( \sum_{j=1}^{N} e^{\boldsymbol{u}_i^{\mathsf{T}} \boldsymbol{u}_j / \tau} \right). \tag{14}$$

Since $\|\boldsymbol{u}_i\| = 1$ for any $i \in [N]$, we can derive the following relations:

$$\|\boldsymbol{u}_i - \boldsymbol{u}_j\|^2 = 2 - 2\boldsymbol{u}_i^{\mathsf{T}} \boldsymbol{u}_j, \quad \boldsymbol{u}_i^{\mathsf{T}} \boldsymbol{u}_j = 1 - \frac{\|\boldsymbol{u}_i - \boldsymbol{u}_j\|^2}{2}.$$

We incorporate these relations into Equation (14) as follows:

$$\mathcal{L}^{\text{con}}(\boldsymbol{U}, \boldsymbol{V}) = -\frac{2}{\tau} + \frac{2}{N} \sum_{i=1}^{N} \log \left( \sum_{j=1}^{N} e^{1/\tau - \|\boldsymbol{u}_i - \boldsymbol{u}_j\|^2 / 2\tau} \right)$$

$$= \frac{2}{N} \sum_{i=1}^{N} \log \left( \sum_{j=1}^{N} e^{-\|\boldsymbol{u}_i - \boldsymbol{u}_j\|^2 / 2\tau} \right).$$

The antipodal property of $\boldsymbol{U}$ indicates that for each $i \in [N]$, there exists a $j(i)$ such that $u_{j(i)} = -u_i$. By applying this property, we can manipulate the summation of $e^{-\|\boldsymbol{u}_i - \boldsymbol{u}_j\|^2 / 2}$ over $j$ as the following:

$$\sum_{j=1}^{N} e^{-\|\boldsymbol{u}_i - \boldsymbol{u}_j\|^2 / 2\tau} = e^{-\|\boldsymbol{u}_i - \boldsymbol{u}_i\|^2 / 2\tau} + e^{-\|\boldsymbol{u}_i - \boldsymbol{u}_{j(i)}\|^2 / 2\tau} + \sum_{j \neq i, j(i)} e^{-\|\boldsymbol{u}_i - \boldsymbol{u}_j\|^2 / 2\tau}$$

$$= 1 + e^{-2/\tau} + \sum_{j \neq i, j(i)} e^{-\|\boldsymbol{u}_i - \boldsymbol{u}_j\|^2 / 2\tau}.$$

Therefore,

$$\mathcal{L}^{\text{con}}(\boldsymbol{U}, \boldsymbol{V}) = \frac{2}{N} \sum_{i=1}^{N} \log \left( 1 + e^{-2/\tau} + \sum_{j \neq i, j(i)} e^{-\|\boldsymbol{u}_i - \boldsymbol{u}_j\|^2 / 2\tau} \right)$$

$$\overset{(a)}{\geq} \frac{2}{N(N-2)} \sum_{i=1}^{N} \sum_{j \neq i, j(i)} \log \left( 1 + e^{-2/\tau} + (N-2) e^{-\|\boldsymbol{u}_i - \boldsymbol{u}_j\|^2 / 2\tau} \right)$$

$$= \frac{2}{N(N-2)} \sum_{i=1}^{N} \sum_{j \neq i} \log \left( 1 + e^{-2/\tau} + (N-2) e^{-\|\boldsymbol{u}_i - \boldsymbol{u}_j\|^2 / 2\tau} \right) - \frac{2}{N-2} \log(1 + (N-1) e^{-2/\tau})$$

$$\overset{(b)}{\geq} \frac{2}{N(N-2)} \sum_{i=1}^{N} \sum_{j \neq i} \log \left( 1 + e^{-2/\tau} + (N-2) e^{-\|\boldsymbol{u}_i^\star - \boldsymbol{u}_j^\star\|^2 / 2\tau} \right) - \frac{2}{N-2} \log(1 + (N-1) e^{-2/\tau}),$$

where (a) follows by applying Jensen's inequality to the concave function $f(t) = \log(1 + e^{-2/\tau} + t)$; and (b) follows by Lemma 3, and the fact that function $g(t) = \log[1 + e^{-2/\tau} + (N-2)e^{-t/2\tau}]$ is convex and monotonically decreasing. $\{\boldsymbol{u}_1^\star, \cdots, \boldsymbol{u}_N^\star\}$ denotes a set of vectors which forms a cross-polytope.

Both inequalities in $(a)$ and $(b)$ are equalities only when the columns of $\boldsymbol{U}$ form a cross-polytope. Therefore, the columns of $\boldsymbol{U}^\star$ form a cross-polytope. $\qquad\square$

**Lemma 3.** *Given a function $g(t)$ is convex and monotonically decreasing, let*

$$\boldsymbol{U}^* := \arg\min_{\boldsymbol{U} \in \mathcal{A}} \sum_{i=1}^{N} \sum_{j \neq i} g(\|\boldsymbol{u}_i - \boldsymbol{u}_j\|^2) \quad s.t. \quad \|\boldsymbol{u}_i\| = 1, \|\boldsymbol{v}_i\| = 1 \quad \forall i \in [N], \tag{15}$$

*where $\mathcal{A} := \{\boldsymbol{U} : \boldsymbol{U} \text{ is antipodal}\}$. Then, the columns of $\boldsymbol{U}^*$ form a simplex cross-polytope for $N = 2d$.*

*Proof.* Suppose $N = 2d$ and $\boldsymbol{U} \in \mathcal{A}$. Given a function $g(t)$ is convex and monotonically decreasing. $j(i)$ denotes the corresponding index for $i$ such that $\boldsymbol{u}_{j(i)} = -\boldsymbol{u}_i$, and $\|\boldsymbol{u}_i - \boldsymbol{u}_{j(i)}\|^2 = 4$. Under these conditions, we derive the following:

$$\sum_{i=1}^{N} \sum_{j \neq i} g(\|\boldsymbol{u}_i - \boldsymbol{u}_j\|^2) = Ng(4) + \sum_{i=1}^{N} \sum_{j \neq i, j(i)} g(\|\boldsymbol{u}_i - \boldsymbol{u}_j\|^2)$$

$$\overset{(a)}{\geq} Ng(4) + N(N-2)g\Big(\frac{1}{N(N-2)} \sum_{i=1}^{N} \sum_{j \neq i, j(i)} \|\boldsymbol{u}_i - \boldsymbol{u}_j\|^2\Big)$$

$$= Ng(4) + N(N-2)g\Big(\frac{1}{N(N-2)}\Big(-4N + \sum_{i=1}^{N} \sum_{j=1}^{N} \|\boldsymbol{u}_i - \boldsymbol{u}_j\|^2\Big)\Big)$$

$$= Ng(4) + N(N-2)g\Big(\frac{1}{N(N-2)}\Big(-4N + \sum_{i=1}^{N} \sum_{j=1}^{N} (2 - 2\boldsymbol{u}_i^\intercal \boldsymbol{u}_j)\Big)\Big)$$

$$= Ng(4) + N(N-2)g\Big(\frac{1}{N(N-2)}\Big(-4N + 2N^2 - \Big\|\sum_{i=1}^{N} \boldsymbol{u}_i\Big\|^2\Big)\Big)$$

$$\overset{(b)}{\geq} Ng(4) + N(N-2)g\Big(\frac{1}{N(N-2)}\Big(-4N + 2N^2\Big)\Big)$$

$$= Ng(4) + N(N-2)g(2),$$

where $(a)$ follows by Jensen's inequality; and (b) follows from the fact that $\|\sum_{i=1}^{N} \boldsymbol{u}_i\|^2 \geq 0$ and the function $g(t)$ is monotonically decreasing. The equality conditions for $(a)$ and $(b)$ only hold when the columns of $\boldsymbol{U}$ form a cross-polytope. We can conclude that the columns of $\boldsymbol{U}^\star$ form a cross polytope. $\qquad\square$

**Proposition 1.** *The mini-batch loss and full-batch loss are not identical, nor is one a simple scaling of the other by a constant factor. In other words, when $\mathcal{S}_B = \left[\binom{N}{B}\right]$, for all $B \geq 2$, there exists no constant $c$ such that $\mathcal{L}_{\text{mini}}^{\text{con}}(\boldsymbol{U}, \boldsymbol{V}; \mathcal{S}_B) = c \cdot \mathcal{L}^{\text{con}}(\boldsymbol{U}, \boldsymbol{V}) \quad \text{for all} \quad \boldsymbol{U}, \boldsymbol{V}.$*

*Proof.* Consider $\widetilde{\boldsymbol{U}}, \widetilde{\boldsymbol{V}}$ defined such that $\tilde{\boldsymbol{u}}_i = \tilde{\boldsymbol{v}}_i = \boldsymbol{e}_i \; \forall i \in [N]$, where $\boldsymbol{e}_i$ is $i$-th unit vector in $\mathbb{R}^N$. First note that $\tilde{\boldsymbol{u}}_i^\intercal \tilde{\boldsymbol{v}}_i = 1 \; \forall i \in [N]$ and $\tilde{\boldsymbol{u}}_i^\intercal \tilde{\boldsymbol{v}}_j = 0 \; \forall i \neq j$. Then,

$$\mathcal{L}(\widetilde{\boldsymbol{U}}, \widetilde{\boldsymbol{V}}) = \log(e^{1/\tau} + N - 1) - 1/\tau, \tag{16}$$

$$\frac{1}{\binom{N}{B}} \sum_{i=1}^{\binom{N}{B}} \mathcal{L}(\widetilde{\boldsymbol{U}}_{\mathcal{B}_i}, \widetilde{\boldsymbol{V}}_{\mathcal{B}_i}) = \log(e^{1/\tau} + B - 1) - 1/\tau. \tag{17}$$

We now consider the second part of the statement. For contradiction, assume that there exists some $c \in \mathbb{R}$ such that $\mathcal{L}_{\text{mini}}^{\text{con}}(\boldsymbol{U}, \boldsymbol{V}; \mathcal{S}_B) = c \cdot \mathcal{L}^{\text{con}}(\boldsymbol{U}, \boldsymbol{V})$ for all $\boldsymbol{U}, \boldsymbol{V}$. Let $\widehat{\boldsymbol{U}}, \widehat{\boldsymbol{V}}$ be defined such that $\hat{\boldsymbol{u}}_i = \hat{\boldsymbol{v}}_i = \boldsymbol{e}_1 \; \forall i \in [N]$, where $\boldsymbol{e}_1 = (1, 0, \cdots, 0)$. Note that $\hat{\boldsymbol{u}}_i^\mathsf{T} \hat{\boldsymbol{v}}_j = 1 \; \forall i, j \in [N]$. Then,

$$\mathcal{L}(\widehat{\boldsymbol{U}}, \widehat{\boldsymbol{V}}) = \log(N), \tag{18}$$

$$\frac{1}{\binom{N}{B}} \sum_{i=1}^{\binom{N}{B}} \mathcal{L}(\widehat{\boldsymbol{U}}_{\mathcal{B}_i}, \widehat{\boldsymbol{V}}_{\mathcal{B}_i}) = \log(B). \tag{19}$$

From Equation (16) and 17, we have that $c = \frac{\log(e^{1/\tau} + B - 1) - 1/\tau}{\log(e^{1/\tau} + N - 1) - 1/\tau}$. Whereas from Equation (18) and (19), we have that $c = \frac{\log(B)}{\log(N)}$ which is a contradiction. Therefore, there exists no $c \in \mathbb{R}$ satisfying the given condition. □

**Theorem 4** (Optimization with all possible $\binom{N}{B}$ mini-batches). *Suppose $B \geq 2$. The set of minimizers of the $\binom{N}{B}$ mini-batch problem in Equation (4) is the same as that of the full-batch problem in Equation (1) for two cases: (i) $N \leq d + 1$, and (ii) $N = 2d$ and the pairs $(\boldsymbol{U}, \boldsymbol{V})$ are restricted to those satisfying the conditions stated in Def. 2. In such cases, the solutions $(\boldsymbol{U}, \boldsymbol{V})$ for the $\binom{N}{B}$ mini-batch optimization problem satisfies the following: Case (i) $\{\boldsymbol{u}_i\}_{i=1}^N$ forms a simplex ETF and $\boldsymbol{u}_i = \boldsymbol{v}_i$ for all $i \in [N]$; Case (ii): $\{\boldsymbol{u}_i\}_{i=1}^N$ forms a simplex cross-polytope.*

*Proof.* **Case (i)**: Suppose $N \leq d + 1$.

For simplicity, first consider just one of the two terms in the two-sided loss. Therefore, the optimization problem becomes

$$\min_{\boldsymbol{U}, \boldsymbol{V}} \quad \frac{1}{\binom{N}{B}} \sum_{i=1}^{\binom{N}{B}} \mathcal{L}(\boldsymbol{U}_{\mathcal{B}_i}, \boldsymbol{V}_{\mathcal{B}_i}) \quad s.t. \quad \|\boldsymbol{u}_i\| = 1, \|\boldsymbol{v}_i\| = 1 \; \forall i \in [N].$$

Similar to the proof of Lemma 1, we have that

$$\sum_{i=1}^{\binom{N}{B}} \mathcal{L}(\boldsymbol{U}_{\mathcal{B}_i}, \boldsymbol{V}_{\mathcal{B}_i}) = \frac{1}{B} \sum_{i=1}^{\binom{N}{B}} \sum_{j \in \mathcal{B}_i} \log \left( 1 + \sum_{\substack{k \in \mathcal{B}_i \\ k \neq j}} e^{\boldsymbol{u}_j^\mathsf{T} (\boldsymbol{v}_k - \boldsymbol{v}_j)/\tau} \right)$$

$$\overset{(a)}{\geq} \frac{1}{B} \sum_{i=1}^{\binom{N}{B}} \sum_{j \in \mathcal{B}_i} \log \left( 1 + (B - 1) \exp \left( \frac{\sum_{k \in \mathcal{B}_i, k \neq j} \boldsymbol{u}_j^\mathsf{T} (\boldsymbol{v}_k - \boldsymbol{v}_j)}{\tau (B - 1)} \right) \right)$$

$$= \frac{1}{B} \sum_{i=1}^{\binom{N}{B}} \sum_{j \in \mathcal{B}_i} \log \left( 1 + (B - 1) \exp \left( \frac{\sum_{k \in \mathcal{B}_i} (\boldsymbol{u}_j^\mathsf{T} \boldsymbol{v}_k - B \boldsymbol{u}_j^\mathsf{T} \boldsymbol{v}_j)}{\tau (B - 1)} \right) \right)$$

$$\overset{(b)}{\geq} \binom{N}{B} \log \left( 1 + (B - 1) \exp \left( \frac{\sum_{i=1}^{\binom{N}{B}} \sum_{j \in \mathcal{B}_i} \sum_{k \in \mathcal{B}_i} \boldsymbol{u}_j^\mathsf{T} \boldsymbol{v}_k - \sum_{i=1}^{\binom{N}{B}} \sum_{j \in \mathcal{B}_i} B \boldsymbol{u}_j^\mathsf{T} \boldsymbol{v}_j}{\tau \binom{N}{B} \cdot B \cdot (B - 1)} \right) \right),$$

where $(a)$ and $(b)$ follows by applying Jensen's inequality to $e^t$ and $\log(1 + ae^{bt})$ for $a, b > 0$, respectively. Note that for equalities to hold in Jensen's inequalities, we need constants $c_j, c$ such that

$$\boldsymbol{u}_j^\mathsf{T} \boldsymbol{v}_k = c_j \quad \forall k \neq j, \tag{20}$$

$$\frac{\boldsymbol{u}^\mathsf{T} \boldsymbol{v}_i}{N - 1} - \frac{N(\boldsymbol{u}_i^\mathsf{T} \boldsymbol{v}_i)}{N - 1} = c \quad \forall i \in [N]. \tag{21}$$

Now, we carefully consider the two terms in the numerator:

$$A_1 := \sum_{i=1}^{\binom{N}{B}} \sum_{j \in \mathcal{B}_i} \sum_{k \in \mathcal{B}_i} \boldsymbol{u}_j^{\mathsf{T}} \boldsymbol{v}_k, \quad A_2 := \sum_{i=1}^{\binom{N}{B}} \sum_{j \in \mathcal{B}_i} B \boldsymbol{u}_j^{\mathsf{T}} \boldsymbol{v}_j.$$

To simplify $A_1$, first note that for any fixed $l, m \in [N]$ such that $l \neq m$, there are $\frac{N-2}{B-2}$ batches that contain $l$ and $m$. And for $l = m$, there are $\binom{N-1}{B-1}$ batches that contain that pair. Since these terms all occur in $A_1$, we have that

$$A_1 = \binom{N-2}{B-2} \sum_{l=1}^{N} \sum_{m=1}^{N} \boldsymbol{u}_l^{\mathsf{T}} \boldsymbol{v}_m + \left[ \binom{N-1}{B-1} - \binom{N-2}{B-2} \right] \sum_{l=1}^{N} \boldsymbol{u}_l^{\mathsf{T}} \boldsymbol{v}_l$$

$$= \binom{N-2}{B-2} \sum_{l=1}^{N} \sum_{m=1}^{N} \boldsymbol{u}_l^{\mathsf{T}} \boldsymbol{v}_m + \binom{N-2}{B-2} \left( \frac{N-B}{B-1} \right) \sum_{l=1}^{N} \boldsymbol{u}_l^{\mathsf{T}} \boldsymbol{v}_l.$$

Similarly, we have that

$$A_2 = \binom{N-1}{B-1} B \sum_{l=1}^{N} \boldsymbol{u}_l^{\mathsf{T}} \boldsymbol{v}_l.$$

Plugging these back into the above inequality, we have that

$$\sum_{i=1}^{\binom{N}{B}} \mathcal{L}(\boldsymbol{U}_{\mathcal{B}_i}, \boldsymbol{V}_{\mathcal{B}_i}) \geq \binom{N}{B} \log \left( 1 + (B-1) \exp \left( \frac{\sum_{l=1}^{N} \sum_{m=1}^{N} \boldsymbol{u}_l^{\mathsf{T}} \boldsymbol{v}_m - N \sum_{l=1}^{N} \boldsymbol{u}_l^{\mathsf{T}} \boldsymbol{v}_l}{\tau N(N-1)} \right) \right)$$

$$= \binom{N}{B} \log \left( 1 + (B-1) \exp \left( \frac{\boldsymbol{u}^{\mathsf{T}} \boldsymbol{v} - N \sum_{i=1}^{N} \boldsymbol{u}_i^{\mathsf{T}} \boldsymbol{v}_i}{\tau N(N-1)} \right) \right).$$

Observe that the term inside the exponential is identical to Equation (9) and therefore, we can reuse the same spectral analysis argument to show that the simplex ETF also minimizes $\sum_{i=1}^{\binom{N}{B}} \mathcal{L}(\boldsymbol{U}_{\mathcal{B}_i}, \boldsymbol{V}_{\mathcal{B}_i})$. Once again, since the proof is symmetric the simplex ETF also minimizes $\sum_{i=1}^{\binom{N}{B}} \mathcal{L}(\boldsymbol{V}_{\mathcal{B}_i}, \boldsymbol{U}_{\mathcal{B}_i})$.

**Case (ii)**: Suppose $N = 2d$, and $\boldsymbol{U}, \boldsymbol{V}$ are symmetric and antipodal. Next, we consider the following optimization problem

$$\min_{(\boldsymbol{U},\boldsymbol{V}) \in \mathcal{A}} \quad \frac{1}{\binom{N}{B}} \sum_{i=1}^{\binom{N}{B}} \mathcal{L}^{\mathrm{con}}(\boldsymbol{U}_{\mathcal{B}_i}, \boldsymbol{V}_{\mathcal{B}_i}) \quad s.t. \quad \|\boldsymbol{u}_i\| = 1, \|\boldsymbol{v}_i\| = 1 \; \forall i \in [N], \tag{22}$$

where $\mathcal{A} := \{(\boldsymbol{U}, \boldsymbol{V}) : \boldsymbol{U}, \boldsymbol{V} \text{ are symmetric and antipodal}\}$. Since $\boldsymbol{U} = \boldsymbol{V}$ (*symmetric* property) the contrastive loss satisfies

$$\mathcal{L}^{\mathrm{con}}(\boldsymbol{U}_{\mathcal{B}_i}, \boldsymbol{V}_{\mathcal{B}_i}) = 2\mathcal{L}(\boldsymbol{U}_{\mathcal{B}_i}, \boldsymbol{U}_{\mathcal{B}_i})$$

$$= -\frac{2}{B} \sum_{j \in \mathcal{B}_i} \left[ \boldsymbol{u}_j^{\mathsf{T}} \boldsymbol{u}_j / \tau - \log \left( \sum_{k \in \mathcal{B}_i} e^{\boldsymbol{u}_j^{\mathsf{T}} \boldsymbol{u}_k / \tau} \right) \right]$$

$$= -\frac{2}{\tau} + \frac{2}{B} \sum_{j \in \mathcal{B}_i} \log \left( \sum_{k \in \mathcal{B}_i} e^{\boldsymbol{u}_j^{\mathsf{T}} \boldsymbol{u}_k / \tau} \right). \tag{23}$$

Therefore, the solution of the optimization problem in Equation (22) is identical to the minimizer of the following optimization problem:

$$\boldsymbol{U}^{\star} := \arg\min_{\boldsymbol{U}} \quad \sum_{i=1}^{\binom{N}{B}} \sum_{j\in\mathcal{B}_i} \log\Big(\sum_{k\in\mathcal{B}_i} e^{\boldsymbol{u}_j^\intercal \boldsymbol{u}_k/\tau}\Big).$$

The objective of the optimization problem can be rewritten by reorganizing summations as

$$\sum_{j=1}^{N} \sum_{i\in\mathcal{I}_j} \log\Big(\sum_{k\in\mathcal{B}_i} e^{\boldsymbol{u}_j^\intercal \boldsymbol{u}_k/\tau}\Big), \tag{24}$$

where $\mathcal{I}_j := \{i : j \in \mathcal{B}_i\}$ represents the set of batch indices containing $j$. We then divide the summation term in Equation (24) into two terms:

$$\sum_{j=1}^{N} \sum_{i\in\mathcal{I}_j} \log\Big(\sum_{k\in\mathcal{B}_i} e^{\boldsymbol{u}_j^\intercal \boldsymbol{u}_k/\tau}\Big) = \sum_{j=1}^{N} \sum_{i\in\mathcal{A}_j} \log\Big(\sum_{k\in\mathcal{B}_i} e^{\boldsymbol{u}_j^\intercal \boldsymbol{u}_k/\tau}\Big) + \sum_{j=1}^{N} \sum_{i\in\mathcal{A}_j^c} \log\Big(\sum_{k\in\mathcal{B}_i} e^{\boldsymbol{u}_j^\intercal \boldsymbol{u}_k/\tau}\Big), \tag{25}$$

by partitioning the set $\mathcal{I}_j$ for each $j \in [N]$ into as the following with $k(j)$ being the index for which $u_{k(j)} = -u_j$:

$$\mathcal{A}_j := \{i : j \in \mathcal{B}_i, \text{ and } k(j) \in \mathcal{B}_i\}; \quad \mathcal{A}_j^c := \{i : j \in \mathcal{B}_i, \text{ and } k(j) \notin \mathcal{B}_i\}.$$

We will prove that the columns of $\boldsymbol{U}^*$ form a cross-polytope by showing that the minimizer of each term of the RHS in Equation (25) also forms a cross-polytope. Let us start with the first term of the RHS in Equation (25). Starting with applying Jensen's inequality to the concave function $f(x) := \log(e^{1/\tau} + e^{-1/\tau} + x)$, we get:

$$\sum_{j=1}^{N} \sum_{i\in\mathcal{A}_j} \log\Big(\sum_{k\in\mathcal{B}_i} e^{\boldsymbol{u}_j^\intercal \boldsymbol{u}_k/\tau}\Big) = \sum_{j=1}^{N} \sum_{i\in\mathcal{A}_j} \log\Big(e^{1/\tau} + e^{-1/\tau} + \sum_{k\in\mathcal{B}_i\setminus\{j,k(j)\}} e^{\boldsymbol{u}_j^\intercal \boldsymbol{u}_k/\tau}\Big)$$

$$\geq \frac{1}{B-2} \sum_{j=1}^{N} \sum_{i\in\mathcal{A}_j} \sum_{k\in\mathcal{B}_i\setminus\{j,k(j)\}} \log\big(e^{1/\tau} + e^{-1/\tau} + (B-2)e^{\boldsymbol{u}_j^\intercal \boldsymbol{u}_k/\tau}\big)$$

$$= \frac{1}{B-2} \sum_{j=1}^{N} \sum_{k\notin\{j,k(j)\}} \binom{N-3}{B-3} \log\big(e^{1/\tau} + e^{-1/\tau} + (B-2)e^{\boldsymbol{u}_j^\intercal \boldsymbol{u}_k/\tau}\big)$$

$$= \frac{\binom{N-3}{B-3}}{B-2} \Big[\sum_{j=1}^{N} \sum_{k\neq j} \log\big(e^{1/\tau} + e^{-1/\tau} + (B-2)e^{\boldsymbol{u}_j^\intercal \boldsymbol{u}_k/\tau}\big) - N\log\big(e^{1/\tau} + (B-1)e^{-1/\tau}\big)\Big]$$

$$= \frac{\binom{N-3}{B-3}}{B-2} \Big[\sum_{j=1}^{N} \sum_{k\neq j} \log\big(e^{1/\tau} + e^{-1/\tau} + (B-2)e^{1/\tau}\cdot e^{-\frac{\|\boldsymbol{u}_j - \boldsymbol{u}_k\|^2}{2\tau}}\big) - N\log\big(e^{1/\tau} + (B-1)e^{-1/\tau}\big)\Big]$$

$$\overset{(a)}{\geq} \frac{\binom{N-3}{B-3}}{B-2} \Big[\sum_{j=1}^{N} \sum_{k\neq j} \log\big(e^{1/\tau} + e^{-1/\tau} + (B-2)e^{1/\tau}\cdot e^{-\frac{\|\boldsymbol{u}_j^\star - \boldsymbol{u}_k^\star\|^2}{2\tau}}\big) - N\log\big(e^{1/\tau} + (B-1)e^{-1/\tau}\big)\Big],$$

where $(a)$ follows by Lemma 3 and the fact that $g(t) = \log(a + be^{-\frac{t}{2\tau}})$ for $a, b > 0$ is convex and monotonically decreasing. $\{\boldsymbol{u}_1^\star, \cdots, \boldsymbol{u}_N^\star\}$ denotes a set of vectors which forms a cross-polytope. All equalities hold only when the columns of $\boldsymbol{U}$ form a cross-polytope.

Next consider the second term of the RHS in Equation (25). By following a similar procedure above, we get:

$$
\sum_{j=1}^{N} \sum_{i \in \mathcal{A}_j^c} \log \Big( \sum_{k \in \mathcal{B}_i} e^{\boldsymbol{u}_j^\mathsf{T} \boldsymbol{u}_k/\tau} \Big) \geq \frac{1}{B-1} \sum_{j=1}^{N} \sum_{i \in \mathcal{A}_j} \sum_{k \in \mathcal{B}_i \setminus \{j\}} \log \Big( e^{1/\tau} + (B-1)e^{\boldsymbol{u}_j^\mathsf{T} \boldsymbol{u}_k/\tau} \Big)
$$

$$
= \frac{1}{B-1} \sum_{j=1}^{N} \sum_{k \notin \{j, k(j)\}} \binom{N-3}{B-2} \log \Big( e^{1/\tau} + (B-1)e^{\boldsymbol{u}_j^\mathsf{T} \boldsymbol{u}_k/\tau} \Big)
$$

$$
= \frac{\binom{N-3}{B-2}}{B-1} \Big[ \sum_{j=1}^{N} \sum_{k \neq j} \log \big( e^{1/\tau} + (B-1)e^{\boldsymbol{u}_j^\mathsf{T} \boldsymbol{u}_k/\tau} \big) - N \log \big( e^{1/\tau} + (B-1)e^{-1/\tau} \big) \Big]
$$

$$
\geq \frac{\binom{N-3}{B-2}}{B-1} \Big[ \sum_{j=1}^{N} \sum_{k \neq j} \log \big( e^{1/\tau} + (B-1)e^{1/\tau} \cdot e^{-\frac{\|\boldsymbol{u}_j^\star - \boldsymbol{u}_k^\star\|^2}{2\tau}} \big) - N \log \big( e^{1/\tau} + (B-1)e^{-1/\tau} \big) \Big],
$$

where $\{\boldsymbol{u}_1^\star, \cdots, \boldsymbol{u}_N^\star\}$ denotes a set of vectors which forms a cross-polytope.

Both terms of RHS in Equation (25) have the minimum value when $\boldsymbol{U}$ forms a cross-polytope. Therefore, we can conclude that the columns of $\boldsymbol{U}^\star$ form a cross-polytope. □

**Theorem 5** (Optimization with fewer than $\binom{N}{B}$ mini-batches). *Suppose $B = 2$ and $N \leq d+1$. Then, the minimizer of Equation (4) for $\mathcal{S}_B \subsetneq \left[\binom{N}{B}\right]$ is not the minimizer of the full-batch optimization in Equation (1).*

*Proof.* Consider a set of batches $\mathcal{S}_B \subset \left[\binom{N}{2}\right]$ with the batch size $B = 2$. Without loss of generality, assume that $(1, 2) \notin \bigcup_{i \in \mathcal{S}_B} \{\mathcal{B}_i\}$. For contradiction, assume that the simplex ETF - $(\boldsymbol{U}^\star, \boldsymbol{V}^\star)$ is indeed the optimal solution of the loss over these $\mathcal{S}_B$ batches. Then, by definition, we have that for any $(\boldsymbol{U}, \boldsymbol{V}) \neq (\boldsymbol{U}^\star, \boldsymbol{V}^\star)$,

$$
\frac{1}{|\mathcal{S}_B|} \sum_{i \in \mathcal{S}_B} \mathcal{L}(\boldsymbol{U}_{\mathcal{B}_i}^\star, \boldsymbol{V}_{\mathcal{B}_i}^\star) \leq \frac{1}{|\mathcal{S}_B|} \sum_{i \in \mathcal{S}_B} \mathcal{L}(\boldsymbol{U}_{\mathcal{B}_i}, \boldsymbol{V}_{\mathcal{B}_i})
$$

$$
\Rightarrow \sum_{i \in \mathcal{S}_B} \mathcal{L}(\boldsymbol{U}_{\mathcal{B}_i}^\star, \boldsymbol{V}_{\mathcal{B}_i}^\star) \leq \sum_{i \in \mathcal{S}_B} \mathcal{L}(\boldsymbol{U}_{\mathcal{B}_i}, \boldsymbol{V}_{\mathcal{B}_i}), \tag{26}
$$

where $(\boldsymbol{U}^\star, \boldsymbol{V}^\star)$ is defined such that $\boldsymbol{u}_i^\star = \boldsymbol{v}_i^\star$ for all $i \in [N]$ and $\boldsymbol{u}_i^{\star\mathsf{T}} \boldsymbol{v}_j^\star = -1/(N-1)$ for all $i \neq j$. Also recall that $\|\boldsymbol{u}_i\| = \|\boldsymbol{v}_i\| = 1$ for all $i \in [N]$. Therefore, we also have

$$
\sum_{i \in \mathcal{S}_B} \mathcal{L}(\boldsymbol{U}_{\mathcal{B}_i}^\star, \boldsymbol{V}_{\mathcal{B}_i}^\star) = \sum_{i \in \mathcal{S}_B} \sum_{j \in \mathcal{B}_i} \log \Big( 1 + \sum_{k \in \mathcal{B}_i, k \neq j} \exp \big( \boldsymbol{u}_j^{\star\mathsf{T}} (\boldsymbol{v}_k^\star - \boldsymbol{v}_j^\star)/\tau \big) \Big)
$$

$$
= \sum_{i \in \mathcal{S}_B} \sum_{j \in \mathcal{B}_i} \log \Big( 1 + \sum_{k \in \mathcal{B}_i, k \neq j} \exp \Big( -\frac{1}{\tau(N-1)} - \frac{1}{\tau} \Big) \Big)
$$

$$
= \sum_{i \in \mathcal{S}_B} \sum_{j \in \mathcal{B}_i} \log \Big( 1 + \exp \Big( -\frac{1}{\tau(N-1)} - \frac{1}{\tau} \Big) \Big), \tag{27}
$$

where the last equality is due to the fact that $|\mathcal{B}_i| = 2$.

Now, let us consider $(\widetilde{\boldsymbol{U}}, \widetilde{\boldsymbol{V}})$ defined such that $\tilde{\boldsymbol{u}}_i = \tilde{\boldsymbol{v}}_i$ for all $i \in [N]$, and $\tilde{\boldsymbol{u}}_i^\mathsf{T} \tilde{\boldsymbol{v}}_j = -1/(N-2)$ for all $i \neq j, (i, j) \notin \{(1, 2), (2, 1)\}$. Intuitively, this is equivalent to placing $\tilde{\boldsymbol{u}}_2, \ldots, \tilde{\boldsymbol{u}}_N$ on a simplex ETF of $N-1$ points and setting $\tilde{\boldsymbol{u}}_1 = \tilde{\boldsymbol{u}}_2$. This is clearly possible because $d > N-1 \Rightarrow d > N-2$, which is the condition

required to place $N - 1$ points on a simplex ETF in $\mathbb{R}^d$. Therefore,

$$
\begin{aligned}
\sum_{i \in \mathcal{S}_B} \mathcal{L}(\widetilde{\boldsymbol{U}}_{\mathcal{B}_i}, \widetilde{\boldsymbol{V}}_{\mathcal{B}_i}) &= \sum_{i \in \mathcal{S}_B} \sum_{j \in \mathcal{B}_i} \log \left( 1 + \sum_{k \in \mathcal{B}_i, k \neq j} \exp \left( \tilde{\boldsymbol{u}}_j^{\mathsf{T}} (\tilde{\boldsymbol{v}}_k - \tilde{\boldsymbol{v}}_j) / \tau \right) \right) \\
&= \sum_{i \in \mathcal{S}_B} \sum_{j \in \mathcal{B}_i} \log \left( 1 + \sum_{k \in \mathcal{B}_i, k \neq j} \exp \left( -\frac{1}{\tau(N-2)} - \frac{1}{\tau} \right) \right) \\
&= \sum_{i \in \mathcal{S}_B} \sum_{j \in \mathcal{B}_i} \log \left( 1 + \exp \left( -\frac{1}{\tau(N-2)} - \frac{1}{\tau} \right) \right),
\end{aligned}
\tag{28}
$$

where the last equality follows since $(1,2) \notin \bigcup_{i \in \mathcal{S}_B} \{\mathcal{B}_i\}$. It is easy to see from Equation (27) and (28) that $\sum_{i \in \mathcal{S}_B} \mathcal{L}(\widetilde{\boldsymbol{U}}_{\mathcal{B}_i}, \widetilde{\boldsymbol{V}}_{\mathcal{B}_i}) < \sum_{i \in \mathcal{S}_B} \mathcal{L}(\boldsymbol{U}_{\mathcal{B}_i}^\star, \boldsymbol{V}_{\mathcal{B}_i}^\star)$ which contradicts Equation (26). Therefore, the optimal solution of minimizing the contrastive loss over any $\mathcal{S}_B \subset \left[ \binom{N}{2} \right]$ batches is not the simplex ETF completing the proof. $\qquad \square$

**Proposition 2.** *Suppose $B \geq 2$, and let $\mathcal{S}_B \subseteq \left[ \binom{N}{B} \right]$ be a set of mini-batch indices. If there exist two data points that never belong together in any mini-batch, i.e., $\exists i, j \in [N]$ s.t. $\{i, j\} \not\subset \mathcal{B}_k$ for all $k \in \mathcal{S}_B$, then the optimal solution of Equation (4) is not the minimizer of the full-batch problem in Equation (1).*

*Proof.* The proof follows in a fairly similar manner to that of Theorem 5. Consider a set of batches of size $B \geq 2$, $\mathcal{S}_B \subset [\binom{N}{B}]$. Without loss of generality, assume that $\{1, 2\} \not\subset \mathcal{B}_k$ for any $k \in \mathcal{S}_B$. For contradiction, assume that the simplex ETF - $(\boldsymbol{U}^\star, \boldsymbol{V}^\star)$ is the optimal solution of the loss over these $\mathcal{S}_B$ batches. Then, by definition, we have that for any $(\boldsymbol{U}, \boldsymbol{V}) \neq (\boldsymbol{U}^\star, \boldsymbol{V}^\star)$

Once again, for contradiction assume that the simplex ETF - $(\boldsymbol{U}^\star, \boldsymbol{V}^\star)$ is indeed the optimal solution of the loss over these $\mathcal{S}_B$ batches. Then, by definition for any $(\boldsymbol{U}, \boldsymbol{V}) \neq (\boldsymbol{U}^\star, \boldsymbol{V}^\star)$,

$$
\begin{aligned}
&\frac{1}{|\mathcal{S}_B|} \sum_{i \in \mathcal{S}_B} \mathcal{L}(\boldsymbol{U}_{\mathcal{B}_i}^\star, \boldsymbol{V}_{\mathcal{B}_i}^\star) \leq \frac{1}{|\mathcal{S}_B|} \sum_{i \in \mathcal{S}_B} \mathcal{L}(\boldsymbol{U}_{\mathcal{B}_i}, \boldsymbol{V}_{\mathcal{B}_i}) \\
&\Rightarrow \sum_{i \in \mathcal{S}_B} \mathcal{L}(\boldsymbol{U}_{\mathcal{B}_i}^\star, \boldsymbol{V}_{\mathcal{B}_i}^\star) \leq \sum_{i \in \mathcal{S}_B} \mathcal{L}(\boldsymbol{U}_{\mathcal{B}_i}, \boldsymbol{V}_{\mathcal{B}_i}),
\end{aligned}
\tag{29}
$$

where $(\boldsymbol{U}^\star, \boldsymbol{V}^\star)$ is defined such that $\boldsymbol{u}_i^\star = \boldsymbol{v}_i^\star$ for all $i \in [N]$ and $\boldsymbol{u}_i^{\star\mathsf{T}} \boldsymbol{v}_j^\star = -1/(N-1)$ for all $i \neq j$. Also recall that $\|\boldsymbol{u}_i\| = \|\boldsymbol{v}_i\| = 1$ for all $i \in [N]$. Therefore, we also have

$$
\begin{aligned}
\sum_{i \in \mathcal{S}_B} \mathcal{L}(\boldsymbol{U}_{\mathcal{B}_i}^\star, \boldsymbol{V}_{\mathcal{B}_i}^\star) &= \frac{1}{B} \sum_{i \in \mathcal{S}_B} \sum_{j \in \mathcal{B}_i} \log \left( 1 + \sum_{k \in \mathcal{B}_i, k \neq j} \exp \left( \boldsymbol{u}_j^{\star\mathsf{T}} (\boldsymbol{v}_k^\star - \boldsymbol{v}_j^\star) / \tau \right) \right) \\
&= \frac{1}{B} \sum_{i \in \mathcal{S}_B} \sum_{j \in \mathcal{B}_i} \log \left( 1 + \sum_{k \in \mathcal{B}_i, k \neq j} \exp \left( -\frac{1}{\tau(N-1)} - \frac{1}{\tau} \right) \right) \\
&= \frac{1}{B} \sum_{i \in \mathcal{S}_B} \sum_{j \in \mathcal{B}_i} \log \left( 1 + (B-1) \exp \left( -\frac{1}{\tau(N-1)} - \frac{1}{\tau} \right) \right).
\end{aligned}
\tag{30}
$$

Now, let us consider $(\widetilde{\boldsymbol{U}}, \widetilde{\boldsymbol{V}})$ defined such that $\tilde{\boldsymbol{u}}_i = \tilde{\boldsymbol{v}}_i$ for all $i \in [N]$, $\tilde{\boldsymbol{u}}_2 = \tilde{\boldsymbol{v}}_2$ and $\tilde{\boldsymbol{u}}_i^{\mathsf{T}} \tilde{\boldsymbol{v}}_j = -1/(N-2)$ for all $i \neq j, (i, j) \notin \{(1, 2), (2, 1)\}$. Once again, note that this is equivalent to placing $\tilde{\boldsymbol{u}}_2, \ldots, \tilde{\boldsymbol{u}}_N$ on a simplex

ETF of $N-1$ points and setting $\tilde{\boldsymbol{u}}_1 = \tilde{\boldsymbol{u}}_2$. Hence,

$$
\begin{aligned}
\sum_{i \in \mathcal{S}_B} \mathcal{L}(\widetilde{\boldsymbol{U}}_{\mathcal{B}_i}, \widetilde{\boldsymbol{V}}_{\mathcal{B}_i}) &= \frac{1}{B} \sum_{i \in \mathcal{S}_B} \sum_{j \in \mathcal{B}_i} \log \left( 1 + \sum_{k \in \mathcal{B}_i, k \neq j} \exp \left( \tilde{\boldsymbol{u}}_j^\intercal (\tilde{\boldsymbol{v}}_k - \tilde{\boldsymbol{v}}_j)/\tau \right) \right) \\
&= \frac{1}{B} \sum_{i \in \mathcal{S}_B} \sum_{j \in \mathcal{B}_i} \log \left( 1 + \sum_{k \in \mathcal{B}_i, k \neq j} \exp \left( -\frac{1}{\tau(N-2)} - \frac{1}{\tau} \right) \right) \\
&= \frac{1}{B} \sum_{i \in \mathcal{S}_B} \sum_{j \in \mathcal{B}_i} \log \left( 1 + (B-1) \exp \left( -\frac{1}{\tau(N-2)} - \frac{1}{\tau} \right) \right),
\end{aligned}
\tag{31}
$$

where for the final equality note that following. The only pair for which $\tilde{\boldsymbol{u}}_j^\intercal \tilde{\boldsymbol{v}}_k \neq -1/(N-2)$ is $(j,k) = (1,2)$. Since there is no $i \in \mathcal{S}_B$ such that $\{1,2\} \in \mathcal{B}_i$, this term never appears in our loss. From Equation (30) and Equation (31), we have that $\sum_{i \in \mathcal{S}_B} \mathcal{L}(\widetilde{\boldsymbol{U}}_{\mathcal{B}_i}, \widetilde{\boldsymbol{V}}_{\mathcal{B}_i}) < \sum_{i \in \mathcal{S}_B} \mathcal{L}(\boldsymbol{U}_{\mathcal{B}_i}^\star, \boldsymbol{V}_{\mathcal{B}_i}^\star)$ which contradicts Equation (29). Therefore, we conclude that the optimal solution of the contrastive loss over any $\mathcal{S}_B \subset \left[ \binom{N}{2} \right]$ batches is not the simplex ETF. $\qquad \square$

**Proposition 3.** *Suppose $B \geq 2$, and let $\mathcal{S}_B \subseteq \left[ \binom{N}{B} \right]$ be a set of mini-batch inidices satisfying $\mathcal{B}_i \bigcap \mathcal{B}_j = \varnothing, \forall i,j \in \mathcal{S}_B$ and $\bigcup_{i \in \mathcal{S}_B} \mathcal{B}_i = [N]$, i.e., $\{\mathcal{B}_i\}_{i \in \mathcal{S}_B}$ forms non-overlapping mini-batches that cover all data samples. Then, the minimizer of the mini-batch loss optimization problem in Equation (4) is different from the minimizer of the full-batch loss optimization problem in Equation (1).*

*Proof.* Applying Lemma 1 specifically to a single batch $\mathcal{B}_i$ gives us that the optimal solution for just the loss over this batch is the simplex ETF over $B$ points. In the case of non-overlapping batches, the objective function can be separated across batches and therefore the optimal solution for the sum of the losses is equal to the solution of minimizing each term independently. More precisely, we have

$$
\min_{\boldsymbol{U}, \boldsymbol{V}} \sum_{i=1}^{N/B} \mathcal{L}^{\mathrm{con}}(\boldsymbol{U}_{\mathcal{B}_i}, \boldsymbol{V}_{\mathcal{B}_i}) = \sum_{i=1}^{N/B} \min_{\boldsymbol{U}_{\mathcal{B}_i}, \boldsymbol{V}_{\mathcal{B}_i}} \mathcal{L}^{\mathrm{con}}(\boldsymbol{U}_{\mathcal{B}_i}, \boldsymbol{V}_{\mathcal{B}_i}),
$$

where $\boldsymbol{U}_{\mathcal{B}_i} = \{\boldsymbol{u}_j : j \in \mathcal{B}_i\}$ and $\boldsymbol{V}_{\mathcal{B}_i} = \{\boldsymbol{v}_j : j \in \mathcal{B}_i\}$, respectively, and the equality follows from the fact that $\mathcal{B}_i$'s are disjoint. $\qquad \square$

## C.2 Proofs of Results From Section 5

**Lemma 2.** *Contrastive loss $\mathcal{L}^{\mathrm{con}}(\boldsymbol{U}, \boldsymbol{V})$ is a geodesically non-quasi-convex function of $\boldsymbol{U}, \boldsymbol{V}$ on $\mathcal{T} = \{(\boldsymbol{U}, \boldsymbol{V}) : \|\boldsymbol{u}_i\| = \|\boldsymbol{v}_i\| = 1, \forall i \in [N]\}$.*

*Proof.* Without loss of generality, we focus on the case where $\tau = 1$. The contrastive loss function $\mathcal{L}^{\mathrm{con}}$ is geodesically quasi-convex if for any two points $(\boldsymbol{U}, \boldsymbol{V})$ and $(\boldsymbol{U}', \boldsymbol{V}')$ in the domain and for all $t$ in $[0,1]$:

$$
\mathcal{L}^{\mathrm{con}}(t(\boldsymbol{U}, \boldsymbol{V}) + (1-t)(\boldsymbol{U}', \boldsymbol{V}')) \leq \max\{\mathcal{L}^{\mathrm{con}}(\boldsymbol{U}, \boldsymbol{V}), \mathcal{L}^{\mathrm{con}}(\boldsymbol{U}', \boldsymbol{V}')\}.
$$

We provide a counter-example for geodesically quasi-convexity, which is a triplet of points $(\boldsymbol{U}^1, \boldsymbol{V}^1)$, $(\boldsymbol{U}^2, \boldsymbol{V}^2)$, $(\boldsymbol{U}^3, \boldsymbol{V}^3)$ where $(\boldsymbol{U}^3, \boldsymbol{V}^3)$ is on the geodesic between other two points and satisfies $\mathcal{L}^{\mathrm{con}}(\boldsymbol{U}^3, \boldsymbol{V}^3) > \max\{\mathcal{L}^{\mathrm{con}}(\boldsymbol{U}^1, \boldsymbol{V}^1), \mathcal{L}^{\mathrm{con}}(\boldsymbol{U}^2, \boldsymbol{V}^2)\}$. Let $N = 2$ and

$$
\boldsymbol{U}^1 = \begin{bmatrix} \sqrt{\frac{1}{2}} & \sqrt{\frac{2}{5}} \\ \sqrt{\frac{1}{2}} & \sqrt{\frac{1}{5}} \end{bmatrix}, \boldsymbol{U}^2 = \begin{bmatrix} \sqrt{\frac{1}{2}} & \sqrt{\frac{1}{2}} \\ \sqrt{\frac{1}{2}} & \sqrt{\frac{1}{2}} \end{bmatrix}, \boldsymbol{V}^1 = \begin{bmatrix} \sqrt{\frac{1}{2}} & \sqrt{\frac{1}{2}} \\ \sqrt{\frac{1}{2}} & \sqrt{\frac{1}{2}} \end{bmatrix}, \boldsymbol{V}^2 = \begin{bmatrix} \sqrt{\frac{2}{5}} & \sqrt{\frac{1}{2}} \\ \sqrt{\frac{1}{5}} & \sqrt{\frac{1}{2}} \end{bmatrix}.
$$

Now, define $\boldsymbol{U}^3 = \mathrm{normalize}((\boldsymbol{U}^1 + \boldsymbol{U}^2)/2)$ and $\boldsymbol{V}^3 = \mathrm{normalize}((\boldsymbol{V}^1 + \boldsymbol{V}^2)/2)$, which is the "midpoint" of the geodesic between $(\boldsymbol{U}^1, \boldsymbol{V}^1)$ and $(\boldsymbol{U}^2, \boldsymbol{V}^2)$. By direct calculation, we obtain $\mathcal{L}^{\mathrm{con}}(\boldsymbol{U}^3, \boldsymbol{V}^3) \approx 2.798 > 2.773 \approx \max(\mathcal{L}^{\mathrm{con}}(\boldsymbol{U}^1, \boldsymbol{V}^1), \mathcal{L}^{\mathrm{con}}(\boldsymbol{U}^2, \boldsymbol{V}^2))$, which indicates $\mathcal{L}^{\mathrm{con}}$ is geodesically non-quasi-convex. $\qquad \square$

**Theorem 8** (Theorem 6 restated). *Consider $N = 4$ samples and their embedding vectors $\{\boldsymbol{u}_i\}_{i=1}^N$, $\{\boldsymbol{v}_i\}_{i=1}^N$ with dimension $d = 2$. Suppose $\boldsymbol{u}_i$'s are parametrized by $\boldsymbol{\theta}^{(t)} = [\theta_1^{(t)}, \theta_2^{(t)}, \theta_3^{(t)}, \theta_4^{(t)}]$ as in the setting described in Section 5.1 (see Figure 2(a)). Consider initializing $\boldsymbol{u}_i^{(0)} = \boldsymbol{v}_i^{(0)}$ and $\theta_i^{(0)} = \epsilon > 0$ for all $i$, then updating $\boldsymbol{\theta}^{(t)}$ via OSGD and SGD with the batch size $B = 2$ as described in Section 5.1. Let $T_{\mathrm{OSGD}}, T_{\mathrm{SGD}}$ be the minimal time required for OSGD, SGD algorithm to have $\mathbb{E}[\boldsymbol{\theta}^{(T)}] \in (\pi/4 - \rho, \pi/4)^N$. Suppose there exist $\tilde{\epsilon}$, $\overline{T}$ such that for all $t$ satisfying $\mathcal{B}^{(t)} = \{1,3\}$ or $\{2,4\}$, $\|\nabla_{\boldsymbol{\theta}^{(t)}} \mathcal{L}^{\mathrm{con}}(\boldsymbol{U}_{\mathcal{B}^{(t)}}, \boldsymbol{V}_{\mathcal{B}^{(t)}})\| \leq \tilde{\epsilon}$, and $T_{\mathrm{OSGD}}, T_{\mathrm{SGD}} < \overline{T}$. Then,*

$$T_{\mathrm{OSGD}} \geq \tau \cdot \frac{\pi/4 - \rho - \epsilon + O(\eta^2 \epsilon + \eta \epsilon^3)}{\eta \epsilon}, \quad T_{\mathrm{SGD}} \geq \frac{3(e^{2/\tau} + 1)}{e^{2/\tau} - 1} \tau \cdot \frac{\pi/4 - \rho - \epsilon + O(\eta^2 \epsilon + \eta^2 \tilde{\epsilon})}{\eta \epsilon + O(\eta \epsilon^3 + \eta \tilde{\epsilon})}.$$

*Proof.* We begin with the proof of

$$T_{\mathrm{OSGD}} \geq \tau \cdot \frac{\pi/4 - \rho - \epsilon + O(\eta^2 \epsilon + \eta \epsilon^3)}{\eta \epsilon}.$$

Assume that the parameters are initialized at $(\theta_1^{(0)}, \theta_2^{(0)}, \theta_3^{(0)}, \theta_4^{(0)}) = (\epsilon, \epsilon, \epsilon, \epsilon)$. Then, there are six batches with the batch size $B = 2$, and we can categorize the batches according to the mini-batch contrastive loss ($\tau = 1$):

1. $\mathcal{B} = \{1, 2\}$ or $\{3, 4\}$: $\mathcal{L}^{\mathrm{con}}(\boldsymbol{U}_{\mathcal{B}}, \boldsymbol{V}_{\mathcal{B}}) = -2/\tau + 2\log(e^{1/\tau} + e^{\cos 2\epsilon/\tau})$;

2. $\mathcal{B} = \{1, 3\}$ or $\{2, 4\}$: $\mathcal{L}^{\mathrm{con}}(\boldsymbol{U}_{\mathcal{B}}, \boldsymbol{V}_{\mathcal{B}}) = -2/\tau + 2\log(e^{1/\tau} + e^{-1/\tau})$;

3. $\mathcal{B} = \{1, 4\}$ or $\{2, 3\}$: $\mathcal{L}^{\mathrm{con}}(\boldsymbol{U}_{\mathcal{B}}, \boldsymbol{V}_{\mathcal{B}}) = -2/\tau + 2\log(e^{1/\tau} + e^{-\cos 2\epsilon/\tau})$.

Without loss of generality, we assume that OSGD algorithm described in Algorithm 5 chooses the mini-batch $\mathcal{B} = \{1, 2\}$ corresponding to the highest loss at time $t = 0$, and updates the parameter as

$$\theta_1^{(1)} = \epsilon - \eta \nabla_{\theta_1} \mathcal{L}^{\mathrm{con}}(\boldsymbol{U}_{\mathcal{B}}, \boldsymbol{V}_{\mathcal{B}}), \ \theta_2^{(1)} = \epsilon - \eta \nabla_{\theta_2} \mathcal{L}^{\mathrm{con}}(\boldsymbol{U}_{\mathcal{B}}, \boldsymbol{V}_{\mathcal{B}}).$$

Then, for the next update, OSGD choose $\boldsymbol{u}_3, \boldsymbol{u}_4$ which is now closer than updated $\boldsymbol{u}_1, \boldsymbol{u}_2$. And $\boldsymbol{u}_3, \boldsymbol{u}_4$ would be updated as same as what previously $\boldsymbol{u}_1, \boldsymbol{u}_2$ have changed. Thus, $\theta_1$ updates only at the even time, and stays at the odd time, i.e.

$$\theta_1^{(t+1)} = \begin{cases} \theta_1^{(t)} - \eta \nabla_{\theta_1} \mathcal{L}^{\mathrm{con}}(\boldsymbol{U}_{\mathcal{B}}, \boldsymbol{V}_{\mathcal{B}}) & \text{if } t \text{ is even,} \\ \theta_1^{(t)} & \text{if } t \text{ is odd.} \end{cases}$$

Iterating this procedure, we can view OSGD algorithm as one-parameterized algorithm of parameter $\phi^{(t)} = \theta_1^{(2t)}$ as:

$$\phi^{(0)} = \epsilon, \quad \phi^{(t)} = \phi^{(t-1)} + \eta \, g(\phi^{(t-1)}), \quad \phi^{(T_{\mathrm{half}})} \in \left(\frac{\pi}{4} - \rho, \ \frac{\pi}{4}\right),$$

where $g(\phi) = 2\sin(2\phi)/\tau(1 + e^{(1-\cos(2\phi))/\tau})$, and $T_{\mathrm{half}} := T_{\mathrm{OSGD}}/2$. In the procedure of updates, we may assume that $\phi^{(t)} \in (0, \frac{\pi}{4})$ for all $t$. To analyze the drift of $\phi^{(t)}$, we firstly study smoothness of $g$;

$$g'(\phi) = \frac{4}{\tau} \cdot \frac{\cos 2\phi (1 + e^{(1-\cos 2\phi)/\tau}) - \frac{\sin^2 2\phi}{\tau} e^{(1-\cos 2\phi)/\tau}}{(1 + e^{(1-\cos 2\phi)/\tau})^2}.$$

We can observe that $\max_{\phi \in [0, \frac{\pi}{4}]} |g'(\phi)| = 2/\tau$, hence $g(\phi)$ has Lipschitz constant $2/\tau$, i.e.

$$\left| g(\phi^{(t-1)}) - g(\phi^{(0)}) \right| \leq \frac{2}{\tau} \left| \phi^{(t-1)} - \phi^{(0)} \right|.$$

Therefore,

$$\phi^{(t)} - \phi^{(t-1)} = \eta \left| g(\phi^{(t-1)}) \right|$$
$$\leq \eta |g(\epsilon)| + \frac{2\eta}{\tau}(\phi^{(t-1)} - \epsilon)$$
$$= \frac{2\eta}{\tau}\phi^{(t-1)} + O(\eta\epsilon^3),$$

where the first inequality is from Lipschitz-continuity of $g(\phi)$, and the second equality is from Taylor expansion of $g$ at $\epsilon = 0$ as;

$$g(\epsilon) = \frac{2}{\tau}\epsilon - \frac{4\tau + 6}{3\tau^2}\epsilon^3 + \cdots.$$

Hence, $\phi^{(t)} \leq (1 + 2\eta/\tau)\phi^{(t-1)} + O(\eta\epsilon^3)$ indicates that

$$\phi^{(T_{\mathrm{half}})} \leq (1 + 2\eta/\tau)^{T_{\mathrm{half}}}\epsilon + \overline{T}\,O(\eta\epsilon^3)$$
$$\leq (1 + (2\eta/\tau)T_{\mathrm{half}})\epsilon + O(\eta^2\epsilon + \eta\epsilon^3),$$

for some constant $\overline{T} > T_{\mathrm{OSGD}}$. Moreover $\frac{\pi}{4} - \rho < \phi^{(T_{\mathrm{half}})}$ implies that

$$T_{\mathrm{half}} \geq \frac{\tau}{2} \cdot \frac{\pi/4 - \rho - \epsilon + O(\eta\epsilon^3 + \eta^2\epsilon)}{\eta\epsilon}.$$

So, we obtain the lower bound of $T_{\mathrm{OSGD}}$ by doubling $T_{\mathrm{half}}$. We estimate of $T_{\mathrm{OSGD}}$.

Now, we study convergence rate of SGD algorithm. We claim that

$$T_{\mathrm{SGD}} \geq \frac{3(e^{2/\tau} + 1)}{e^{2/\tau} - 1}\tau \cdot \frac{\pi/4 - \rho - \epsilon + O(\eta^2(\epsilon + \tilde{\epsilon}))}{\eta\epsilon + O(\eta(\epsilon^3 + \tilde{\epsilon}))}.$$

Without loss of generality, we firstly focus on the drift of $\theta_1$. Since batch selection is random, given $\boldsymbol{\theta}^{(t)} = (\theta_1^{(t)}, \theta_2^{(t)}, \theta_3^{(t)}, \theta_4^{(t)})$:

1. $\mathcal{B} = \{1, 2\}$ with probability $1/6$. Then, $\mathcal{L}^{\mathrm{con}}(\boldsymbol{U}_\mathcal{B}, \boldsymbol{V}_\mathcal{B}) = -2/\tau + 2\log(e^{1/\tau} + e^{\cos(\theta_1^{(t)} + \theta_2^{(t)})/\tau})$ implies

$$\theta_1^{(t+1)} = \theta_1^{(t)} + \eta\frac{2\sin(\theta_1^{(t)} + \theta_2^{(t)})/\tau}{1 + e^{(1 - \cos(\theta_1^{(t)} + \theta_2^{(t)}))/\tau}}.$$

2. $\mathcal{B} = \{1, 3\}$ with probability $1/6$. At $t = 0$, the initial batch selection can be primarily categorized into three distinct sets; closely positioned vectors $\{\boldsymbol{u}_1, \boldsymbol{u}_2\}$ or $\{\boldsymbol{u}_3, \boldsymbol{u}_4\}$, vectors that form obtuse angles $\{\boldsymbol{u}_1, \boldsymbol{u}_4\}$ or $\{\boldsymbol{u}_2, \boldsymbol{u}_3\}$, and vectors diametrically opposed at $180°$, $\{\boldsymbol{u}_1, \boldsymbol{u}_3\}$ or $\{\boldsymbol{u}_2, \boldsymbol{u}_4\}$. Given that $\epsilon$ is substantially small, the possibility of consistently selecting batches from the same category for subsequent updates is relatively low. As such, it is reasonable to infer that each batch is likely to maintain its position within the initially assigned categories. From this, one can deduce that vector sets such as $\{\boldsymbol{u}_1, \boldsymbol{u}_3\}$ or $\{\boldsymbol{u}_2, \boldsymbol{u}_4\}$ continue to sustain an angle close to $180°$. Given these conditions, it is feasible to postulate that if the selected batch $\mathcal{B}$ encompasses either $\{1, 3\}$ or $\{2, 4\}$, the magnitude of the gradient of the loss function $\mathcal{L}^{\mathrm{con}}(U_\mathcal{B}, V_\mathcal{B})$, denoted by $\|\nabla\mathcal{L}^{\mathrm{con}}(U_\mathcal{B}, V_\mathcal{B})\|$, would be less than a particular threshold $\tilde{\epsilon}$, i.e.

$$\|\nabla\mathcal{L}^{\mathrm{con}}(U_\mathcal{B}, V_\mathcal{B})\| < \tilde{\epsilon}.$$

Then,

$$\theta_1^{(t+1)} = \theta_1^{(t)} + \eta O(\tilde{\epsilon}).$$

3. $\mathcal{B} = \{1, 4\}$ with probability $1/6$. Then, $\mathcal{L}^{\mathrm{con}}(\boldsymbol{U}_\mathcal{B}, \boldsymbol{V}_\mathcal{B}) = -2/\tau + 2\log(e^{1/\tau} + e^{-\cos(\theta_1 + \theta_4)/\tau})$ implies

$$\theta_1^{(t+1)} = \theta_1^{(t)} - \eta\frac{2\sin(\theta_1^{(t)} + \theta_4^{(t)})/\tau}{1 + e^{(1 + \cos(\theta_1^{(t)} + \theta_4^{(t)}))/\tau}}.$$

Since there is no update on $\theta_1$ for the other cases, taking expectation yields

$$\mathbb{E}[\theta_1^{(t+1)} - \theta_1^{(t)}|\boldsymbol{\theta}^{(t)}] = \frac{\eta}{6}F_1(\boldsymbol{\theta}^{(t)}) + O(\eta\tilde{\epsilon}),$$

where $F_1(\boldsymbol{\theta})$ is defined as:

$$F_1(\boldsymbol{\theta}) = \frac{2\sin(\theta_1 + \theta_2)/\tau}{1 + e^{(1-\cos(\theta_1+\theta_2))/\tau}} - \frac{2\sin(\theta_1 + \theta_4)/\tau}{1 + e^{(1+\cos(\theta_1+\theta_4))/\tau}}.$$

We study smoothness of $F_1$ by setting $F_1(\boldsymbol{\theta}) = f_-(\theta_1 + \theta_2) - f_+(\theta_1 + \theta_4)$, where

$$f_-(t) := \frac{2\sin t/\tau}{1 + e^{(1-\cos t)/\tau}}, \quad f_+(t) := \frac{2\sin t/\tau}{1 + e^{(1+\cos t)/\tau}}.$$

Note that

$$\max_{t\in[0,\pi/2]}|f_-'(t)| = 1/\tau, \quad \max_{t\in[0,\pi/2]}|f_+'(t)| = C/\tau,$$

for some constant $C \in (0,1)$. Then for $\boldsymbol{\theta} = (\theta_1, \theta_2, \theta_3, \theta_4), \boldsymbol{\theta}' = (\theta_1', \theta_2', \theta_3', \theta_4')$,

$$\begin{aligned}
|F_1(\boldsymbol{\theta}') - F_1(\boldsymbol{\theta})| &\leq |f_-(\theta_1' + \theta_2') - f_-(\theta_1 + \theta_2)| + |f_+(\theta_1' + \theta_4') - f_+(\theta_1 + \theta_4)| \\
&\leq (1/\tau)\cdot|\theta_1' + \theta_2' - \theta_1 - \theta_2| + (C/\tau)\cdot|\theta_1' + \theta_4' - \theta_1 - \theta_4| \\
&\leq (2(1+C)/\tau)\|\boldsymbol{\theta}' - \boldsymbol{\theta}\|.
\end{aligned}$$

In the same way, we can define the functions $F_2, F_3, F_4$ all having Lipschitz constant $2(1+C)/\tau$. As we define $F(\boldsymbol{\theta}) = (F_1(\boldsymbol{\theta}), F_2(\boldsymbol{\theta}), F_3(\boldsymbol{\theta}), F_4(\boldsymbol{\theta}))$, it has Lipschitz constant $4(1+C)/\tau$ satisfying that

$$\mathbb{E}[\boldsymbol{\theta}' - \boldsymbol{\theta}|\boldsymbol{\theta}] = \frac{\eta}{6}F(\boldsymbol{\theta}) + O(\eta\tilde{\epsilon}),$$

where Big $O(\cdot)$ is applied elementwise to the vector, denoting that each element follows $O(\cdot)$ independently. From Lipschitzness of $F$, for any $t \geq 1$,

$$\begin{aligned}
\mathbb{E}[\|\boldsymbol{\theta}^{(t)} - \boldsymbol{\theta}^{(t-1)}\||\boldsymbol{\theta}^{(t-1)}] &\leq \frac{\eta}{6}\|F(\boldsymbol{\theta}^{(t-1)})\| + O(\eta\tilde{\epsilon}) \\
&\leq \frac{\eta}{6}\|F(\boldsymbol{\theta}^{(0)})\| + \frac{\eta}{6}\|F(\boldsymbol{\theta}^{(t-1)}) - F(\boldsymbol{\theta}^{(0)})\| + O(\eta\tilde{\epsilon}) \\
&\leq \frac{\eta}{6}\|F(\boldsymbol{\theta}^{(0)})\| + \frac{2\eta(1+C)}{3\tau}\|\boldsymbol{\theta}^{(t-1)} - \boldsymbol{\theta}^{(0)}\| + O(\eta\tilde{\epsilon}).
\end{aligned}$$

By taking expecations for both sides,

$$\mathbb{E}[\|\boldsymbol{\theta}^{(t)} - \boldsymbol{\theta}^{(t-1)}\|] \leq \frac{\eta}{6}\|F(\boldsymbol{\theta}^{(0)})\| + \frac{2\eta(1+C)}{3\tau}\mathbb{E}[\|\boldsymbol{\theta}^{(t-1)} - \boldsymbol{\theta}^{(0)}\|] + O(\eta\tilde{\epsilon}).$$

Applying the triangle inequality, $\|\boldsymbol{\theta}^{(t)} - \boldsymbol{\theta}^{(0)}\| \leq \|\boldsymbol{\theta}^{(t)} - \boldsymbol{\theta}^{(t-1)}\| + \|\boldsymbol{\theta}^{(t-1)} - \boldsymbol{\theta}^{(0)}\|$, we further deduce that

$$\mathbb{E}[\|\boldsymbol{\theta}^{(t)} - \boldsymbol{\theta}^{(0)}\|] \leq \left(1 + \frac{2\eta(1+C)}{3\tau}\right)\mathbb{E}[\|\boldsymbol{\theta}^{(t-1)} - \boldsymbol{\theta}^{(0)}\|] + \left(\frac{\eta\|F(\boldsymbol{\theta}^{(0)})\|}{6} + O(\eta\tilde{\epsilon})\right).$$

Setting $\Gamma = \frac{3\tau}{2\eta(1+C)}\left(\frac{\eta\|F(\boldsymbol{\theta}^{(0)})\|}{6} + O(\eta\tilde{\epsilon})\right)$, we can write

$$\mathbb{E}[\|\boldsymbol{\theta}^{(t)} - \boldsymbol{\theta}^{(0)}\| + \Gamma] \leq \left(1 + \frac{2\eta(1+C)}{3\tau}\right)\mathbb{E}[\|\boldsymbol{\theta}^{(t-1)} - \boldsymbol{\theta}^{(0)}\| + \Gamma],$$

Thus, with constant $\overline{T} > T_{\text{SGD}}$,

$$\begin{aligned}
\mathbb{E}[\|\boldsymbol{\theta}^{(T_{\text{SGD}})} - \boldsymbol{\theta}^{(0)}\| + \Gamma] &\leq \left(1 + \frac{2\eta(1+C)}{3\tau}\right)^{T_{\text{SGD}}}\Gamma \\
&\leq \left(1 + \frac{2\eta(1+C)}{3\tau}T_{\text{SGD}}\right)\Gamma + \overline{T}\,O(\eta^2\Gamma).
\end{aligned}$$

By Taylor expansion of $F_1$ near $\epsilon \approx 0$:

$$F_1(\epsilon, \epsilon, \epsilon, \epsilon) = \frac{2(e^{2/\tau} - 1)}{\tau(e^{2/\tau} + 1)}\epsilon + O(\epsilon^3), \quad \|F(\boldsymbol{\theta}^0)\| = \frac{4(e^{2/\tau} - 1)}{\tau(e^{2/\tau} + 1)}\epsilon + O(\epsilon^3),$$

we get

$$\Gamma = \frac{e^{2/\tau} - 1}{(1 + C)(e^{2/\tau} + 1)}\epsilon + O(\epsilon^3 + \tilde{\epsilon}) = O(\epsilon + \tilde{\epsilon}).$$

Since $\mathbb{E}[\|\boldsymbol{\theta}^{(T_{\mathrm{SGD}})} - \boldsymbol{\theta}^{(0)}\|] \geq 2(\frac{\pi}{4} - \rho - \epsilon)$,

$$\frac{2\eta(1 + C)\Gamma}{3\tau}T_{\mathrm{SGD}} \geq \mathbb{E}[\|\boldsymbol{\theta}^{(T_{\mathrm{SGD}})} - \boldsymbol{\theta}^{(0)}\|] + O(\eta^2(\epsilon + \tilde{\epsilon}))$$

$$\geq 2(\frac{\pi}{4} - \rho - \epsilon) + O(\eta^2(\epsilon + \tilde{\epsilon})).$$

Therefore,

$$T_{\mathrm{SGD}} \geq \frac{3(e^{2/\tau} + 1)}{e^{2/\tau} - 1}\tau \cdot \frac{\pi/4 - \rho - \epsilon + O(\eta^2(\epsilon + \tilde{\epsilon}))}{\eta\epsilon + O(\eta(\epsilon^3 + \tilde{\epsilon}))}.$$

$\square$

**Remark 2.** To simply compare the convergence rates of two algorithms, we assumed that there is some constant $\overline{T}$ such that $T_{\mathrm{SGD}}, T_{\mathrm{OSGD}} < \overline{T}$ in Theorem 8. However, without this assumption, we could still obtain lower bounds of both algorithms at $\tau = 1$ as;

$$T_{\mathrm{OSGD}} \geq \frac{2}{\log(1 + 2\eta)}\log\left[\frac{\frac{\pi}{4} - \rho + O(\epsilon^3)}{\epsilon + O(\epsilon^3)}\right],$$

$$T_{\mathrm{SGD}} \geq \frac{1}{\log\left(1 + \frac{2(1+C)}{3}\eta\right)}\log\left[\frac{1}{\tilde{C}}\frac{\frac{\pi}{4} - \rho - (1 - \tilde{C})\epsilon + O(\epsilon^3 + \tilde{\epsilon})}{\epsilon + O(\epsilon^3 + \tilde{\epsilon})}\right],$$

where $\tilde{C} = (e^2 - 1)/2(C + 1)(e^2 + 1)$, $C := \max_{x \in [0, \frac{\pi}{2}]}[2\sin x/(1 + e^{1+\cos x})]'$, and their approximations are $\tilde{C} \approx 0.265, C \approx 0.436$. For small enough $\eta, \epsilon, \tilde{\epsilon}$, we can observe OSGD algorithm converges faster than SGD algorithm, if the inequalities are tight.

**Direct Application of OSGD and its Convergence**    We now focus exclusively on the convergence of OSGD. We prove Theorem 7, which establishes the convergence of an application of OSGD to the mini-batch contrastive learning problem, with respect to the loss function $\widetilde{\mathcal{L}}^{\mathrm{con}}$.

---

**Algorithm 2:** The direct application of OSGD to our problem

---

1: **Parameters:**    $k$: the number of batches to be randomly chosen at each iteration; $q$: the number of batches of the largest losses to be chosen among $k$ batches at each iteration; $T$: the number of iterations.

2: **Inputs:** an initial feature vector $(\boldsymbol{U}^{(0)}, \boldsymbol{V}^{(0)})$, the set of learning rates $\{\eta_t\}_{t=0}^{T-1}$.

3: **for** $t = 0$ **to** $T - 1$ **do**

    Randomly choose $S \subset [\binom{N}{B}]$ with $|S| = k$

    Choose $i_1, \ldots, i_q \in S$ having the largest losses, i.e., $\mathcal{L}^{\mathrm{con}}(\boldsymbol{U}_{\mathcal{B}_i}^{(t)}, \boldsymbol{V}_{\mathcal{B}_i}^{(t)})$

    $g \leftarrow \frac{1}{q}\sum_{i \in S}\nabla_{\boldsymbol{U}, \boldsymbol{V}}\mathcal{L}^{\mathrm{con}}(\boldsymbol{U}_{\mathcal{B}_i}^{(t)}, \boldsymbol{V}_{\mathcal{B}_i}^{(t)})$

    $(\boldsymbol{U}^{(t+1)}, \boldsymbol{V}^{(t+1)}) \leftarrow (\boldsymbol{U}^{(t)}, \boldsymbol{V}^{(t)}) - \eta_t g$

    $(\boldsymbol{U}^{(t+1)}, \boldsymbol{V}^{(t+1)}) \leftarrow \mathrm{normalize}(\boldsymbol{U}^{(t+1)}, \boldsymbol{V}^{(t+1)})$

---

For ease of reference, we repeat the following definition:

$$\widetilde{\mathcal{L}}^{\mathrm{con}}(\boldsymbol{U}, \boldsymbol{V}) := \frac{1}{q} \sum_{j=1}^{\binom{N}{B}} \gamma_j \mathcal{L}^{\mathrm{con}}(\boldsymbol{U}_{\mathcal{B}_{(j)}}, \boldsymbol{V}_{\mathcal{B}_{(j)}}), \quad \gamma_j = \frac{\sum_{l=0}^{q-1} \binom{j-1}{l}\binom{\binom{N}{B}-j}{k-l-1}}{\binom{\binom{N}{B}}{k}}, \tag{32}$$

where $\mathcal{B}_{(j)}$ represents the batch with the $j$-th largest loss among all possible $\binom{N}{B}$ batches, and $q$, $k$ are parameters for the OSGD.

**Theorem 7** (Convergence results)**.** *Consider sampling $t^\star$ from $[T]$ with probability proportional to $\{\eta_t\}_{t=0}^T$, that is, $\mathbb{P}(t^\star = t) = \eta_t / (\sum_{i=0}^T \eta_i)$. Then $\forall \rho > \rho_0 = \sqrt{8/B\tau^2} + 4e^{2/\tau}/B\tau^2$,*

$$\mathbb{E}\left[\left\|\nabla\widetilde{\mathcal{L}}^{\mathrm{con}}(\boldsymbol{U}^{(t^\star)}, \boldsymbol{V}^{(t^\star)})\right\|^2\right] \le \frac{(\rho + \rho_0)^2}{\rho(\rho - \rho_0)} \frac{\left(\widetilde{\mathcal{L}}^{\mathrm{con}}(\boldsymbol{U}^{(0)}, \boldsymbol{V}^{(0)}) - \widetilde{\mathcal{L}}^{\mathrm{con}\star}\right) + 8\rho\sum_{t=0}^T \eta_t^2}{\sum_{t=0}^T \eta_t},$$

*where $\widetilde{\mathcal{L}}^{\mathrm{con}\star}$ denotes the minimized value of $\widetilde{\mathcal{L}}^{\mathrm{con}}$.*

*Proof.* Define $(\widehat{\boldsymbol{U}}^{(t^\star)}, \widehat{\boldsymbol{V}}^{(t^\star)}) = \underset{\boldsymbol{U}', \boldsymbol{V}'}{\mathrm{argmin}} \left\{ \widetilde{\mathcal{L}}^{\mathrm{con}}(\boldsymbol{U}', \boldsymbol{V}') + \frac{\rho}{2}\|(\boldsymbol{U}', \boldsymbol{V}') - (\boldsymbol{U}^{(t^\star)}, \boldsymbol{V}^{(t^\star)})\|^2 \right\}$. We begin by referring to Lemma 2.2. in Davis & Drusvyatskiy (2019), which provides the following equations:

$$\|(\boldsymbol{U}^{(t^\star)}, \boldsymbol{V}^{(t^\star)}) - (\widehat{\boldsymbol{U}}^{(t^\star)}, \widehat{\boldsymbol{V}}^{(t^\star)})\| = \frac{1}{\rho}\|\nabla\widetilde{\mathcal{L}}_\rho^{\mathrm{con}}(\boldsymbol{U}^{(t^\star)}, \boldsymbol{V}^{(t^\star)})\|,$$

$$\|\nabla\widetilde{\mathcal{L}}^{\mathrm{con}}(\widehat{\boldsymbol{U}}^{(t^\star)}, \widehat{\boldsymbol{V}}^{(t^\star)})\| \le \|\nabla\widetilde{\mathcal{L}}_\rho^{\mathrm{con}}(\boldsymbol{U}^{(t^\star)}, \boldsymbol{V}^{(t^\star)})\|.$$

Furthermore, we have that $\nabla\widetilde{\mathcal{L}}^{\mathrm{con}}$ is $\rho_0$-Lipschitz in $((B_d(0,1))^N)^2$ by Theorem 11. This gives

$$\|\nabla\widetilde{\mathcal{L}}^{\mathrm{con}}(\boldsymbol{U}^{(t^\star)}, \boldsymbol{V}^{(t^\star)}) - \nabla\widetilde{\mathcal{L}}^{\mathrm{con}}(\widehat{\boldsymbol{U}}^{(t^\star)}, \widehat{\boldsymbol{V}}^{(t^\star)})\| \le \rho_0\|(\boldsymbol{U}^{(t^\star)}, \boldsymbol{V}^{(t^\star)}) - (\widehat{\boldsymbol{U}}^{(t^\star)}, \widehat{\boldsymbol{V}}^{(t^\star)})\|$$

Therefore,

$$\begin{aligned}
\|\nabla\widetilde{\mathcal{L}}^{\mathrm{con}}(\boldsymbol{U}^{(t^\star)}, \boldsymbol{V}^{(t^\star)})\| &\le \|\nabla\widetilde{\mathcal{L}}^{\mathrm{con}}(\widehat{\boldsymbol{U}}^{(t^\star)}, \widehat{\boldsymbol{V}}^{(t^\star)})\| + \|\nabla\widetilde{\mathcal{L}}^{\mathrm{con}}(\boldsymbol{U}^{(t^\star)}, \boldsymbol{V}^{(t^\star)}) - \nabla\widetilde{\mathcal{L}}^{\mathrm{con}}(\widehat{\boldsymbol{U}}^{(t^\star)}, \widehat{\boldsymbol{V}}^{(t^\star)})\| \\
&\le \|\nabla\widetilde{\mathcal{L}}^{\mathrm{con}}(\widehat{\boldsymbol{U}}^{(t^\star)}, \widehat{\boldsymbol{V}}^{(t^\star)})\| + \rho_0\|(\boldsymbol{U}^{(t^\star)}, \boldsymbol{V}^{(t^\star)}) - (\widehat{\boldsymbol{U}}^{(t^\star)}, \widehat{\boldsymbol{V}}^{(t^\star)})\| \\
&\le \frac{\rho + \rho_0}{\rho}\|\nabla\widetilde{\mathcal{L}}_\rho^{\mathrm{con}}(\boldsymbol{U}^{(t^\star)}, \boldsymbol{V}^{(t^\star)})\|.
\end{aligned}$$

As a consequence of Thm 9,

$$\begin{aligned}
\mathbb{E}\left[\left\|\nabla\widetilde{\mathcal{L}}^{\mathrm{con}}(\boldsymbol{U}^{(t^\star)}, \boldsymbol{V}^{(t^\star)})\right\|^2\right] &\le \frac{(\rho + \rho_0)^2}{\rho^2}\mathbb{E}\left[\left\|\nabla\widetilde{\mathcal{L}}_\rho^{\mathrm{con}}(\boldsymbol{U}^{(t^\star)}, \boldsymbol{V}^{(t^\star)})\right\|^2\right] \\
&\le \frac{(\rho + \rho_0)^2}{\rho(\rho - \rho_0)} \frac{\left(\widetilde{\mathcal{L}}_\rho^{\mathrm{con}}(\boldsymbol{U}^{(0)}, \boldsymbol{V}^{(0)}) - \widetilde{\mathcal{L}}_\rho^{\mathrm{con}\star}\right) + 8\rho\sum_{t=0}^T \eta_t^2}{\sum_{t=0}^T \eta_t} \\
&\le \frac{(\rho + \rho_0)^2}{\rho(\rho - \rho_0)} \frac{\left(\widetilde{\mathcal{L}}^{\mathrm{con}}(\boldsymbol{U}^{(0)}, \boldsymbol{V}^{(0)}) - \widetilde{\mathcal{L}}^{\mathrm{con}\star}\right) + 8\rho\sum_{t=0}^T \eta_t^2}{\sum_{t=0}^T \eta_t}.
\end{aligned}$$

Note that $\widetilde{\mathcal{L}}_\rho^{\mathrm{con}\star}$ is the minimized value of $\widetilde{\mathcal{L}}_\rho^{\mathrm{con}}$, and the last inequality is due to $\widetilde{\mathcal{L}}_\rho^{\mathrm{con}}(\boldsymbol{U}^{(0)}, \boldsymbol{V}^{(0)}) - \widetilde{\mathcal{L}}_\rho^{\mathrm{con}\star} \le \widetilde{\mathcal{L}}^{\mathrm{con}}(\boldsymbol{U}^{(0)}, \boldsymbol{V}^{(0)}) - \widetilde{\mathcal{L}}^{\mathrm{con}\star}$, because

$$\begin{aligned}
\widetilde{\mathcal{L}}_\rho^{\mathrm{con}}(\boldsymbol{U}^{(0)}, \boldsymbol{V}^{(0)}) &= \min_{\boldsymbol{U}', \boldsymbol{V}'}\left\{\widetilde{\mathcal{L}}^{\mathrm{con}}(\boldsymbol{U}', \boldsymbol{V}') + \frac{\rho}{2}\|(\boldsymbol{U}', \boldsymbol{V}') - (\boldsymbol{U}^{(0)}, \boldsymbol{V}^{(0)})\|^2\right\} \\
&\le \widetilde{\mathcal{L}}^{\mathrm{con}}(\boldsymbol{U}^{(0)}, \boldsymbol{V}^{(0)})
\end{aligned}$$

by putting $(\boldsymbol{U}', \boldsymbol{V}') = (\boldsymbol{U}^{(0)}, \boldsymbol{V}^{(0)})$, and

$$\begin{aligned}
\widetilde{\mathcal{L}}^{\mathrm{con}\,\star} &= \min_{\boldsymbol{U}',\boldsymbol{V}'} \left\{ \widetilde{\mathcal{L}}^{\mathrm{con}}(\boldsymbol{U}', \boldsymbol{V}') \right\} \\
&\leq \min_{\boldsymbol{U}',\boldsymbol{V}'} \left\{ \widetilde{\mathcal{L}}^{\mathrm{con}}(\boldsymbol{U}', \boldsymbol{V}') + \frac{\rho}{2} \|(\boldsymbol{U}', \boldsymbol{V}') - (\boldsymbol{U}, \boldsymbol{V})\|^2 \right\} \\
&= \widetilde{\mathcal{L}}_\rho^{\mathrm{con}}(\boldsymbol{U}, \boldsymbol{V})
\end{aligned}$$

for any $\boldsymbol{U}$, $\boldsymbol{V}$, implying that $\widetilde{\mathcal{L}}^{\mathrm{con}\,\star} \leq \widetilde{\mathcal{L}}_\rho^{\mathrm{con}\,\star}$. □

We provide details, including proof of theorems and lemmas in the sequel.

**Theorem 9.** *Consider sampling $t^\star$ from $[T]$ with probability $\mathbb{P}(t^\star = t) = \eta_t / (\sum_{i=0}^T \eta_i)$. Then $\forall \rho > \rho_0 = 2\sqrt{2/B\tau^2} + 4e^{2/\tau}/B\tau^2$, we have*

$$\mathbb{E}\left[ \left\| \nabla \widetilde{\mathcal{L}}_\rho^{\mathrm{con}}(\boldsymbol{U}^{(t^\star)}, \boldsymbol{V}^{(t^\star)}) \right\|^2 \right] \leq \frac{\rho}{\rho - \rho_0} \frac{\left( \widetilde{\mathcal{L}}_\rho^{\mathrm{con}}(\boldsymbol{U}^{(0)}, \boldsymbol{V}^{(0)}) - \widetilde{\mathcal{L}}_\rho^{\mathrm{con}\,\star} \right) + 8\rho\tau^{-2} \sum_{t=0}^T \eta_t^2}{\sum_{t=0}^T \eta_t},$$

*where $\widetilde{\mathcal{L}}_\rho^{\mathrm{con}}(\boldsymbol{U}, \boldsymbol{V}) \coloneqq \min_{\boldsymbol{U}',\boldsymbol{V}'} \left\{ \widetilde{\mathcal{L}}^{\mathrm{con}}(\boldsymbol{U}', \boldsymbol{V}') + \frac{\rho}{2} \|(\boldsymbol{U}', \boldsymbol{V}') - (\boldsymbol{U}, \boldsymbol{V})\|^2 \right\}$, and $\widetilde{\mathcal{L}}_\rho^{\mathrm{con}\,\star}$ denotes the minimized value of $\widetilde{\mathcal{L}}_\rho^{\mathrm{con}}$.*

*Proof.* $\nabla \widetilde{\mathcal{L}}^{\mathrm{con}}$ is $\rho_0$-Lipschitz in $((B_d(0,1))^N)^2$ by Theorem 11. Hence, it is $\rho_0$-weakly convex by Lemma 5. Furthermore, the gradient norm of a mini-batch loss, or $\|\nabla_{\boldsymbol{U},\boldsymbol{V}} \mathcal{L}^{\mathrm{con}}(\boldsymbol{U}_{\mathcal{B}_i}, \boldsymbol{V}_{\mathcal{B}_i})\|$ is bounded by $L = 4/\tau$. Finally, (Kawaguchi & Lu, 2020, Theorem 1) states that the expected value of gradients of the OSGD algorithm is $\nabla_{\boldsymbol{U},\boldsymbol{V}} \widetilde{\mathcal{L}}^{\mathrm{con}}(\boldsymbol{U}^{(t)}, \boldsymbol{V}^{(t)})$ at each iteration $t$. Therefore, we can apply (Davis & Drusvyatskiy, 2019, Theorem 3.1) to the OSGD algorithm to obtain the desired result. □

Roughly speaking, Theorem 7 shows that $(\boldsymbol{U}^{(t^\star)}, \boldsymbol{V}^{(t^\star)})$ are close to a stationary point of $\widetilde{\mathcal{L}}_\rho^{\mathrm{con}}$. We refer readers to Davis & Drusvyatskiy (2019) which illustrates the role of the norm of the gradient of the Moreau envelope, $\|\nabla \widetilde{\mathcal{L}}_\rho^{\mathrm{con}}(\boldsymbol{U}^{(t^\star)}, \boldsymbol{V}^{(t^\star)})\|$, being small in the context of stochastic optimization.

We leave the results of some auxiliary theorems and lemmas to Subsection C.3.

### C.3 Auxiliaries for the Proof of Theorem 7

For a square matrix $A$, we denote its trace by $\mathrm{tr}(A)$. If matrices $A$ and $C$ are of the same shape, we define the canonical inner product $\langle A, C \rangle$ by

$$\langle A, C \rangle = \sum_{i,j} A_{ij} C_{ij} = \mathrm{tr}(A^\mathsf{T} C).$$

Following a pythonic notation, we write $A_{i,:}$ and $A_{:,j}$ for the $i$-th row and $j$-th column of a matrix $A$, respectively. The Cauchy–Schwarz inequality for matrices is given by

$$\langle A, C \rangle \leq \|A\| \|C\|,$$

where a norm $\|\cdot\|$ is a Frobenius norm in matrix i.e. $\|A\| = \left( \sum_{i,j} A_{ij}^2 \right)^{1/2}$.

**Lemma 4.** *Let $A \in \mathbb{R}^{m \times n}$, $C \in \mathbb{R}^{n \times k}$. Then, $\|AC\| \leq \|A\| \|C\|$.*

*Proof.* By a basic calculation, we have

$$\|AC\|^2 = \mathrm{tr}(C^\mathsf{T} A^\mathsf{T} A C) = \mathrm{tr}(CC^\mathsf{T} A^\mathsf{T} A) = \langle CC^\mathsf{T}, A^\mathsf{T} A \rangle \leq \|CC^\mathsf{T}\| \|A^\mathsf{T} A\|.$$

Meanwhile, for any positive semidefinite matrix $D$, let $D = U\Lambda U^\mathsf{T}$ be a spectral decomposition of $D$. Then, we have

$$\mathrm{tr}(D^2) = \mathrm{tr}(U\Lambda^2 U^\mathsf{T}) = \mathrm{tr}(\Lambda^2 U^\mathsf{T}U) = \mathrm{tr}(\Lambda^2) \leq (\mathrm{tr}(\Lambda))^2 = (\mathrm{tr}(D))^2,$$

where $\lambda_i(D)$ denotes the $i$-th eigenvalue of a matrix $D$. Invoking this fact, we have

$$\|CC^\mathsf{T}\|^2 = \mathrm{tr}((CC^\mathsf{T})^2) \leq (\mathrm{tr}(CC^\mathsf{T}))^2 = \|C\|^4,$$

or equivalently, $\|CC^\mathsf{T}\| \leq \|C\|^2$. Similarly, we have $\|A^\mathsf{T}A\| = \|A\|^2$. Therefore, we obtain

$$\|AC\|^2 \leq \|CC^\mathsf{T}\|\|A^\mathsf{T}A\| \leq \|A\|^2\|C\|^2,$$

which means $\|AC\| \leq \|A\|\|C\|$. $\qquad\square$

If $\mathcal{L}\colon \mathbb{R}^{m\times n} \to \mathbb{R}$ is a function of a matrix $X \in \mathbb{R}^{m\times n}$, we write a gradient of $\mathcal{L}$ with respect to $X$ as a matrix-valued function defined by

$$(\nabla_X \mathcal{L})_{ij} = \left(\frac{\partial \mathcal{L}}{\partial X}\right)_{ij} = \frac{\partial \mathcal{L}}{\partial X_{ij}}.$$

Then, the chain rule gives

$$\frac{d}{dt}\mathcal{L}(X) = \left\langle \frac{dX}{dt}, \nabla_X \mathcal{L} \right\rangle$$

for a scalar variable $t$. If $\mathcal{L}(\boldsymbol{U}, \boldsymbol{V})$ is a function of two matrices $\boldsymbol{U}, \boldsymbol{V} \in \mathbb{R}^{m\times n}$, we define $\nabla_{\boldsymbol{U},\boldsymbol{V}}\mathcal{L}$ as a horizontal stack of two gradient matrices, i.e., $\nabla_{\boldsymbol{U},\boldsymbol{V}}\mathcal{L} = (\nabla_{\boldsymbol{U}}\mathcal{L}, \nabla_{\boldsymbol{V}}\mathcal{L})$.

Now, we briefly review some necessary facts about Lipschitz functions.

**Lemma 5** (Rendering of weak convexity by a Lipschitz gradient). *Let $f\colon \mathbb{R}^d \to \mathbb{R}$ be a $\rho$-smooth function, i.e., $\nabla f$ is a $\rho$-Lipschitz function. Then, $f$ is $\rho$-weakly convex.*

*Proof.* For the sake of simplicity, assume $f$ is twice differentiable. We claim that $\nabla^2 f \succeq -\rho\mathbb{I}_d$, where $\mathbb{I}_d$ is the $d \times d$ identity matrix and $A \succeq B$ means $A - B$ is a positive semidefinite matrix. It is clear that this claim renders $f + \frac{\rho}{2}\|\cdot\|^2$ to be convex.

Let us assume, contrary to our claim, that there exists $\boldsymbol{x}_0 \in \mathbb{R}^d$ with $\nabla^2 f(\boldsymbol{x}_0) \not\succeq -\rho\mathbb{I}_d$. Therefore, $\nabla^2 f(\boldsymbol{x}_0)$ has an eigenvalue $\lambda < -\rho$. Denote corresponding eigenvector by $\boldsymbol{u}$, so we have $\nabla^2 f(\boldsymbol{x}_0)\boldsymbol{u} = \lambda\boldsymbol{u}$, and consider $g(\epsilon) = \nabla f(\boldsymbol{x}_0 + \epsilon\boldsymbol{u})$; the (elementwise) Taylor expansion of $g$ at $\epsilon = 0$ gives

$$\nabla f(\boldsymbol{x}_0 + \epsilon\boldsymbol{u}) = \nabla f(\boldsymbol{x}_0) + \epsilon\nabla^2 f(\boldsymbol{x}_0)\boldsymbol{u} + o(\epsilon),$$

which gives

$$\frac{\|\nabla f(\boldsymbol{x}_0 + \epsilon\boldsymbol{u}) - \nabla f(\boldsymbol{x}_0)\|}{\epsilon} = \left\|\nabla^2 f(\boldsymbol{x}_0)\boldsymbol{u} + \frac{o(\epsilon)}{\epsilon}\right\|.$$

Taking $\epsilon \to 0$, we obtain $\|\nabla f(\boldsymbol{x}_0 + \epsilon\boldsymbol{u}) - \nabla f(\boldsymbol{x}_0)\|/\epsilon \geq |\lambda| > \rho$, which is contradictory to $\rho$-Lipschitzness of $\nabla f$. $\qquad\square$

For $X \in \mathbb{R}^{B\times B}$, let us define

$$\mathcal{L}^M(X) = \frac{1}{B}\left(-2\mathrm{tr}(X)/\tau + \sum_{i=1}^{B}\log\sum_{j=1}^{B}\exp(X_{ij}/\tau) + \sum_{i=1}^{B}\log\sum_{j=1}^{B}\exp(X_{ji}/\tau)\right).$$

Using this function, we can write the loss corresponding to a mini-batch $\mathcal{B}$ of size $B$ by

$$\mathcal{L}^M(\boldsymbol{U}_\mathcal{B}^\mathsf{T}\boldsymbol{V}_\mathcal{B}) = \mathcal{L}^{\mathrm{con}}(\boldsymbol{U}_\mathcal{B}, \boldsymbol{V}_\mathcal{B}).$$

We now claim the following:

**Lemma 6.** *Consider $X \in \mathbb{R}^{B \times B}$, where $|X_{ij}| \leq 1$ for all $1 \leq i, j \leq B$. Then, $\nabla_X \mathcal{L}^M(X)$ is bounded by $\sqrt{8/B\tau^2}$ and $\frac{2e^{2/\tau}}{B^2\tau^2}$-Lipschitz.*

*Proof.* With basic calculus rules, we obtain

$$\tau B \nabla_X \mathcal{L}^M(X) = -2\mathbb{I}_B + P_X + Q_X, \tag{33}$$

where $\mathbb{I}_B$ is the $B \times B$ identity matrix and

$$(P_X)_{ij} = \exp(X_{ij}/\tau) / \sum_{k=1}^{B} \exp(X_{ik}/\tau), \quad (Q_X)_{ij} = \exp(X_{ij}/\tau) / \sum_{k=1}^{B} \exp(X_{kj}/\tau).$$

From $\sum_j P_{ij} = 1$ for all $i$, it is easy to see that $\|(\mathbb{I}_B - P)_{i,:}\|^2 \leq 2$. This gives $\|\mathbb{I}_B - P_X\|^2 \leq 2B$, and similarly $\|\mathbb{I}_B - Q_X\|^2 \leq 2B$. Therefore, we have

$$\|B\tau \nabla_X \mathcal{L}^M(X)\| \leq \|\mathbb{I}_B - P_X\| + \|\mathbb{I}_B - Q_X\| \leq \sqrt{8B}, \tag{34}$$

or equivalently

$$\|\nabla_X \mathcal{L}^M(X)\| \leq \sqrt{8/B\tau^2}. \tag{35}$$

We now show that $\nabla_X \mathcal{L}^M$ is $\frac{2e^{2/\tau}}{B^2\tau^2}$-Lipschitz. Define $p: \mathbb{R}^B \to \mathbb{R}^B$ by

$$(p(x))_i = \frac{\exp(x_i/\tau)}{\sum_{k=1}^{B} \exp(x_k/\tau)}.$$

Then, we have

$$\frac{\partial}{\partial x} p(x) = \frac{1}{\tau} [\mathrm{diag}(p(x)) - p(x)p(x)^{\mathsf{T}}].$$

For $x \in [-1, 1]^B$, we have $p(x)_i \leq \frac{e^{2/\tau}}{B-1+e^{2/\tau}} < \frac{e^{2/\tau}}{B}$ for any $i$. Thus,

$$0 \preceq \frac{\partial}{\partial x} p(x) \preceq \mathrm{diag}(p(x)) \preceq \frac{e^{2/\tau}}{B\tau} \mathbb{I}_B,$$

which means $p(x)$ is $\frac{e^{2/\tau}}{B\tau}$-Lipschitz, i.e., $\|p(x) - p(y)\| \leq \frac{e^{2/\tau}}{B\tau} \|x - y\|$ for any $x, y \in [-1, 1]^B$. Using this fact, we can bound $\|P_X - P_Y\|$ for $X, Y \in [-1, 1]^{B \times B}$ as follows:

$$\|P_X - P_Y\|^2 = \sum_{i=1}^{B} \|p(X_{i,:}) - p(Y_{i,:})\|^2 \leq \left(\frac{e^{2/\tau}}{B\tau}\right)^2 \sum_{i=1}^{B} \|X_{i,:} - Y_{i,:}\|^2 = \left(\frac{e^{2/\tau}}{B\tau}\right)^2 \|X - Y\|^2.$$

Similarly, we have $\|Q_X - Q_Y\| \leq \frac{e^{2/\tau}}{B\tau} \|X - Y\|$. Summing up,

$$\|B\tau \nabla_X \mathcal{L}^M(X) - B\nabla_X \mathcal{L}^M(Y)\| \leq \|P_X - P_Y\| + \|Q_X - Q_Y\| \leq \frac{2e^{2/\tau}}{B\tau} \|X - Y\|.$$

which renders

$$\|\nabla_X \mathcal{L}^M(X) - \nabla_X \mathcal{L}^M(Y)\| \leq \frac{2e^{2/\tau}}{B^2\tau^2} \|X - Y\|.$$

$\square$

Recall that $\mathcal{L}^{\mathrm{con}}(\boldsymbol{U}_{\mathcal{B}}, \boldsymbol{V}_{\mathcal{B}}) = \mathcal{L}^M(\boldsymbol{U}_{\mathcal{B}}^{\mathsf{T}} \boldsymbol{V}_{\mathcal{B}})$ for $\boldsymbol{U}_{\mathcal{B}}, \boldsymbol{V}_{\mathcal{B}} \in \mathbb{R}^{d \times B}$ (They correspond to embeddings corresponding to a mini-batch $\mathcal{B}$). Using this relation, we can calculate the gradient of $\mathcal{L}^{\mathrm{con}}$ with respect to $\boldsymbol{U}_{\mathcal{B}}$. Denote

$E_{ij} \in \mathbb{R}^{d \times B}$ a one-hot matrix, which is a matrix of zero entries except for $(i, j)$ indices being 1, and write $G = \nabla_X \mathcal{L}^M(\boldsymbol{U}_\mathcal{B}^\intercal \boldsymbol{V}_\mathcal{B})$. Then,

$$
\begin{aligned}
\frac{\partial}{\partial (\boldsymbol{U}_\mathcal{B})_{ij}} \mathcal{L}^{\mathrm{con}}(\boldsymbol{U}_\mathcal{B}, \boldsymbol{V}_\mathcal{B}) &= \left\langle \frac{\partial (\boldsymbol{U}_\mathcal{B}^\intercal \boldsymbol{V}_\mathcal{B})}{\partial \boldsymbol{U}_{\mathcal{B}ij}}, \nabla_X \mathcal{L}^M(\boldsymbol{U}_\mathcal{B}^\intercal \boldsymbol{V}_\mathcal{B}) \right\rangle \\
&= \left\langle E_{ij}^\intercal \boldsymbol{V}_\mathcal{B}, G \right\rangle \\
&= \mathrm{tr}\left( \boldsymbol{V}_\mathcal{B}^\intercal E_{ij} G \right) \\
&= \mathrm{tr}\left( E_{ij}(G \boldsymbol{V}_\mathcal{B}^\intercal) \right) \\
&= (G \boldsymbol{V}_\mathcal{B}^\intercal)_{ji} \\
&= (\boldsymbol{V}_\mathcal{B} G^\intercal)_{ij}.
\end{aligned}
$$

This elementwise relation means

$$
\frac{\partial}{\partial \boldsymbol{U}_\mathcal{B}} \mathcal{L}^{\mathrm{con}}(\boldsymbol{U}_\mathcal{B}, \boldsymbol{V}_\mathcal{B}) = \boldsymbol{V}_\mathcal{B} G^\intercal = \boldsymbol{V}_\mathcal{B}(\nabla_X \mathcal{L}^M(\boldsymbol{U}_\mathcal{B}^\intercal \boldsymbol{V}_\mathcal{B}))^\intercal, \tag{36}
$$

and similarly,

$$
\frac{\partial}{\partial \boldsymbol{V}_\mathcal{B}} \mathcal{L}^{\mathrm{con}}(\boldsymbol{U}_\mathcal{B}, \boldsymbol{V}_\mathcal{B}) = \boldsymbol{U}_\mathcal{B} \nabla_X \mathcal{L}^M(\boldsymbol{U}_\mathcal{B}^\intercal \boldsymbol{V}_\mathcal{B}). \tag{37}
$$

We introduce a simple lemma for bounding the difference between two multiplication of matrices.

**Lemma 7.** *For $A_1$, $A_2 \in \mathbb{R}^{m \times n}$ and $B_1$, $B_2 \in \mathbb{R}^{n \times k}$, we have*

$$
\|A_1 B_1 - A_2 B_2\| \leq \|A_1 - A_2\|\|B_1\| + \|A_2\|\|B_1 - B_2\|.
$$

*Proof.* This follows from a direct calculation and Lemma 4

$$
\begin{aligned}
\|A_1 B_1 - A_2 B_2\| &= \|A_1 B_1 - A_1 B_2 + A_1 B_2 - A_2 B_2\| \\
&\leq \|A_1(B_1 - B_2)\| + \|(A_1 - A_2)B_2\| \\
&\leq \|A_1 - A_2\|\|B_1\| + \|A_2\|\|B_1 - B_2\|.
\end{aligned}
$$

$\square$

**Theorem 10.** *For any $\boldsymbol{U}$, $\boldsymbol{V} \in (B_d(0,1))^N$ and any batch $\mathcal{B}$ of size $B$, we have $\|\nabla_{\boldsymbol{U}, \boldsymbol{V}} \mathcal{L}^{\mathrm{con}}(\boldsymbol{U}_\mathcal{B}, \boldsymbol{V}_\mathcal{B})\| \leq \frac{4}{\tau}$.*

*Proof.* Suppose $\boldsymbol{U}_\mathcal{B}, \boldsymbol{V}_\mathcal{B} \in (B_d(0,1))^B$, we have

$$
\nabla_{\boldsymbol{U}_\mathcal{B}, \boldsymbol{V}_\mathcal{B}} \mathcal{L}^{\mathrm{con}}(\boldsymbol{U}_\mathcal{B}, \boldsymbol{V}_\mathcal{B}) = (\boldsymbol{V}_\mathcal{B}(\nabla_X \mathcal{L}^M(\boldsymbol{U}_\mathcal{B}^\intercal \boldsymbol{V}_\mathcal{B}))^\intercal, \boldsymbol{U}_\mathcal{B} \nabla_X \mathcal{L}^M(\boldsymbol{U}_\mathcal{B}^\intercal \boldsymbol{V}_\mathcal{B}))
$$

from Equation (36) and (37). By following the fact that $\|\boldsymbol{U}_\mathcal{B}\|, \|\boldsymbol{V}_\mathcal{B}\| \leq \sqrt{B}$ and $\nabla_X \mathcal{L}^M(X) \leq \sqrt{8/B\tau^2}$ (see Lemma 6), we get

$$
\|\boldsymbol{V}_\mathcal{B}(\nabla_X \mathcal{L}^M(\boldsymbol{U}_\mathcal{B}^\intercal \boldsymbol{V}_\mathcal{B}))^\intercal\| \leq \|\boldsymbol{V}_\mathcal{B}\|\|\nabla_X \mathcal{L}^M(\boldsymbol{U}_\mathcal{B}^\intercal \boldsymbol{V}_\mathcal{B})\| \leq \frac{\sqrt{8}}{\tau},
$$

and

$$
\|\boldsymbol{U}_\mathcal{B} \nabla_X \mathcal{L}^M(\boldsymbol{U}_\mathcal{B}^\intercal \boldsymbol{V}_\mathcal{B})\| \leq \|\boldsymbol{U}_\mathcal{B}\|\|\nabla_X \mathcal{L}^M(\boldsymbol{U}_\mathcal{B}^\intercal \boldsymbol{V}_\mathcal{B})\| \leq \frac{\sqrt{8}}{\tau}.
$$

Then,

$$
\|\nabla_{\boldsymbol{U}_\mathcal{B}, \boldsymbol{V}_\mathcal{B}} \mathcal{L}^{\mathrm{con}}(\boldsymbol{U}_\mathcal{B}, \boldsymbol{V}_\mathcal{B})\| = \sqrt{\|\boldsymbol{V}_\mathcal{B}(\nabla_X \mathcal{L}^M(\boldsymbol{U}_\mathcal{B}^\intercal \boldsymbol{V}_\mathcal{B}))^\intercal\|^2 + \|\boldsymbol{U}_\mathcal{B} \nabla_X \mathcal{L}^M(\boldsymbol{U}_\mathcal{B}^\intercal \boldsymbol{V}_\mathcal{B})\|^2} \leq \frac{4}{\tau}.
$$

Since $\mathcal{L}^{\mathrm{con}}(\boldsymbol{U}_{\mathcal{B}}, \boldsymbol{V}_{\mathcal{B}})$ is independent of $\boldsymbol{U}_{[N]\backslash\mathcal{B}}$ and $\boldsymbol{V}_{[N]\backslash\mathcal{B}}$, we have

$$\|\nabla_{\boldsymbol{U},\boldsymbol{V}}\mathcal{L}^{\mathrm{con}}(\boldsymbol{U}_{\mathcal{B}}, \boldsymbol{V}_{\mathcal{B}})\| = \|\nabla_{\boldsymbol{U}_{\mathcal{B}},\boldsymbol{V}_{\mathcal{B}}}\mathcal{L}^{\mathrm{con}}(\boldsymbol{U}_{\mathcal{B}}, \boldsymbol{V}_{\mathcal{B}})\| \le \frac{4}{\tau}.$$

$\square$

**Theorem 11.** $\nabla\widetilde{\mathcal{L}}^{\mathrm{con}}(\boldsymbol{U}, \boldsymbol{V})$ *is $\rho_0$-Lipschitz for $\boldsymbol{U}$, $\boldsymbol{V} \in (B_d(0,1))^N$, or to clarify,*

$$\|\nabla\widetilde{\mathcal{L}}^{\mathrm{con}}(\boldsymbol{U}^1, \boldsymbol{V}^1) - \nabla\widetilde{\mathcal{L}}^{\mathrm{con}}(\boldsymbol{U}^2, \boldsymbol{V}^2)\| \le \rho_0\|(\boldsymbol{U}^1, \boldsymbol{V}^1) - (\boldsymbol{U}^2, \boldsymbol{V}^2)\|$$

*for any $\boldsymbol{U}^1$, $\boldsymbol{V}^1$, $\boldsymbol{U}^2$, $\boldsymbol{V}^2 \in (B_d(0,1))^N$, where $\rho_0 = \sqrt{8/B\tau^2} + 4e^{2/\tau}/B\tau^2$.*

*Proof.* Denoting $\boldsymbol{U}_{\mathcal{B}}^i$, $\boldsymbol{V}_{\mathcal{B}}^i$ as parts of $\boldsymbol{U}^i$, $\boldsymbol{V}^i$ that correspond to a mini-batch $\mathcal{B}$, we first show $\|\nabla_{\boldsymbol{U}_{\mathcal{B}},\boldsymbol{V}_{\mathcal{B}}}\mathcal{L}^{\mathrm{con}}(\boldsymbol{U}_{\mathcal{B}}^1, \boldsymbol{V}_{\mathcal{B}}^1) - \nabla_{\boldsymbol{U}_{\mathcal{B}},\boldsymbol{V}_{\mathcal{B}}}\mathcal{L}^{\mathrm{con}}(\boldsymbol{U}_{\mathcal{B}}^2, \boldsymbol{V}_{\mathcal{B}}^2)\| \le \rho_0\|(\boldsymbol{U}_{\mathcal{B}}^1, \boldsymbol{V}_{\mathcal{B}}^1) - (\boldsymbol{U}_{\mathcal{B}}^2, \boldsymbol{V}_{\mathcal{B}}^2)\|$ holds. For any $\boldsymbol{U}_{\mathcal{B}}$, $\boldsymbol{V}_{\mathcal{B}} \in (B_d(0,1))^B$, we have

$$\nabla_{\boldsymbol{U}_{\mathcal{B}},\boldsymbol{V}_{\mathcal{B}}}\mathcal{L}^{\mathrm{con}}(\boldsymbol{U}_{\mathcal{B}}, \boldsymbol{V}_{\mathcal{B}}) = (\boldsymbol{V}_{\mathcal{B}}(\nabla_X\mathcal{L}^M(\boldsymbol{U}_{\mathcal{B}}^{\mathsf{T}}\boldsymbol{V}_{\mathcal{B}}))^{\mathsf{T}}, \boldsymbol{U}_{\mathcal{B}}\nabla_X\mathcal{L}^M(\boldsymbol{U}_{\mathcal{B}}^{\mathsf{T}}\boldsymbol{V}_{\mathcal{B}})).$$

from Equation 36 and Equation 37. Recall Lemma 6; for any $\boldsymbol{U}_{\mathcal{B}}^i$, $\boldsymbol{V}_{\mathcal{B}}^i \in (B_d(0,1))^B$ $(i = 1, 2)$, we have

$$\|\nabla_X\mathcal{L}^M((\boldsymbol{U}_{\mathcal{B}}^i)^{\mathsf{T}}\boldsymbol{V}_{\mathcal{B}}^i)\| \le \sqrt{8/B\tau^2}$$

and

$$\|\nabla_X\mathcal{L}^M((\boldsymbol{U}_{\mathcal{B}}^1)^{\mathsf{T}}\boldsymbol{V}_{\mathcal{B}}^1) - \nabla_X\mathcal{L}^M((\boldsymbol{U}_{\mathcal{B}}^2)^{\mathsf{T}}\boldsymbol{V}_{\mathcal{B}}^2)\| \le \frac{2e^{2/\tau}}{B^2\tau^2}\|(\boldsymbol{U}_{\mathcal{B}}^1)^{\mathsf{T}}\boldsymbol{V}_{\mathcal{B}}^1 - (\boldsymbol{U}_{\mathcal{B}}^2)^{\mathsf{T}}\boldsymbol{V}_{\mathcal{B}}^2\|.$$

We invoke Lemma 7 and obtain

$$\begin{aligned}
&\|\boldsymbol{U}_{\mathcal{B}}^1\nabla_X\mathcal{L}^M((\boldsymbol{U}_{\mathcal{B}}^1)^{\mathsf{T}}\boldsymbol{V}_{\mathcal{B}}^1) - \boldsymbol{U}_{\mathcal{B}}^2\nabla_X\mathcal{L}^M((\boldsymbol{U}_{\mathcal{B}}^2)^{\mathsf{T}}\boldsymbol{V}_{\mathcal{B}}^2)\| \\
&\le \|\boldsymbol{U}_{\mathcal{B}}^1 - \boldsymbol{U}_{\mathcal{B}}^2\|\|\nabla_X\mathcal{L}^M((\boldsymbol{U}_{\mathcal{B}}^1)^{\mathsf{T}}\boldsymbol{V}_{\mathcal{B}}^1)\| + \|\boldsymbol{U}_{\mathcal{B}}^2\|\|\nabla_X\mathcal{L}^M((\boldsymbol{U}_{\mathcal{B}}^1)^{\mathsf{T}}\boldsymbol{V}_{\mathcal{B}}^1) - \nabla_X\mathcal{L}^M((\boldsymbol{U}_{\mathcal{B}}^2)^{\mathsf{T}}\boldsymbol{V}_{\mathcal{B}}^2)\| \\
&\le \sqrt{8/B\tau^2}\|\boldsymbol{U}_{\mathcal{B}}^1 - \boldsymbol{U}_{\mathcal{B}}^2\| + \frac{2e^{2/\tau}}{B^{3/2}\tau^2}\|(\boldsymbol{U}_{\mathcal{B}}^1)^{\mathsf{T}}\boldsymbol{V}_{\mathcal{B}}^1 - (\boldsymbol{U}_{\mathcal{B}}^2)^{\mathsf{T}}\boldsymbol{V}_{\mathcal{B}}^2\| \\
&\le \sqrt{8/B\tau^2}\|\boldsymbol{U}_{\mathcal{B}}^1 - \boldsymbol{U}_{\mathcal{B}}^2\| + \frac{2e^{2/\tau}}{B^{3/2}\tau^2}(\|\boldsymbol{U}_{\mathcal{B}}^1 - \boldsymbol{U}_{\mathcal{B}}^2\|\|\boldsymbol{V}_{\mathcal{B}}^1\| + \|\boldsymbol{U}_{\mathcal{B}}^2\|\|\boldsymbol{V}_{\mathcal{B}}^1 - \boldsymbol{V}_{\mathcal{B}}^2\|) \\
&\le (\sqrt{8/B\tau^2} + 2e^{2/\tau}/B\tau^2)\|\boldsymbol{U}_{\mathcal{B}}^1 - \boldsymbol{U}_{\mathcal{B}}^2\| + (2e^{2/\tau}/B\tau^2)\|\boldsymbol{V}_{\mathcal{B}}^1 - \boldsymbol{V}_{\mathcal{B}}^2\|,
\end{aligned}$$

and similarly

$$\begin{aligned}
&\|\boldsymbol{V}_{\mathcal{B}}^1\nabla_X(\mathcal{L}^M((\boldsymbol{U}_{\mathcal{B}}^1)^{\mathsf{T}}\boldsymbol{V}_{\mathcal{B}}^1))^{\mathsf{T}} - \boldsymbol{V}_{\mathcal{B}}^2\nabla_X(\mathcal{L}^M((\boldsymbol{U}_{\mathcal{B}}^2)^{\mathsf{T}}\boldsymbol{V}_{\mathcal{B}}^2))^{\mathsf{T}}\| \\
&\le (2e^{2/\tau}/B\tau^2)\|\boldsymbol{U}_{\mathcal{B}}^1 - \boldsymbol{U}_{\mathcal{B}}^2\| + (\sqrt{8/B\tau^2} + 2e^{2/\tau}/B\tau^2)\|\boldsymbol{V}_{\mathcal{B}}^1 - \boldsymbol{V}_{\mathcal{B}}^2\|.
\end{aligned}$$

Using the fact that

$$(ax + by)^2 + (bx + ay)^2 = (a^2 + b^2)(x^2 + y^2) + 4abxy \le (a + b)^2(x^2 + y^2)$$

holds for any $a$, $b \ge 0$ and $x$, $y \in \mathbb{R}$, we obtain

$$\begin{aligned}
&\|\nabla\mathcal{L}^{\mathrm{con}}(\boldsymbol{U}_{\mathcal{B}}^1, \boldsymbol{V}_{\mathcal{B}}^1) - \nabla\mathcal{L}^{\mathrm{con}}(\boldsymbol{U}_{\mathcal{B}}^2, \boldsymbol{V}_{\mathcal{B}}^2)\|^2 \\
&= \|\boldsymbol{V}_{\mathcal{B}}^1\nabla_X(\mathcal{L}^M((\boldsymbol{U}_{\mathcal{B}}^1)^{\mathsf{T}}\boldsymbol{V}_{\mathcal{B}}^1))^{\mathsf{T}} - \boldsymbol{V}_{\mathcal{B}}^2\nabla_X(\mathcal{L}^M((\boldsymbol{U}_{\mathcal{B}}^2)^{\mathsf{T}}\boldsymbol{V}_{\mathcal{B}}^2))^{\mathsf{T}}\|^2 \\
&\quad + \|\boldsymbol{U}_{\mathcal{B}}^1\nabla_X\mathcal{L}^M((\boldsymbol{U}_{\mathcal{B}}^1)^{\mathsf{T}}\boldsymbol{V}_{\mathcal{B}}^1) - \boldsymbol{U}_{\mathcal{B}}^2\nabla_X\mathcal{L}^M((\boldsymbol{U}_{\mathcal{B}}^2)^{\mathsf{T}}\boldsymbol{V}_{\mathcal{B}}^2)\|^2 \\
&\le (\sqrt{8/B\tau^2} + 4e^{2/\tau}/B\tau^2)^2(\|\boldsymbol{U}_{\mathcal{B}}^1 - \boldsymbol{U}_{\mathcal{B}}^2\|^2 + \|\boldsymbol{V}_{\mathcal{B}}^1 - \boldsymbol{V}_{\mathcal{B}}^2\|^2) \\
&= (\sqrt{8/B\tau^2} + 4e^{2/\tau}/B\tau^2)^2\|(\boldsymbol{U}_{\mathcal{B}}^1, \boldsymbol{V}_{\mathcal{B}}^1) - (\boldsymbol{U}_{\mathcal{B}}^2, \boldsymbol{V}_{\mathcal{B}}^2)\|^2.
\end{aligned}$$

Restating this with $\rho_0 = \sqrt{8/B\tau^2 + 4e^{2/\tau}/B\tau^2}$, we have

$$\|\nabla\mathcal{L}^{\mathrm{con}}(\boldsymbol{U}_{\mathcal{B}}^1, \boldsymbol{V}_{\mathcal{B}}^1) - \nabla\mathcal{L}^{\mathrm{con}}(\boldsymbol{U}_{\mathcal{B}}^2, \boldsymbol{V}_{\mathcal{B}}^2)\| \leq \rho_0\|(\boldsymbol{U}_{\mathcal{B}}^1, \boldsymbol{V}_{\mathcal{B}}^1) - (\boldsymbol{U}_{\mathcal{B}}^2, \boldsymbol{V}_{\mathcal{B}}^2)\|. \tag{38}$$

Recall the definition of $\widetilde{\mathcal{L}}^{\mathrm{con}}$:

$$\widetilde{\mathcal{L}}^{\mathrm{con}}(\boldsymbol{U}, \boldsymbol{V}) = \frac{1}{q}\sum_j \gamma_j \mathcal{L}^{\mathrm{con}}(\boldsymbol{U}_{\mathcal{B}_{(j)}}, \boldsymbol{V}_{\mathcal{B}_{(j)}}),$$

where $\gamma_j = \frac{\sum_{l=0}^{q-1}\binom{j-1}{l}\binom{\binom{N}{B}-j}{k-l-1}}{\binom{\binom{N}{B}}{k}}$ and $\sum_j \gamma_j = q$. For any $\boldsymbol{U}, \boldsymbol{V} \in (\mathbb{S}^d)^N$, we can find a neighborhood of $(\boldsymbol{U}, \boldsymbol{V})$
so that value rank of $\mathcal{L}^{\mathrm{con}}(\boldsymbol{U}_{\mathcal{B}_i}, \boldsymbol{V}_{\mathcal{B}_i})$ over $i \in \{1, \ldots, \binom{N}{B}\}$ does not change, since $\mathcal{L}^{\mathrm{con}}$ is $\rho_0$-Lipschitz. More
precisely speaking, we can find a rank that can be accepted by all points in the neighborhood. Therefore,
we have

$$\nabla_{\boldsymbol{U}, \boldsymbol{V}}\widetilde{\mathcal{L}}^{\mathrm{con}}(\boldsymbol{U}, \boldsymbol{V}) = \frac{1}{q}\sum_j \gamma_j \nabla_{\boldsymbol{U}, \boldsymbol{V}}\mathcal{L}^{\mathrm{con}}(\boldsymbol{U}_{\mathcal{B}_{(j)}}, \boldsymbol{V}_{\mathcal{B}_{(j)}}),$$

and since $\|\boldsymbol{U}_{\mathcal{B}_{(j)}} - \boldsymbol{V}_{\mathcal{B}_{(j)}}\| \leq \|\boldsymbol{U} - \boldsymbol{V}\|$, $\nabla_{\boldsymbol{U}, \boldsymbol{V}}\widetilde{\mathcal{L}}^{\mathrm{con}}(\boldsymbol{U}, \boldsymbol{V})$ is locally $\rho_0$-Lipschitz. Since $\mathcal{L}^{\mathrm{con}}$ is smooth,
such property is equivalent to $-\rho_0\mathbb{I}_N \preceq \nabla_{\boldsymbol{U}, \boldsymbol{V}}^2\widetilde{\mathcal{L}}^{\mathrm{con}}(\boldsymbol{U}, \boldsymbol{V}) \preceq \rho_0\mathbb{I}_N$, where $\mathbb{I}_N$ is the $N \times N$ identity matrix.
Therefore, $\widetilde{\mathcal{L}}^{\mathrm{con}}$ is $\rho_0$-Lipschitz on $((B_d(0,1))^N)^2$. $\qquad\square$

# D  Algorithm Details

## D.1  Stochastic Gradient Descent (SGD)

We consider two SGD algorithms:

1. SGD *with replacement* (Algorithm 3) with $k = 1$ for the theoretical analysis in Section 5.1.

2. SGD *without replacement* (Algorithm 4) for experimental results in Section 6, which is widely employed
   in practical settings.

In the more practical setting where $\boldsymbol{u}_i = f_\theta(\boldsymbol{x}_i)$ and $\boldsymbol{v}_i = g_\phi(\boldsymbol{y}_i)$, SGD updates the model parameters $\theta, \phi$ using the gradients $\frac{1}{k}\sum_{i \in S_{\mathcal{B}}}\nabla_{\theta, \phi}\mathcal{L}^{\mathrm{con}}(\boldsymbol{U}_{\mathcal{B}_i}, \boldsymbol{V}_{\mathcal{B}_i})$ instead of explicitly updating $\boldsymbol{U}$ and $\boldsymbol{V}$.

---

**Algorithm 3:** SGD with replacement

**Input:** the number of positive pairs $N$, mini-batch size $B$, the number of mini-batches $k$, the number of iterations $T$, the learning rate $\eta$, initial embedding matrices: $\boldsymbol{U}, \boldsymbol{V}$

1 **for** $t = 1$ **to** $T$ **do**
2 $\quad$ Randomly select $k$ mini-batch indices $S_{\mathcal{B}} \subset [\binom{N}{B}]$ ($|S_{\mathcal{B}}| = k$)
3 $\quad$ Compute the gradient: $g \leftarrow \frac{1}{k}\sum_{i \in S_{\mathcal{B}}}\nabla_{\boldsymbol{U}, \boldsymbol{V}}\mathcal{L}^{\mathrm{con}}(\boldsymbol{U}_{\mathcal{B}_i}, \boldsymbol{V}_{\mathcal{B}_i})$
4 $\quad$ Update the weights: $(\boldsymbol{U}, \boldsymbol{V}) \leftarrow (\boldsymbol{U}, \boldsymbol{V}) - \eta^{(t)} \cdot g$
5 $\quad$ Normalize column vectors of embedding matrices $(\boldsymbol{U}, \boldsymbol{V})$

---

**Algorithm 4:** SGD without replacement

**Input:** the number of positive pairs $N$, mini-batch size $B$, the number of mini-batches $k$, the number of epochs $E$, the learning rate $\eta$, initial embedding matrices: $\boldsymbol{U}, \boldsymbol{V}$

1 **for** $e = 1$ **to** $E$ **do**
2 $\quad$ Randomly partition the $N$ positive pairs into $N/B$ mini-batches: $\{\mathcal{B}_i\}_{i=1}^{N/B}$
3 $\quad$ **for** $j = 1$ **to** $N/Bk$ **do**
4 $\quad\quad$ Select $k$ mini-batch indices $S_{\mathcal{B}} = \{k(j-1)+1, k(j-1)+2, \ldots, kj\}$
5 $\quad\quad$ Compute the gradient: $g \leftarrow \frac{1}{k}\sum_{i \in S_{\mathcal{B}}}\nabla_{\boldsymbol{U}, \boldsymbol{V}}\mathcal{L}^{\mathrm{con}}(\boldsymbol{U}_{\mathcal{B}_i}, \boldsymbol{V}_{\mathcal{B}_i})$
6 $\quad\quad$ Update the weights: $(\boldsymbol{U}, \boldsymbol{V}) \leftarrow (\boldsymbol{U}, \boldsymbol{V}) - \eta \cdot g$
7 $\quad\quad$ Normalize column vectors of embedding matrices $(\boldsymbol{U}, \boldsymbol{V})$

---

### D.2 Ordered SGD (OSGD)

We consider two OSGD algorithms:

1. OSGD (Algorithm 5) with $k = \binom{N}{B}$ for the theoretical analysis in Section 5.1 and experimental results on synthetic datasets in Section 6.1.

2. OSGD *without replacement* (Algorithm 6) for experimental results on real datasets in Section 6.2, which is implemented for practical settings.

In the more practical setting where $\boldsymbol{u}_i = f_\theta(\boldsymbol{x}_i)$ and $\boldsymbol{v}_i = g_\phi(\boldsymbol{y}_i)$, OSGD updates the model parameters $\theta, \phi$ using the gradients $\frac{1}{k}\sum_{i \in S_\mathcal{B}} \nabla_{\theta,\phi} \mathcal{L}^{\mathrm{con}}(\boldsymbol{U}_{\mathcal{B}_i}, \boldsymbol{V}_{\mathcal{B}_i})$ instead of explicitly updating $\boldsymbol{U}$ and $\boldsymbol{V}$.

---
**Algorithm 5:** OSGD

---
**Input:** the number of positive pairs $N$, mini-batch size $B$, the number of mini-batches $k$, the number of iterations $T$, the set of learning rates $\{\eta^{(t)}\}_{t=1}^{T}$, initial embedding matrices: $\boldsymbol{U}, \boldsymbol{V}$

1 **for** $t = 1$ **to** $T$ **do**
2     Randomly select $k$ mini-batch indices $S_\mathcal{B} \subseteq \left[\binom{N}{B}\right]$ ($|S_\mathcal{B}| = k$)
3     Choose $q$ mini-batch indices $S_q := \{i_1, i_2, \ldots, i_q\} \subset S_\mathcal{B}$ having the largest losses i.e., $\mathcal{L}^{\mathrm{con}}(\boldsymbol{U}_{\mathcal{B}_i}, \boldsymbol{V}_{\mathcal{B}_i})$
4     Compute the gradient: $g \leftarrow \frac{1}{q} \sum_{i \in S_q} \nabla_{\boldsymbol{U},\boldsymbol{V}} \mathcal{L}^{\mathrm{con}}(\boldsymbol{U}_{\mathcal{B}_i}, \boldsymbol{V}_{\mathcal{B}_i})$
5     Update the weights: $(\boldsymbol{U}, \boldsymbol{V}) \leftarrow (\boldsymbol{U}, \boldsymbol{V}) - \eta^{(t)} \cdot g$
6     Normalize column vectors of embedding matrices $(\boldsymbol{U}, \boldsymbol{V})$

---
**Algorithm 6:** OSGD without replacement

---
**Input:** the number of positive pairs $N$, mini-batch size $B$, the numbers of mini-batches $k$ and $q$, the number of epochs $E$, the set of learning rate $\eta$, initial embedding matrices: $\boldsymbol{U}, \boldsymbol{V}$

1 **for** $e = 1$ **to** $E$ **do**
2     Randomly partition the $N$ positive pairs into $N/B$ mini-batches: $\{\mathcal{B}_i\}_{i=1}^{N/B}$
3     **for** $j = 1$ **to** $N/Bk$ **do**
4        Select $k$ mini-batch indices $S_\mathcal{B} = \{k(j-1)+1, k(j-1)+2, \ldots, kj\}$
5        Choose $q$ mini-batch indices $S_q := \{i_1, i_2, \ldots, i_q\} \subset S_\mathcal{B}$ having the largest losses i.e., $\mathcal{L}^{\mathrm{con}}(\boldsymbol{U}_{\mathcal{B}_i}, \boldsymbol{V}_{\mathcal{B}_i})$
6        Compute the gradient: $g \leftarrow \frac{1}{k} \sum_{i \in S_q} \nabla_{\boldsymbol{U},\boldsymbol{V}} \mathcal{L}^{\mathrm{con}}(\boldsymbol{U}_{\mathcal{B}_i}, \boldsymbol{V}_{\mathcal{B}_i})$
7        Update the weights: $(\boldsymbol{U}, \boldsymbol{V}) \leftarrow (\boldsymbol{U}, \boldsymbol{V}) - \eta \cdot g$
8        Normalize column vectors of embedding matrices $(\boldsymbol{U}, \boldsymbol{V})$

---

## E   Additional Experimental Results

In this section, we provide additional experimental results. First, we present histograms of mini-batch counts for different loss values from models trained with different batch selection methods. Next, we provide the results for $N \in \{4, 16\}$ on the synthetic dataset.

### E.1   Batch Counts: SC method vs. Random Batch Selection

We provide additional results comparing the mini-batch counts of two batch selection algorithms: the proposed SC method and random batch selection. The mini-batch counts are based on the mini-batch contrastive loss $\mathcal{L}^{\mathrm{con}}(\boldsymbol{U}_\mathcal{B}, \boldsymbol{V}_\mathcal{B})$ with $\tau = 1$. We measure mini-batch losses from ResNet-18 models trained on CIFAR-100 using the gradient descent algorithm with different batch selection methods: (i) SGD (Algorithm 4), (ii) OSGD (Algorithm 6), and (iii) the SC method (Algorithm 1). Figure 5 illustrates histograms of mini-batch counts for $N/B$ mini-batches, where $N = 50000$ and $B = 20$. The results show that mini-batches generated through the proposed spectral clustering method tend to contain a higher proportion of large loss values when compared to the random batch selection, regardless of the pre-trained models used.

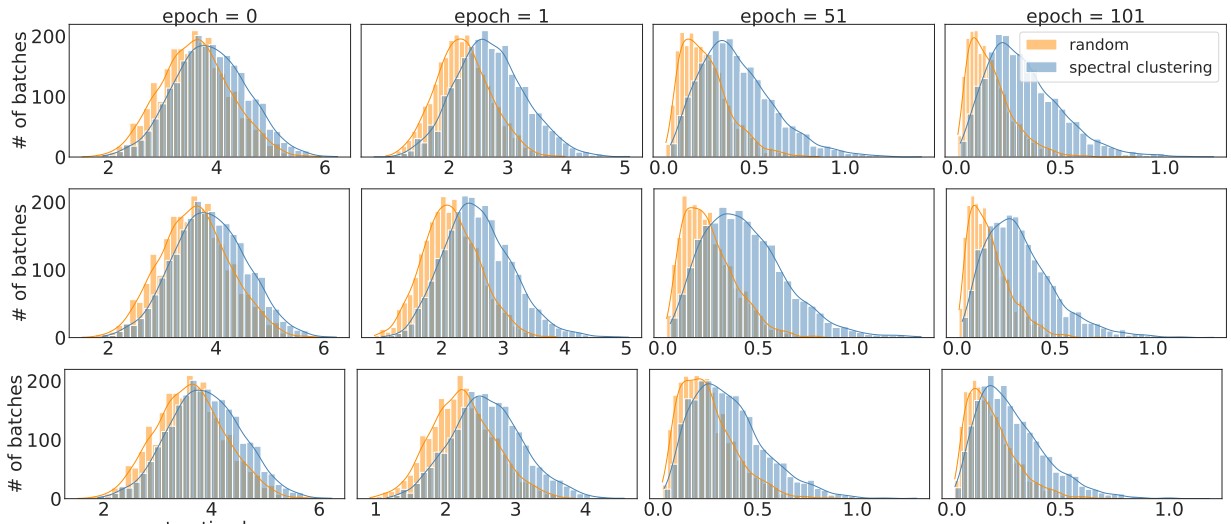

Figure 5: Histograms of mini-batch counts for $N/B$ mini-batches, for the contrastive loss measured from ResNet-18 models trained on CIFAR-100 using different batch selection methods: (i) SGD (Top), (ii) OSGD (Middle), (iii) SC method (Bottom), where $N$=50,000 and $B$=20. Each column of plots is derived from a distinct training epoch. Here we compare two batch selection methods: (i) randomly shuffling $N$ samples and partition them into $N/B$ mini-batches of size $B$, (ii) the proposed SC method given in Algorithm 1. The histograms show that mini-batches generated through the proposed spectral clustering method tend to contain a higher proportion of large loss values when compared to random batch selection, regardless of the pre-trained models used.

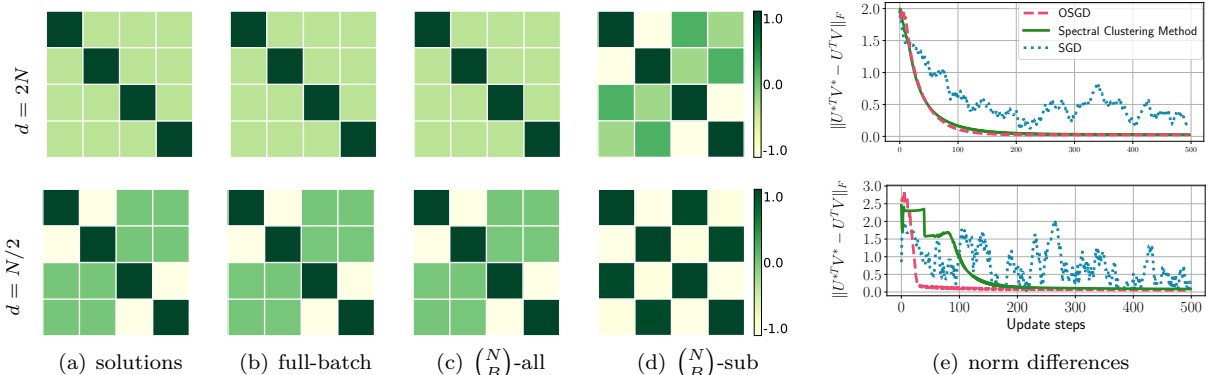

Figure 6: Heatmap of $N \times N$ matrix visualizing the resulting values from the same settings with Fig 4 except $N = 4$.

## E.2 Synthetic Dataset

With the settings from Section 6.1, where each column of embedding matrices $\boldsymbol{U}, \boldsymbol{V}$ is initialized as a multivariate normal vector and then normalized as $\|\boldsymbol{u}_i\| = \|\boldsymbol{v}_i\| = 1$, for all $i$, we provide the results for $N \in \{4, 16\}$ and $d = 2N$ or $d = N/2$. Figure 6 and 7 show the results for $N = 4$ and $N = 16$, respectively. We additionally present the results for theoretically unproven cases, specifically for $N = 8$ and $d \in \{3, 5\}$ (see Figure 8). The results provide empirical evidence that all combinations of mini-batches leads to the optimal solution of full-batch minimization for the theoretically unproven cases.

## F Computational Complexity

We compare the complexities of three batch selection algorithms: (i) SGD (Algorithm 4), (ii) OSGD (Algorithm 6), and (iii) the SC method (Algorithm 1). We first present the time complexity for $N/B$ batch selection for each algorithm using big-O notation:

- For SGD, batches are selected through just random shuffling resulting in $O(N)$.

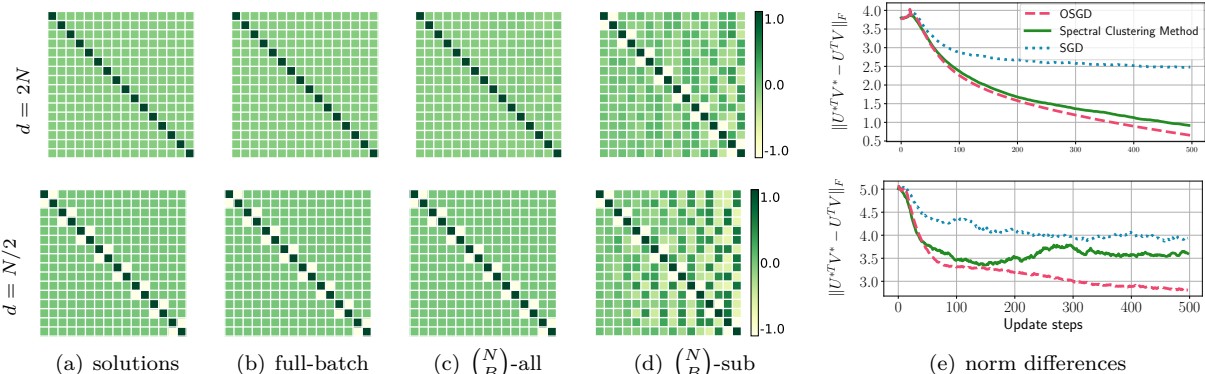

| (a) solutions | (b) full-batch | (c) $\binom{N}{B}$-all | (d) $\binom{N}{B}$-sub | (e) norm differences |

Figure 7: Heatmap of $N \times N$ matrix visualizing the resulting values from the same settings with Fig 4 except $N = 16$.

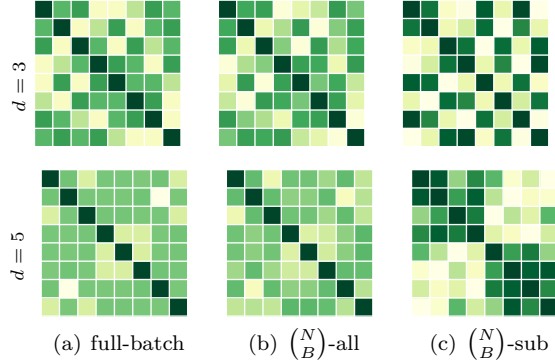

| (a) full-batch | (b) $\binom{N}{B}$-all | (c) $\binom{N}{B}$-sub |

Figure 8: Theoretically unproven setting. Heatmap of $N \times N$ matrix when $N = 8$ and $d < N - 1$.

- For OSGD, we randomly select $k$ batches, each of size $B$, and then measure the loss for each of these batches. After measuring, we sort the results and select the top-$q$ batches based on their losses. The computational complexity of evaluating the loss and sorting to generate $q$ batches is $O(kB^2 + k \log k)$. Consequently, the complexity for processing $N/B$ batches becomes $O((kB^2 + k \log k) \times N/Bq) = O((kB/q + k \log k/Bq)N)$.

- For the SC method, the primary computational bottleneck arises from the Hungarian algorithm, which operates at a complexity of $O(N^3)$. This complexity is the predominant factor influencing the overall performance of our algorithm. In order to accelerate batch selection in the proposed SC approach, we randomly partition $N$ positive pairs into clusters, each of size $kB$. We then subsequently apply the SC method on every $kB$-sized cluster instead of applying it directly on the entire $N$ pairs. This process generates $k$ mini-batches. In this context, the computational complexity for producing $k$ batches is $O(k^3 B^3)$. Extending this to generate $N/B$ batches, the complexity becomes $O(k^3 B^3 \times N/kB) = O(k^2 B^2 N)$.

We further report the wall-clock time required for the selection and update of $N/B$ batches using each algorithm on the CIFAR100 dataset, where $N = 50,000$ and $B = 32$ (see Section 6.2). The SC method requires an additional 11 minutes for batch selection compared to SGD (2 minutes) per epoch. As mentioned in Section 6.2, we suggest parallel batch selection using the SC method while updating models through gradient descent to mitigate this issue in practice. This approach involves using the same batches for multiple epochs as the next batch selection proceeds, based on the assumption that batches with higher loss values do not rapidly change.

