# OpenReview forum: "Mini-Batch Optimization of Contrastive Loss"
_TMLR — Accepted by TMLR_

### Review · Reviewer_oR4M · 2024-04-16

**Summary Of Contributions:**

In this paper the authors provide a theoretical study of the mini-batch optimization of the popular InfoNCE loss often used in contrastive learning. They start by carefully explaining the intrinsic difficulties of the mini-batch optimization in the contrastive learning setting consisting in the need to draw a prohibitively large number of batches. The paper further presents several theoretical results showing that 1) when all possible mini-batches are selected then the convergence to the full-batch solution is achieved, yet the losses of the two approaches differ; 2) the convergence is never achieved when some mini-batches are omitted. The authors further study a variation of the SGD called OSGD that selects the mini-batches having the highest contrastive loss and show its benefits over vanilla SGD in some cases. The experimental results on a toy examples and several real-world datasets confirm the utility of the provided theory.

**Audience:**

Yes

**Claims And Evidence:**

Yes

**Requested Changes:**

1.	Provide explanation regarding the assumption U=V in Definition 2.
2.	Provide a more comprehestive comparison to the approaches used to prove mini-batch convergence in our learning setups. Explain the differences in the techniques employed and the challenges that need to be overcome.
3.	Explain the improved empirical performance of SC vs. OSGD with a clear example or some sound intutions.

**Strengths And Weaknesses:**

**Strengths**

1.	A well-written study of an important open problem of mini-batch convergence when optimizing InfoNCE loss
2.	Interesting theoretical results regarding the soundness of mini-batch approach and its limits
3.	A practical approach for improving the mini-batch convergence with spectral clustering

**Weaknesses**

1.	The assumption about U=V seems rather strange in Definition 2. Is there any error here? I wonder how meaningful it is to use it for the two matrices and not for the minimizers as done in Lemma 1. It feels that this assumption is too strong to actually hold in practice, otherwise why not directly optimizing over U or V in isolation? Figure 4e confirms that U and V do not immediately behave as U* and V* and are quite far from them in the beginning of training.
	Given that several main contributions of this work rely on this assumption, it is hard to not to 	see it as a very limiting, unless the authors can provide more background on why it is not the 	case.
2.	I do not have a strong background in this field so maybe the results and the techniques of this paper are completely new but I would like the authors to elaborate more on why the proofs of mini-batch convergence in the contrastive learning setting are harder to obtain that similar such results in general SGD setting? The sentence just before the beginning of Section 3 hints on this but could be elaborated a bit more.
3.	From Figure 2, Figure 4e and Table 1 it is not immediately clear why SC outperformance OSGD in the end? It seems that OSGD converges to the oracle in Figure 2, in Figure 4e it convergence faster to the optimal U and V and yet SC improves over it in Table 1. Is there some explanation to it or a visualization showing where exactly SC improves compared to OSGD?

---

> ### Author Response · Authors · 2024-05-20
> **Authors' respond to the Reviewer oR4M**
>
> Thank the reviewer for recognizing the strengths of our work and valuable feedback. We believe that the clarifications provided in our response will effectively resolve any confusion and enhance your understanding of our study.
>
> `[R3-1] Regarding the assumption of U=V in Def. 2`
> > The assumption about U=V seems rather strange in Definition 2. Is there any error here? I wonder how meaningful it is to use it for the two matrices and not for the minimizers as done in Lemma 1. It feels that this assumption is too strong to actually hold in practice, otherwise why not directly optimizing over U or V in isolation? Figure 4e confirms that U and V do not immediately behave as U* and V* and are quite far from them in the beginning of training. Given that several main contributions of this work rely on this assumption, it is hard to not to see it as a very limiting, unless the authors can provide more background on why it is not the case.
>
> We want to clarify that this assumption is regarding the optimal solution, i.e., $U^{\star} = V^{\star}$. This assumption is reasonable in that after the optimization process, the embeddings of the positive pairs $(u_i^{\star}, v_i^{\star})$ align each other for all pair index $i$. We have clarified this in the revised manuscript.
>
> `[R3-2] Regarding convergence of SGD in contrastive learning`
> > I do not have a strong background in this field so maybe the results and the techniques of this paper are completely new but I would like the authors to elaborate more on why the proofs of mini-batch convergence in the contrastive learning setting are harder to obtain that similar such results in general SGD setting? The sentence just before the beginning of Section 3 hints on this but could be elaborated a bit more.
>
> We thank the reviewer for the insightful comment. Analyzing the convergence of the mini-batch contrastive learning has the following challenges. Recall that training with contrastive learning objectives relies on sample pairs categorized as either the positive pair or the negative pair. Although full-batch optimization can access to all positive/negative pairs, mini-batch loss contains the information on a subset of those pairs. Thus, the gradient computed for the mini-batch loss is an inaccurate estimate of the gradient for the full-batch loss.
>
> Fortunately, Theorems 4 and 5 in our paper proved the equivalence of the *optimal solution* of the mini-batch optimization and the full-batch optimization, assuming that all $N \choose B$ batches are loaded. To handle the complexity issue of loading $N \choose B$ batches, we instead come up with efficient algorithms for speeding up the convergence of mini-batch optimization. We have revised the manuscript to clearly mention this challenge and our contribution.
>
> `[R3-3] The performance of the proposed method compared to OSGD`
> > From Figure 2, Figure 4e, and Table 1 it is not immediately clear why SC outperformance OSGD in the end? It seems that OSGD converges to the oracle in Figure 2, in Figure 4e it convergence faster to the optimal U and V and yet SC improves over it in Table 1. Is there some explanation for it or a visualization showing where exactly SC improves compared to OSGD?
>
> We thank the reviewer for the sharp question. We want to clarify that Figure 2 and Figure 4e show the result for synthetic datasets, while the result of Table 1 is for real image datasets. For synthetic datasets, we could test the OSGD algorithm (as it is), but for real datasets, we could not run the original OSGD algorithm, due to the high complexity issue. Thus, as stated in the first paragraph of Sec.6.2, we used an *approximated* OSGD for real datasets, and in such cases, SC outperforms OSGD as in Table 1. We clarify this discrepancy in the experimental setup in the revised manuscript. To be specific, we have changed `OSGD` to `OSGD (approximated)` in the 2nd row of Table 1.

---

### Review · Reviewer_QzrE · 2024-04-30

**Summary Of Contributions:**

This paper studies the optimization for contrastive loss when using stochastic mini-batches. The contributions include
* The paper examines the theoretical aspects of optimizing mini-batches loaded for contrastive learning with the InfoNCE loss and shows that under certain parameter settings, mini-batch optimization is equivalent to full-batch optimization if and only if all N/B mini-batches are selected.
* The paper proposes a batch selection method based on the Ordered SGD algorithm and spectral clustering to efficiently select high-loss mini-batches, which can potentially improve the convergence rate of mini-batch contrastive loss optimization compared to vanilla SGD.
* Numerical experiments are conducted to support the theoretical arguments.

**Audience:**

Yes

**Broader Impact Concerns:**

No concern

**Claims And Evidence:**

Yes

**Requested Changes:**

1. Add discussion on the difference between optimizing $U, V$ and model parameters.
2. Add discussion about the convergence rate comparisons between SGD and OSGD in the setting of Theorem 7.
3. Add the approximation error between min-cut problem and the best $q$ mini-batches problem, and also provide a complete complexity analysis for OSGD.

**Strengths And Weaknesses:**

Strengths:

* This paper establishes the convergence guarantee for the stochastic version of InfoNCE loss, which is close to the practice setting.
* This paper proves the convergence of OrderedSGD and shows its faster convergence than SGD for optimizing InfoNCE loss.

Weaknesses:

* Proving the convergence of optimizing $U$ and $V$ in (1) is not that interesting to me. In practice, one may try to train the model parameters in the function $f_{\theta}(x)$ and $g_\phi(y)$. The training dynamic will be very different from training over $U$ and $V$.

* Besides, if only consider training $U$ and $V$, it is also not surprising that the optimal solution is ETF, as all vectors $u_i$'s and $v_i$'s are symmetric in the InfoNCE loss, as well as its mini-batch version.

* More discussions regarding Theorem 7 need to be added. The authors need to compare the convergence rates of SGD and OSGD in the setting of Theorem 7.

* The authors mention that finding the best $q$ loss among k batches is challenging, and can be kind of reduced to the min-cut problem. However, the authors provide the neither approximation error, nor the complexity for solving the min-cut problem, making it difficult to precisely evaluate the computational complexity of OSGD.

---

> ### Author Response · Authors · 2024-05-20
> **Authors' response to Reviewer QzrE**
>
> We would like to express our sincere gratitude for the constructive feedback and detailed suggestions, which helped us improve the manuscript. We provide point-by-point replies below.
>
> `[R2-1] Regarding the problem setting: directly optimizing the embeddings`
> > Proving the convergence of optimizing U and V in (1) is not that interesting to me. In practice, one may try to train the model parameters in the function f_\theat(x) and g_\phi(y). The training dynamic will be very different from training over U and V. Besides, if only consider training U and V, it is also not surprising that the optimal solution is ETF, as all vectors u_i's and v_i's are symmetric in the InfoNCE loss, as well as its mini-batch version.
>
> We thank the reviewer for the insightful question. We would like to remark that our choice of directly optimizing the embeddings $U, V$, so called the Unconstrained Features Model (UFM), is commonly used (e.g., [Graf et al. 2021], [Kini et al. 2023]) as noted in Section 3, in order to gain theoretical insights into the nature of contrastive learning.
>
> As per the reviewer's suggestion, we agree that it is better to analyze the cases of optimizing the parameters $\theta$ and $\phi$ used for forming the embeddings $U = f_\theta(X)$ and $V = g_\phi(Y)$. In Corollary 1 of our submitted manuscript, we included the analysis for such a case, assuming $f$ and $g$ are linear networks. We have mentioned this in the revised manuscript. In addition, proving the optimality of ETF on our mini-batch setting is not trivial, as explained in our response to [R1-1]. Besides, our main technical novelty is not in showing that the optimal embedding is ETF, but in (1) finding the relationship between the full-batch optimization and the mini-batch optimization, and (2) providing efficient algorithms for improving the performances of mini-batch optimization.
>
> `[R2-2] More discussions regarding Theorem 7`
> > More discussions regarding Theorem 7 need to be added. The authors need to compare the convergence rates of SGD and OSGD in the setting of Theorem 7.
>
> Thank you for your valuable feedback. Unfortunately, the loss function considered in Theorem 7 is tailored for OSGD, to allow the computed gradient to be an unbiased estimator. Under such a setting, the gradient computed for SGD is a biased estimator, thus the convergence analysis for SGD becomes tricky.  Instead, we compared the convergence of OSGD and SGD in Sec.5.1 (in Theorem 6 and Corollary 2), which supports our choice of using (approximated) OSGD for mini-batch contrastive learning.
>
> `[R2-3] Complexity of the proposed method`
> > The authors mention that finding the best loss among k batches is challenging, and can be kind of reduced to the min-cut problem. However, the authors provide the neither approximation error, nor the complexity for solving the min-cut problem, making it difficult to precisely evaluate the computational complexity of OSGD.
>
> Thanks for your great suggestion. We would like to first clarify that the complexity comparison is already provided in Appendix F of our manuscript: OSGD requires the computational complexity of $O(N^B\log N^B)$, while the complexity of our spectral clustering (SC) based algorithm is $O(k^2B^2N)$. We put this discussion in the main body during revision.
>
> We think the reviewer's suggestion of providing the approximation error is a great idea. We can compare the loss values of the batches selected by OSGD and our SC-based method. Due to the complexity of running OSGD, we could not get the result within the author response deadline. We are planning to provide this result once accepted.

---

### Review · Reviewer_geuj · 2024-05-05

**Summary Of Contributions:**

This paper studies the effect of mini-batching in the optimization of contrastive loss. This paper first compares the optimal embeddings of minimizing the full-batch and mini-batch contrastive losses, in which the authors prove that the optimal embeddings are identical if and only if all the possible mini-batches are involved in the objective function. Then, the authors studies the convergences of OSGD and SGD, which shows that OSGD can be faster than SGD in a toy example, and proposes a spectral clustering method to improve the minibatch selection problem of OSGD.

**Audience:**

Yes

**Claims And Evidence:**

Yes

**Requested Changes:**

See above

Other Typos and Questions:
- The first paragraph in sec. 5.1: geodesically non-quasi-convex
- The second paragraph in the Intro: $O(\mathcal{B}^2) = O(1)$, in the case of $\mathcal{B}=4096$, $\mathcal{B}^2$ is at the scale of $10^7=\Theta(N)$, how can we simply treat it as a constant?
- Theorem 6 is studying the convergences of two methods containing randomness, but there is no probablity or expectation in the statement, which has to be wrong.

**Strengths And Weaknesses:**

Strengths:
- The analysis seems solid and correct.
- Studying the effective of mini-batching in contrastive loss is important due to its prevalent usage.

Weaknesses:
- For the first part of the contribution, i.e., proving that the optimal embeddings are identical if and only if all the possible mini-batches are involved in the objective function, I am not so sure about the significance of this result. It seems straight forward to me when considering an objective function with a finite-sum structure. That is, comparing the optimal solutions of full-batch loss and the average of all possible mini-batch losses. Probably the interesting part is that the mini-batch loss is not a simple scaling of the full-batch loss by a constant factor in the case of constrative loss as stated in Proposition 1? I think the authors need to at least comment on the effect of randomness. Does the analysis imply that solving mini-batch constrative loss with SGD using uniformly sampled mini-batches would converge to the same optimal solution as GD solving the full-batch loss? Does the analysis imply that SGD using shuffling without replacement is a nice idea for the contrastive loss since it scans every possible mini-batch in an epoch? I am not sure about the significance or insight of the theoretical results in this part.
- For the second part of the contribition, i.e., showing that OSGD is better than SGD in a toy example with some theoretic supports, establishing the convergence of OSGD in contrstive loss, and proposing a spectral clustering-based OSGD variant with empirical advantages, I feel that the contribution is not closely connected to the first part, i.e., the story line is not so clear to me. The authors show that the problem is geodesically non-quasi-convex, so it is expected to be hard. This part confused me a lot. It seems to me that (mini-batch) SGD for solving general non-convex finite-sum problems has already been thoroughly studied (e.g., "Garrigos, Guillaume, and Robert M. Gower. "Handbook of convergence theorems for (stochastic) gradient methods." arXiv preprint arXiv:2301.11235 (2023)."). What is special in the mini-batch contrastive loss setting, that makes the analysis hard? Is it the unit norm constraint? For handling unit norm constraint, I remember there is a line of work proving convergence for Riemannian SGD in the non-convex case (e.g., "Hosseini, Reshad, and Suvrit Sra. "Recent advances in stochastic Riemannian optimization." Handbook of Variational Methods for Nonlinear Geometric Data (2020): 527-554."). I think it would be great if the authors could provide some discussion on the setup of contrastive loss, comparing to the standard finite-sum one and the riemannian finite-sum one.
- Does the Spectral Clustering-based OSGD enjoy the same convergence guarantee as the one in Section 5.2?

---

> ### Author Response · Authors · 2024-05-20
> **Authors' response to Reviewer geuj**
>
> We would like to thank the reviewer for the constructive feedback and spotting unclear points, which helped us improve the manuscript. We provide point-by-point replies below.
>
> `[R1-1] Contribution of our work (1/2)`
> > For the first part of the contribution, i.e., proving that the optimal embeddings are identical if and only if all the possible mini-batches are involved in the objective function, I am not so sure about the significance of this result. It seems straight forward to me when considering an objective function with a finite-sum structure. That is, comparing the optimal solutions of full-batch loss and the average of all possible mini-batch losses...
>
> We thank the reviewer for your valuable feedback. We respectively disagree with the reviewer's comment claiming that it is trivial to prove that the optimal solutions for full-batch loss and mini-batch loss are identical. As in Appendix C.1 (page 22) of the submitted manuscript, relating the full-batch loss and the mini-batch loss is done by Jensen's inequality, and the equality condition should be satisfied to show the equivalence of the optimal embeddings. This non-triviality of our technical contribution has beeen clearly stated in the revised manuscript.
>
> The reviewer also questioned about the significance and the implication of our theoretical results in Theorem 4 and 5. We believe our results guarantee that it is safe to focus on the mini-batch optimization problem instead of the full-batch problem, without losing the optimality. It is true that our results in Theorem 4 and 5 do not give insights on interesting research questions asked by the reviewer, regarding the training trajectory. Instead, we provide an analysis of the algorithmic aspects of mini-batch optimization in Sec.5, by delving into the convergence of OSGD. This technical novelty has been clearly stated in the revised manuscript.
>
> `[R1-2] Contribution of our work (2/2)`
> > For the second part of the contribition, i.e., showing that OSGD is better than SGD in a toy example with some theoretic supports, establishing the convergence of OSGD in contrstive loss, and proposing a spectral clustering-based OSGD variant with empirical advantages, I feel that the contribution is not closely connected to the first part, i.e., the story line is not so clear to me...
>
> The reviewer is exactly right that SGD for solving general non-convex finite-sum problems has already been thoroughly studied for convergence to a *stationary point*. We would like to clarify that our use of "hardness" refers to the difficulty of analyzing convergence to the *global optimum*, not the *stationary point*. We have revised the manuscript to clarify this point. However, it is worth proving that our method converges to the *stationary point*. Given the limited time given to the author's response, we could not finish this proof. We are planning to include the result once accepted, by using the proof techniques developed in the literature mentioned by the reviewer.
>
> The reviewer also questioned the storyline of our manuscript. In Sec. 4, we proved the equivalence of the full-batch optimization and the mini-batch optimization, once the latter counts the losses for all $N \choose B$ mini-batches. This shows that it is safe to rely on the conventional choice of mini-batch optimization, at the computational cost of loading all $N \choose B$ mini-batches. Motivated by this computational cost issue, in Sec.5, we come up with using OSGD instead of SGD, for the purpose of speeding up the convergence of the algorithm. The rest of Sec.5 is about justifying our choice of using OSGD, and speeding up the algorithm further by using the spectral clustering (SC) method. We have clearly mentioned this full storyline throughout the paper, by revising Sec. 4 and Sec. 5.
>
> `[R1-3] Convergence guarantee of the proposed SC algorithm`
> > Does the Spectral Clustering-based OSGD enjoy the same convergence guarantee as the one in Section 5.2?
>
> Thanks for the great question. Unfortunately, we cannot guarantee the convergence of the proposed Spectral Clustering (SC) method for the loss function in Theorem 7. Instead, we empirically demonstrate the efficacy of the proposed method by providing experimental results on synthetic (Fig. 4) and real datasets (Table 1).
>
> `[R1-4] Typo`
>
> Thank you for your careful reading. We have fixed it in the revised manuscript.
>
> `[R1-5] Big O notation for memory requirement `
> > The second paragraph in the Intro: $O({B}^2) = O(1)$, in the case of ${B}=4096$, ${B}^2$ is at the scale of $10^7=\Theta(N)$, how can we simply treat it as a constant?
>
> In practical scenarios, the maximum batch size does not scale with the dataset size $N$. The batch size cannot be increased beyond a specific size due to memory constraints, even as $N$ increases. Thus, we mainly focused on the Big-O notation with respect to $N$, and used the notation $B=O(1)$.

---

> > ### Author Response · Authors · 2024-05-20
> > **Authors' response to Reviewer geuj (continued)**
> >
> > `[1-6] Regarding randomness in Theorem 6`
> > > Theorem 6 is studying the convergences of two methods containing randomness, but there is no probablity or expectation in the statement, which has to be wrong.
> >
> > We apologize for the confusion. Although our submitted manuscript has a proper explanation in the text, we should have provided a more concrete notation. Indeed, there is an expectation in the definitions of $T_{SGD}$ and $T_{OSGD}$. These are defined as follows: Let $\theta_{OSGD}^{(t)}$ and $\theta_{SGD}^{(t)}$ be the model parameters at step $t$, when OSGD and SGD is used, respectively. Recall that the $\theta \in \mathbb{R}^N$.
> > We define $T_{SGD} = \min${$T \ge 0:   \mathbb{E}[\theta_{SGD}(T)] \in (\frac{\pi}{4} - \rho, \frac{\pi}{4})^N$ }, the minimal time required for the algorithms to reach the desired condition that each component of the expected value $\mathbb{E}[\theta(T)]$ falls within the interval $(\pi/4 − \rho, \pi/4)$. Simiarly, we define $T_{OSGD} = \min${$T \ge 0:   \mathbb{E}[\theta_{OSGD}(T)] \in (\frac{\pi}{4} - \rho, \frac{\pi}{4})^N$ }.
> > We have revised the manuscript to clearly show the definitions of $T_{SGD}$ and $T_{OSGD}$.

---

### Author Response · Authors · 2024-05-20
**Common response**

(R1 = R-geuj, R2 = R-QzrE, R3 = R-oR4M)

We sincerely thank the reviewers for their thoughtful and constructive feedback. We appreciate that the reviewers acknowledge the importance of studying mini-batching in contrastive learning, the solid analysis, and the practical implications of our proposed method.

As for the concerns/questions raised, we believe that we have adequately addressed all of them and replied in line with each review. We have revised the manuscript and highlighted the modified parts in blue text.

---

### Decision · Action_Editor_hWZu · 2024-06-10

**Recommendation:** Accept as is

**Comment:**

The paper studies aspects of a very important problem: mini-batch optimization for the widely popular contrastive losses. The paper proves the relationship between full-batch and the mini-batch optimization under some assumptions and provides efficient batch sampling algorithms for improved performance. The reviewers agree that the claims made in the paper are sufficiently supported with theoretical and some empirical evidence.

**Audience:**

The paper studies the very widely used InfoNCE loss so its findings are highly relevant to a wide set of cases and audience.

**Claims And Evidence:**

Reviewers and the AC mostly agree that the claims made in the submission are adequately supported with theoretical and empirical evidence.